# Combined CO$_2$ measurement record indicates Amazon forest carbon uptake is offset by Savanna carbon release

Santiago Botía[1], Saqr Munassar[1], Thomas Koch[1], Danilo Custodio[1], Luana S. Basso[1], Shujiro Komiya[2], Jost V. Lavric[3], David Walter[4], Manuel Gloor[5], Giordane Martins[6], Stijn Naus[10], Gebrand Koren[7], Ingrid T. Luijkx[8], Stijn Hantson[11], John B. Miller[6], Wouter Peters[8,9], Christian Rödenbeck[1], and Christoph Gerbig[1]

[1]Department of Biogeochemical Signals, Max Planck Institute for Biogeochemistry, Jena, Germany
[2]Department of Biogeochemical Processes, Max Planck Institute for Biogeochemistry, Jena, Germany
[3]Acoem GmbH, Hallbergmoos, Germany
[4]Multiphase Chemistry Department, Max Planck Institute for Chemistry, Mainz, Germany
[5]School of Geography, University of Leeds, Leeds, LS2 9JT, UK
[6]NOAA - Global Monitoring Laboratory, Boulder, Colorado, USA
[7]Copernicus Institute of Sustainable Development, Utrecht University, Utrecht, The Netherlands
[8]Environmental Sciences Group, Dept of Meteorology and Air Quality, Wageningen University and Research, Wageningen, The Netherlands
[9]University of Groningen, Centre for Isotope Research, Groningen, The Netherlands
[10]National Institute of Water and Atmospheric Research, Auckland, New Zealand
[11]Facultad de Ciencias Naturales, Universidad del Rosario, Bogotá, Colombia

**Correspondence:** Santiago Botía (sbotia@bgc-jena.mpg.de)

**Abstract.** In tropical South America there has been substantial progress on atmospheric monitoring capacity, but the region still has a limited number of continental atmospheric stations relative to its large area, hindering net carbon flux estimates using atmospheric inversions. In this study, we use dry air CO$_2$ mole fractions measured at the Amazon Tall Tower Observatory (ATTO) and airborne vertical CO$_2$ profiles in an atmospheric inversion system to estimate net carbon exchange in tropical South America from 2010 to 2018. We focus on the biogeographic Amazon, and its neighboring 'Cerrado & Caatinga' biomes. Considering all prior ensemble members, we estimate that the biogeographic Amazon was a net carbon sink with the sum of vegetation uptake, river outgassing and carbon release from fires at a median of -0.33 $\pm$ 0.33 PgC year$^{-1}$ (1-sigma posterior uncertainty). Using only process-based models as input in the inversion system the uptake is reduced to -0.24 $\pm$ 0.33 PgC year$^{-1}$. The 'Cerrado & Caatinga' biomes together represent a median carbon source of 0.31 $\pm$ 0.24 PgC year$^{-1}$, with contributions from both vegetation carbon release and fires. Therefore, we estimate a close-to-neutral net carbon exchange for tropical South America, but we note that the uncertainties straddle zero net exchange. In addition, we calculate the effect of systematic uncertainties in the inverse estimates by proposing a water-vapor correction to measured airborne CO$_2$ profiles. Assuming that the correction brings the observational data closer to the truth, implies that the Amazon is a weaker sink of carbon and that the 'Cerrado & Caatinga' is a larger source. We do not find a strong spatial shift of fluxes within the biogeographic Amazon due to the correction, nor do we find a strong impact on the interannual variations. Finally, to further reduce the uncertainty in

regional carbon balance estimates in tropical South America, we call for an expansion of the atmospheric monitoring network on the continent, mainly in the Amazon-Andes foothills.

## 1   Introduction

Land ecosystems constitute approximately half of the atmospheric $CO_2$ sink (Friedlingstein et al., 2022). However, they re-
main the most uncertain (Ballantyne et al., 2012) and variable (Le Quéré et al., 2018) component of the global carbon cycle.
Numerous independent studies confirm that tropical land ecosystems drive most of the interannual variability (IAV) in the
net land carbon flux (Keeling and Revelle, 1985; Keeling et al., 1995; Bousquet et al., 2000; Jung et al., 2011; Peylin et al.,
2013; Jung et al., 2017; Rödenbeck et al., 2018a; Bastos et al., 2020), which is linked to the atmospheric $CO_2$ growth rate
(Cox et al., 2013; Wang et al., 2013; Piao et al., 2020). Furthermore, tropical ecosystems store substantial carbon reserves
in aboveground ground living biomass (Brando et al., 2019), that can be released rapidly further amplifying the $CO_2$ growth
rate. In other words, the accumulation of $CO_2$ in the atmosphere heavily relies on the carbon uptake and release dynamics
occurring in tropical ecosystems. In particular, South America plays a pivotal role in both aspects, uptake and release, as it
hosts the Amazon rainforest, which contains 49% of tropical biomass carbon (Saatchi et al., 2011) and encompasses a third of
the continent landmass (Goulding et al., 2003).

The emissions of deforestation fires (van der Werf et al., 2010) and degradation (Assis et al., 2020) carbon sources are
particularly important for the Amazon region (Aragão et al., 2018; Matricardi et al., 2020; Qin et al., 2021; Kruid et al., 2021;
Lapola et al., 2023). In the years after 2017 to 2022, clear-cut deforestation increased significantly in Brazil (Alencar et al.,
2019; Silva Junior et al., 2021), not only releasing massive amounts of carbon (Assis et al., 2020) but also exposing larger areas
of forest fragments to degradation (Matricardi et al., 2020) and inducing indirect carbon losses by edge effects (Silva Junior
et al., 2020). In 2023, while deforestation decreased by 22% relative to 2022, there was a disproportionate rise of wildfires in
old-growth forests (Mataveli et al., 2024). Such threats to standing forests pose a risk of gradually releasing the carbon stock
of the Amazon, which amounts to 150-200 PgC (Saatchi et al., 2007; Malhi et al., 2009; Marques et al., 2017; Baccini et al.,
2017). Despite the attention to deforestation and degradation in the Amazon, vegetation loss has largely been overlooked in the
Cerrado biome (da Conceição Bispo et al., 2024), where agricultural expansion is more widespread (Rodrigues et al., 2022).
By 2019, 46% of the original land cover in the Cerrado was converted to pastures and crops (MapBiomas, 2020). Fires in the
Cerrado occur naturally and are crucial for ecosystem functioning, but agricultural expansion has brought more frequent and
intense fires threatening the aboveground biomass and creating the need for near-real-time monitoring (Pletsch et al., 2022).

Top-down atmospheric inversions exploit measured $CO_2$ gradients of an observational network to constrain the $F_{NetLand}$.
The balance between total uptake and release of carbon results in the $F_{NetLand}$, which consists of the Net Biome Exchange
(NBE) and the release of carbon from fossil fuel combustion. The NBE is composed of vegetation-related fluxes, Gross Pri-
mary Productivity (GPP) and Terrestrial Ecosystem Respiration (TER), disturbance-related emission from biomass burning
and degradation ($F_{fire}$), and river $CO_2$ outgassing ($F_{river}$). Estimates of $F_{NetLand}$ for tropical regions (Gurney et al., 2002;
Rödenbeck et al., 2003; Peylin et al., 2013; Liu et al., 2017; Gloor et al., 2018; Palmer et al., 2019; Crowell et al., 2019; Peiro

et al., 2022) and specifically for the Amazon region (Molina et al., 2015; Alden et al., 2016; Gloor et al., 2018; Wang et al., 2023) have been conducted previously using global inversions, typically operating under the assumption that sources from fires and fossil fuels are separately constrained or well known. Top-down studies have shed light on how fire influences the net carbon balance in specific years (2010-2011) (Gatti et al., 2014; van der Laan-Luijkx et al., 2015), the spatial differences of NBE in areas within the Amazon for 2010 to 2012 (Alden et al., 2016), the response of the Amazon carbon cycle relative to other tropical regions in the 2015/2016 El Niño event (Liu et al., 2017; Gloor et al., 2018; Crowell et al., 2019), and the main drivers of the NBE anomaly in 2015/2016 in the Amazon using satellite-constrained inverse estimates (Wang et al., 2023). At the time of these studies, the available data allowed only analyses of the response to a drought year and normal/wet years. The longer measurement records of in-situ $CO_2$ measurements at the Amazon Tall Tower Observatory (ATTO) (Botía et al., 2022) and the vertical $CO_2$ profiles in the Amazon region (Gatti et al., 2021) now enables the community to further examine interannual and seasonal variations using atmospheric inversions (Koren, 2020; Basso et al., 2023) or column budget techniques (Gatti et al., 2021, 2023).

Previous results, using a column budget technique suggested that the $F_{NetLand}$ of the eastern Amazon was on average a carbon source from 2010 to 2018, mainly explained by fire emissions, but also by a vegetation-related source in the southeastern part of the Amazon forest (Gatti et al., 2021). The authors attribute the carbon source to the combined impact of temperature and precipitation anomalies on vegetation, hindering its carbon uptake capacity. Basso et al. (2023) reported a smaller carbon source compared to Gatti et al. (2021) for the same period (2010-2018), using similar $CO_2$ observational constraints but using an atmospheric transport inversion and integrating additional CO observational constraints on fire emissions. Both studies agree on the calculation of a net carbon source (positive $F_{NetLand}$) and a small vegetation-related carbon sink after subtracting fires and fossil fuels (or assuming them negligible). In recent years, a potential bias in $CO_2$ observations collected from the aircraft network using nondried air samples has been recognized (Baier et al., 2020; Gatti et al., 2023; Basso et al., 2023). The presence of water vapor in the collected samples would lead to a loss of $CO_2$ in condensed water in the pressurized flasks. This can lead to an underestimation of fluxes when using this dataset as a constraint on the amount of $CO_2$ that the Amazon exchanges, with a possibility for seasonal biases due to the higher water vapor present during the wet season. However, the effect of water vapor in the aircraft samples and how that propagates to optimized fluxes has not been quantified, adding additional uncertainty to the Amazon-wide carbon budget.

Other studies using local forest plot data that is up-scaled to the Amazon found that old-growth forests are a small carbon sink with a decreasing trend over the last 30 years (Brienen et al., 2015; Hubau et al., 2020). Although these studies are broadly consistent with the existence of an Amazon-wide vegetation carbon sink and net biomass growth in old-growth plots, substantial uncertainties persist regarding their magnitudes. This agrees with a recent synthesis by Rosan et al. (2024) who reported an Amazon-wide carbon sink of $342 \pm 192$ TgC year$^{-1}$ from vegetation uptake over the period 2010-2018, albeit with considerable inter-annual variability. Although all of these studies have investigated the carbon exchange of the Amazon, they each had different definitions in area, be it the biogeographic or a selection of sub-regions defined by Eva et al. (2005). Moreover, a perspective including the neighboring biomes (i.e., 'Cerrado & Caatinga') is lacking.

In this study, we use the CarboScope Global and Regional inversion system to assimilate the 2010-2018 airborne $CO_2$ profile record and the continuous and long-term $CO_2$ record at the Amazon Tall Tower Observatory (ATTO). We build on previous studies using the CarboScope Regional system in Europe (Kountouris et al., 2018b, a; Munassar et al., 2021), to explore its ability to constrain the $F_{NetLand}$ at the continental scale over a larger domain, but with a sparser observational network. The study is structured as follows. First, we aim to quantify where the atmospheric inversion using this set of atmospheric data can provide a constraint based on uncertainty reduction. Second, a sub-continental analysis of the carbon budget, with a strong focus on the biogeographic Amazon, but not limited to it, is performed to shed light on spatial gradients. Last, we present a detailed quantification of how systematic uncertainties in measured mole fractions in the aircraft network affect the estimated fluxes in an atmospheric inversion. With this study, we provide a broad perspective on carbon exchange in tropical South America, going beyond the Amazon biome and highlighting where we need to expand our observational efforts to reduce the uncertainty in carbon exchange estimates in the region.

## 2 Methods and Data

### 2.1 CarboScope Regional Inversion System

#### 2.1.1 Two-step scheme description

The version of the CarboScope Global inversion system used here is described in detail by Rödenbeck et al. (2018b). To refine the resolution of fluxes and atmospheric transport within our study area, we use the two-step scheme described in several publications previously (Rödenbeck et al., 2009; Trusilova et al., 2010; Kountouris et al., 2018b; Munassar et al., 2021). Our inversion set-up follows largely that of Kountouris et al. (2018a), but we use an isotropic exponential decay for the spatial error structure, mainly because in our domain, unlike in mid-latitudes, the climatic gradients are similar in both latitude and longitude.

In the two-step scheme (see Figure 1), two atmospheric transport models with different spatial resolutions are used. In step 1, a global inversion is performed using the CarboScope Global inversion system (Rödenbeck et al., 2018b) to obtain an optimized NBE flux field having Ocean and Fossil Fuel fluxes prescribed. This global inversion is performed on a coarse global scale using the TM3 atmospheric transport model (Heimann and Körner, 2003) at 4 x 5 deg resolution driven by the NCEP reanalysis meteorological fields (Kalnay et al., 1996). Using that optimized NBE flux field and the same atmospheric transport set-up, simulated mole fractions increments for all sites are obtained, except the site left for validation (i.e. the s10 station set plus the South American stations, see Sect 2.1.4). These "forward" runs represent an intermediate step and are done twice, see equation 1 and Figure 1 (adapted from Rödenbeck et al. (2009)). The first one is performed using TM3 at the coarse global resolution and for the entire time-period of the global inversion. The second forward run is performed using NBE fluxes at coarse spatial resolution, only for the regional domain (i.e. with zeros outside the regional domain) tropical South America and the desired period of interest (i.e., 2010-2018). Both forward runs result in simulated mole fraction increments ($\Delta c_{mod1}$ and $\Delta c_{mod2}$) and their difference corresponds to the far-field contribution from fluxes outside of the regional domain. An initial

condition ($c_{ini}$) that corresponds to a well-mixed atmosphere with a given initial tracer mole fraction is then added to the far-field contribution. All this together is subtracted from the measured mole fractions at the sites within the domain of interest. This difference represents a "remaining mole fraction" ($\Delta c_{remain}$), corresponding to signals from fluxes within the regional domain, as defined by Rödenbeck et al. (2009).

$$\Delta c_{remain} = c_{meas} - (\Delta c_{mod1} - \Delta c_{mod2} + c_{ini}) \tag{1}$$

In step 2, the $\Delta c_{remain}$ is assimilated in a regional inversion at high-resolution (0.25 x 0.25°) using the model STILT (Lin et al., 2003). STILT is driven by the ECMWF Integrated Forecasting System (IFS) (following the contemporary IFS cycle development; for more information, see https://www.ecmwf.int/en/publications/ifs-documentation). At each measurement location and time (x, y, z, t) in the regional domain, we released an ensemble of 100 particles back in time (10 days) to calculate the surface influence on the observations. The surface influence, with units of $ppm/(\mu mol m^{-2} s^{-1})$, provides the link between measured mole fractions and the prior surface fluxes. A set of different prior fluxes (representing a prior ensemble) is used (see Section 2.1.6), and the regional inversion is performed for each prior ensemble member individually. The domain in this study extends from 28.875°S to 13.875°N and from 83.875°W to 34.125°W, see Figure 2. We have limited our domain to 28.875°S due to the lack of observational records further south.

### 2.1.2 Definition of CO$_2$ flux components

The total CO$_2$ exchange with the atmosphere is denoted as Net Land Flux ($F_{NetLand}$), which consists of Net Biome Exchange (NBE) and fossil fuel CO$_2$ emissions ($F_{ff}$),

$$F_{NetLand} = NBE + F_{ff} \tag{2}$$

NBE, in turn, is composed of Net Ecosystem Exchange (NEE), the carbon exchange between atmosphere and rivers ($F_{river}$), and emissions from fires ($F_{fire}$),

$$NBE = NEE + F_{river} + F_{fire} \tag{3}$$

For all fluxes we adopt the atmospheric sign convention in which positive (+) denotes a source to the atmosphere and negative (-) denotes a sink.

The atmospheric signals reflect the total CO$_2$ flux, $F_{NetLand}$. However, as $F_{ff}$ is prescribed as a prior the inversion optimizes NBE. In the figures in which the posterior $F_{NetLand}$ is reported, we have added the $F_{ff}$ in post-processing. When subtracting $F_{fire}$ from the NBE we obtain a flux component composed of NEE and $F_{river}$, reported in Section 3.2.

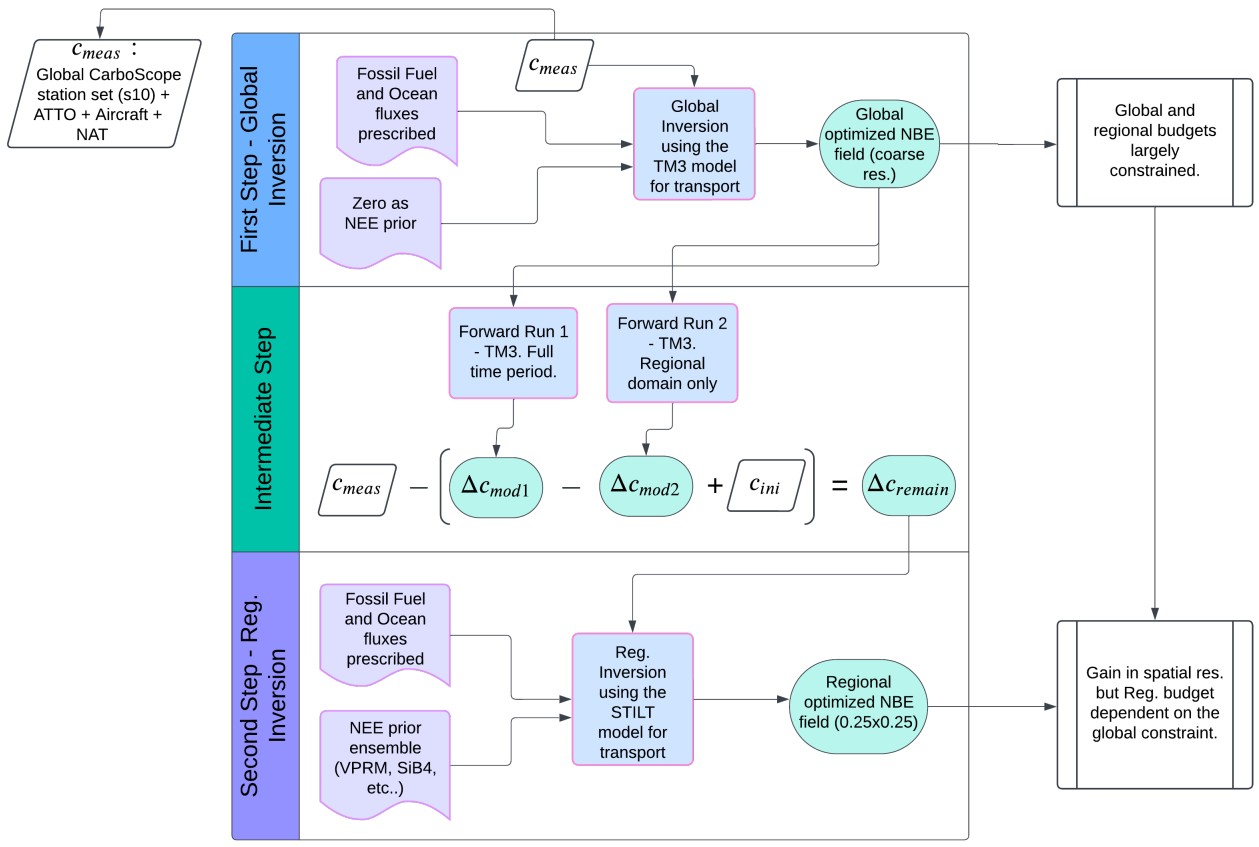

**Figure 1.** CarboScope Regional Inversion Two-step scheme (Rödenbeck et al., 2009; Trusilova et al., 2010) flow diagram showing the inputs (purple polygons) to the processes (blue squares) and their specific output (green circles).

### 2.1.3 Analysis regions in postprocessing

To analyze the spatial distribution of the estimated posterior fluxes and the coverage of the atmospheric network, we have used the definition of sub-regions shown in Figure 2a. One of the criteria for the choice of these regional areas is that they should be independent of our observational network and associated with a biogeographic gradient. Thus, the selection of these areas follows, to some extent, a biome-like distribution, but it does not represent individual biomes strictly. The division within the biogeographic Amazon serves to provide individual sub-regions dividing east-west but also north-south. In addition, we have also kept a separate sub-region for the main branch of the Amazon river as we are interested in the atmospheric constraint over these areas. The biogeographic limit of the Amazon we use here is the same as defined by Albert et al. (2021) as Amazon biome. When reference to the Amazon *sensu-stricto+*, the area corresponds to four subregions defined in Eva et al. (2005): Amazon *sensu stricto*, Andes, Guiana and Gurupi. The Amazon *sensu-stricto+* area is only used for comparative reasons in the

Discussion. As we are using a regional inversion with 0.25 x 0.25 degree of spatial resolution with a correlation length of about 200 km, the posterior budgets as well as posterior uncertainties for these subregions within the Amazon can be individually quantified.

### 2.1.4 Observational network

The location of the measurement sites is shown in Figure 2b, together with their aggregated annual mean surface influence (sensitivities of atmospheric concentrations to surface fluxes, in the following called "footprint") from 2010 to 2018. The coverage of the observational network in our regional domain concentrates on the areas within the Amazon but also in the northeast of Brazil. For the seasonal footprints for each site see Figure A1.

In the global inversion (step 1) we have used the set of stations in the 2022 release version of the CarboScope global system with nearly continuous coverage from 2010 onward (i.e., s10oc_v2022, see http://www.bgc-jena.mpg.de/CarboScope/ ?ID=s10oc_v2022, for details of stations and data providers). To this default 2022 station set, we added the ATTO $CO_2$ record (available at: https://attodata.org/), five sites within the Amazon region where airborne profiles (available at: https: //doi.org/10.1594/PANGAEA.926834) are collected (Gatti et al., 2021), and the weekly flask sampling record in Natal (NAT), a station in the northeast of Brazil (Dlugokencky et al., 2021). While the global inversion (s10, step 1) is augmented with all of the above sites (s10 + ATTO + NAT + aircraft), only the sites within our domain are used for the regional inversion (step 2). The monthly time series at each site is shown in Figure 2c, indicating the data gaps and the evolution of $CO_2$ over the last decade. Note that ATTO has provided continuous data since 2013 from an 80-m tower, $\approx 50$ m above the canopy, and there were major gaps in the aircraft network during 2015 and 2016. For the continuous data (ATTO), we use only daytime measurements (i.e., from 13:00 to 17:00 local time), to ensure we have measurements representative for the well-mixed boundary layer when the transport model errors are smallest (see Fig S4, Supplementary material in Botía et al. (2022)). The average number of aircraft profiles per month is two, see Figure A2. For each measurement in an aircraft profile (full profile goes up to $\approx$4500 m a.s.l), for the weekly flask measurements at NAT, and for every single data point at ATTO, we have simulated the surface influence using the STILT model. Therefore, each measurement has an individual footprint linking the observations with surface fluxes in the regional inversion. The STILT set-up follows that of Botía et al. (2022), but the spatial resolution used here is 0.25x0.25 deg. For validating the estimated posterior fluxes we use airborne data from the Manaus site (MAN) (Figure 2), which was not assimilated and is left as an independent site. The information gained from using the South American stations (Aircraft profile + ATTO) is tested using CarboScope global with (s10sam) and without (s10) these stations.

The model-data mismatch uncertainty (including the representation error of the measurements within the transport model) for the three types of sites (in-situ tower, aircraft, and weekly flasks) is chosen to be 1.5 ppm for weekly time scales, following common practice in CarboScope global (Rödenbeck, 2005; Rödenbeck et al., 2018a) which assimilates a large set of weekly flask samples. To assimilate multiple data streams, we apply a data density weighting Rödenbeck (2005): For the hourly ATTO data, the error will be inflated by $\sqrt{N_{hours/week}}$ (details see Kountouris et al. (2018a)), while for aircraft profiles (composed of several flasks) the error is scaled with $\sqrt{N_{flasks/profile}}$. The data-density weighting practically ensures that one week of

hourly ATTO observations, one aircraft profile, or one weekly flask sample have the same weight in the inversion, reflecting
the assumption that they provide the same amount of information due to roughly weekly error correlations.

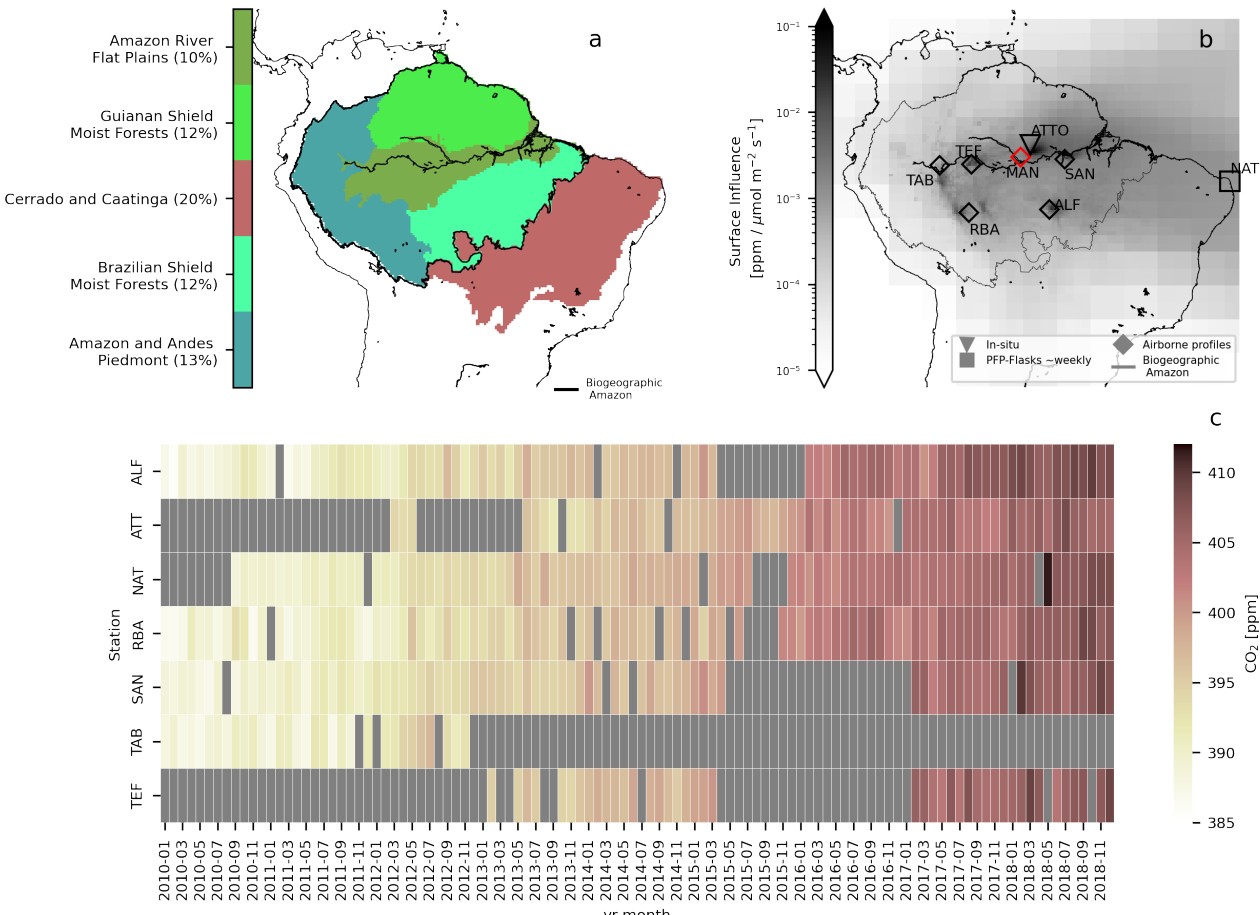

**Figure 2.** South American domain (from 13.875$^{\circ}$N to 28.875$^{\circ}$S and 83.875$^{\circ}$W to 34.125$^{\circ}$W ) with a sub regional division adapted from EPA (2011) (a). The percentages in the legend (adding to 67%) denote the area of the region relative to the total land area in the domain, the remaining 33% corresponds to the white areas. The annual mean footprint, associated with the period where data are available is shown in b. Note that MAN is not assimilated and is used as a validation site. The time series showing the monthly mean $CO_2$ mole fraction at each station or aircraft profile site is shown in c. The gaps are shown in grey and are due to logistical problems and/or instrument malfunction. The Amazon contour corresponds to the biogeographic limit of the Amazon.

### 2.1.5 Addressing systematic uncertainties in vertical profiles

Gatti et al. (2023) (Methods Section) reported a source of uncertainty in the aircraft profiles given by moisture in undried flask samples at time of collection. Such moisture can lead to biases in the measured $CO_2$ mole fractions. The bias manifests itself as an underestimate of the true mole fraction of atmospheric $CO_2$ in the measurements, due to the removal of gaseous $CO_2$ into water between the time of sampling at the aircraft inlet and analysis in the lab. Both liquid water and condensation of water vapor after pressurizing the flasks form the source for this $CO_2$ absorption and thus a relation with water vapor mole fraction exists (hereafter $xH_2O$), as described specifically for this sampling method by Baier et al. (2020) and Paul et al. (2020). Moreover, systematic errors in assimilated data can bias inverse results in various ways (Masarie et al., 2011). To quantify the effect of such systematic uncertainty on the estimated fluxes, we apply a water-vapor-dependent bias correction to the data, before it is assimilated.

To derive the water vapor bias correction, we used $CO_2$ vertical profiles collected approximately 80 km northeast of the city of Manaus in the central Amazon basin (site code MAN; Miller et al. (2023)) aboard a Cessna Grand Caravan. Collocated portable flask packages (PFPs; which have a water-related $CO_2$ bias) and in-situ, on-board measurements (Picarro Inc. CRDS analyzer model G-2401m) were performed (see Figure A4). The in-situ analyzer also provided measurements of $CH_4$, CO, and $H_2O$ roughly every 3 seconds. The MAN time series of vertical profiles, extending from approximately 150 m above ground level (agl) to approximately 5 km above sea level (asl) began in April 2017 and continues to this day at a frequency of approximately twice per month. The in-situ gas stream was undried (but $CO_2$ mole fractions have been converted from wet to dry using the simultaneously measured water vapor), and calibrations occurred on the ground. Previous results have shown this kind of implementation has minimal ($\approx < 0.1$ ppm) impact on $CO_2$ values (Rella et al., 2013). Critically, the fact that the air stream was undried allows us to use the analyzer's water vapor measurement. The PFPs used were version 2 models and $CO_2$ measurements on PFP air samples were performed at INPE LaGEE; the sampling and analysis methodologies have been described previously (Gatti et al., 2014).

Between April 2017 and June 2019, when air sample collection via PFP stopped at MAN, there were 12 vertical profiles available that had collocated PFP discrete samples and in-situ measurements that could be compared. These 12 profiles span a range of water vapor mole fraction from 0.2 to 3.5 % that decrease with altitude (see Figure A5) and allow us to define a relationship between $CO_2$ bias ($\Delta CO_2 = CO_{2_{in-situ}}$ - $CO_{2_{PFP}}$) vs. ambient water vapor. Matching between the independent PFP and on-board analyzer systems could be performed accurately by matching each system's GPS time and position signals. Because the PFP sample is collected over $\approx$15 - 40s (depending on altitude), we tested a wide range of time averaging windows for the in-situ analyzer, ranging from 10 s to 40 s to best match the single value of $CO_2$ measured in the laboratory from the PFP air sample. Fortunately, this averaging window range had a negligible impact on the final relationship between $\Delta CO_2$ and water vapor (see Figure A3); the results presented here use a window of 30 s. We also filtered Picarro data for high variability by excluding any 30 s period where the $CO_2$ standard deviation was greater than 0.5 ppm. Finally, we binned all $\Delta CO_2$ and $xH_2O$ values in 10,000 ppm (or 1%) increments to define a bias relationship. This relationship (Figure A3) shows little bias between 0 and 1% (above 3 km agl), but an increasing, approximately linear bias with increasing $xH_2O$ (below 2 km agl).

We used the STILT model to extract the water vapor mole fraction from the ECMWF-IFS short-term forecasts (as also used
to drive STILT) at the same measurement and sampling locations in the Manaus flights in which PFPs and the CRDS analyzer
were collocated. These flights span altitudes from 280 m agl to 5200 m agl and vertical profiles are taken at $\approx$ 17:00 UTC. The
correlation between the measured water vapor (CRDS analyzer) and that extracted from ECMWF-IFS was 0.94 (p<<0.01,
N=158) (see Figure A5), with a mean bias of -0.2% (ECWMF-IFS biased low relative to the continuous measurements). The
observational record at MAN includes both dry and wet seasons, and in combination with a large altitude range covers the
range of water vapor variability at the other vertical profile sites (see Figure A6). The latter gives us confidence in ECMWF-
IFS as a proxy for the water vapor mole fraction at the time and location (x, y, z) of each flask sample at ALF, RBA, TEF, TAB,
SAN. Therefore, the difference between the dry air $CO_2$ mole fraction from the continuous measurements (using the wet-dry
correction as proposed by Rella et al. (2013) and the PFP on the Manaus flights was fitted to the water vapor mole fraction
from ECMWF-IFS. The linear fit is $y = 0.594x - 0.168$, where $y$ is the resulting $CO_2$ increase [ppm] to be applied to the PFP
data at a given water vapor mole fraction $x$ [%], the slope (0.594) was significantly (p<<0.01, N=158) different from zero
(See Figure A7). The standard error of the slope and the intercept is $\pm$ 0.11 and $\pm$ 0.20, respectively. Note that the negative
values in Figure A7 are due to the variable nature of the atmosphere causing uncertainty in both measurement strategies. We
discard the possibility of being a mismatch in different air parcels sampled because of the collocated sampling lines and the
tests we have done with different averaging times with the Picarro data (Figure A3). Note that these negative values are more
frequent at low water vapor concentrations, which occur more often in the free troposphere (Figures A4 and A5). We have
further quantified the expected error in the PFP mole fractions by the calculating the $\sqrt{\frac{SSR}{N-2}}$. Introducing the correction we
obtain 0.93 ppm and without the correction 1.30 ppm. The sum of squared residuals (SSR) with the correction is obtained
using the linear fit reported previously, and N is the number of measurement points (N=158). In the case without the correction
the SSR is calculated by obtaining the residuals with respect to a line with no slope and intercept at zero in Figure A7. As
introducing the correction leads to a lower $\sqrt{\frac{SSR}{N-2}}$ than without it, it indicates that by applying the water vapor correction we
are decreasing the measurement error.

    Using this function and the ECMWF-IFS-water-vapor at ALF, RBA, TEF, TAB, SAN we applied the water-vapor correction
to each individual PFP sample (see Figure A8). Note that the uncertainty added by the accuracy of the water vapor mole fraction
given by the ECMWF-IFS forecast is dealt with using the fit of the $\Delta CO_2$ to the ECMWF-IFS water vapor. At RBA there were
6 flights (dates: 16/02/2018, 09/03/2018, 06/04/2018, 08/05/2018, 15/07/2018, 19/09/2018) that had a drier installed, so for
these flights we did not apply the correction. With this, we re-ran the inversion system (Global and Regional) assimilating the
data including the water vapor correction, and quantified the potential effect of such systematic uncertainty on the posterior
fluxes (hereafter s10samwvc, for CarboScope global). We note that this approach was implemented to diagnostically quantify
the effect of water vapor in the PFPs and how that propagates to the estimated fluxes in an inversion system. The effects of
water vapor on not-dried flask samples has been established and documented previously (Baier et al., 2020; Paul et al., 2020).
Here, we establish the offsets on this specific set of flask samples collected over the Amazon. The offsets used in this study are
provided as a public dataset (https://edmond.mpg.de/privateurl.xhtml?token=a6fba176-8a6b-4b59-a371-3acc804adaf9), such
that the community can use them in their inversion systems and compare their magnitude to other correction methods.

## 2.1.6 A-priori fluxes

We present an ensemble of inversions based on a set of prior fluxes that differ in several aspects (see Table 1). The first aspect is their conceptual nature, having (a) the Vegetation Photosynthesis and Respiration Model (VPRM) (Mahadevan et al., 2008), a simple diagnostic model using MODIS imagery and fitted to eddy covariance data (Saleska et al. (2013), see Table A2 for the site descriptions) within the domain which provides NEE; (b) the FLUXCOM product (Bodesheim et al., 2018) and its latest version X-BASE$_{NEE}$ (Nelson et al., 2024), which up-scales site-level eddy covariance data to the globe using a random forest regression; and (c) two process-based models, the SiB4 (Haynes et al., 2019) and SiBCASA (Schaefer et al., 2008) models, both having served as biospheric flux priors in earlier published studies focusing on the Amazon (van der Laan-Luijkx et al., 2015; van Schaik et al., 2018). By having this set of priors we have a wide representation of the potential spatio-temporal dynamics of NEE over our domain. The other variable aspect of this prior selection is their flux magnitudes and seasonal patterns (see Figure A9 and A11). Originally the eddy-covariance-based products, like the two FLUXCOM versions and VPRM, have a large sink magnitude for the Amazon. Note that the total land flux in the Amazon is highly uncertain, spanning from -0.34 to 0.29 PgC year$^{-1}$ (Gatti et al., 2021; Rosan et al., 2024), but this range gets larger than 1 PgC considering the uncertainties associated with each estimate, thus we decided to keep the eddy-flux based prior NEE products (VPRM and FLUXCOM), as they can be considered as a plausible first guess in an inversion. Furthermore and regardless of how they compare to current independent estimates we proceeded to make an experiment scaling two of our priors (i.e. VPRM and SiB4) such that NEE = 0.5 and 1 PgC, and thus we can test an opposing (in sign) prior scenario. To achieve this, we scaled ecosystem respiration in VPRM and SiB4 such that the total NEE integral for the biogeographic Amazon equals 0.5 PgC year$^{-1}$ and 1 PgC year$^{-1}$ (namely VPRM-0.5Pg, VPRM-1Pg, SIB4-0.5Pg, and SIB4-1Pg). An example for VPRM-0.5Pg is shown in Figure A9. Two additional sensitivity tests were performed using the original VPRM. In one, we removed the long-term mean, seasonality, and interannual variability (IAV) from VPRM (called VPRM$_{flat}$) and run the inversion only with a diurnal cycle in the prior. In the second one, we used VPRM as prior but left the ATTO data out from the assimilated station set (called VPRMnoATT).

Note that the CarboScope global v2022 has no diurnal cycle in the prior fluxes. The effect of this and how it propagates to the regional inversion is discussed in Munassar et al. (2024). Here, we apply a region-specific correction based on this work. The correction is based on the response of CarboScope global to having a diurnal cycle in the prior fluxes, which was derived by inverting the diurnal anomalies (hourly - dailyMean) of two forward runs (one with and the other without diurnal cycle) using CarboScope Global. The prior flux diurnal cycle was based on FLUXCOM. The posterior fluxes of such inversion correspond to the per-grid-cell correction that should be added to the posterior fluxes of the normal inversion, assuming that the FLUXCOM diurnal cycles are correct. Then, we propagate such a correction using the two-step scheme described above via the lateral boundary conditions.

The fire emissions used in this study (GFAS-opt $CO_2$) are based on the original GFAS product (Kaiser et al., 2012), with an adjustment over the Northern part of South America based on CO inversions by Naus et al. (2022). These inversions were performed using the TM5-4DVAR system (Krol et al., 2005; Meirink et al., 2008) using CO data from MOPITT (Deeter et al., 2019) inside the study area (Northern part of South America) and NOAA station data outside of this region. We calculated

**Table 1.** Inversion runs indicating the type of inversion (Global or regional), the selection of prior NEE fluxes used in the regional inversion and the label identifying each individual run. All these individual runs except VPRMnoATT, represent the prior ensemble. IAV stands for interannual variability and $R_{eco}$ for ecosystem respiration. The s10 station set has nearly continuous coverage from 2010 onward.

| Identifier | Station Set | Type | Comment |
|---|---|---|---|
| s10 | s10 | Global Inv. | |
| s10sam | s10+Aircraft+NAT+ATTO | Global Inv. | |
| VPRM | s10+Aircraft+NAT+ATTO | Regional Inv. | |
| VPRM$_{flat}$ | s10+Aircraft+NAT+ATTO | Regional Inv. | only with diurnal cycle |
| VPRM-0.5Pg | s10+Aircraft+NAT+ATTO | Regional Inv. | $R_{eco}$ scaled to get 0.5 PgC NEE |
| VPRM-1Pg | s10+Aircraft+NAT+ATTO | Regional Inv. | $R_{eco}$ scaled to get 1 PgC NEE |
| VPRMnoATT | s10+Aircraft+NAT | Regional Inv. | |
| FLUXCOM | s10+Aircraft+NAT+ATTO | Regional Inv. | |
| X-BASE$_{NEE}$ | s10+Aircraft+NAT+ATTO | Regional Inv. | |
| SiBCASA | s10+Aircraft+NAT+ATTO | Regional Inv. | |
| SiB4 | s10+Aircraft+NAT+ATTO | Regional Inv. | |
| SIB4-0.5Pg | s10+Aircraft+NAT+ATTO | Regional Inv. | $R_{eco}$ scaled to get 0.5 PgC NEE |
| SIB4-1Pg | s10+Aircraft+NAT+ATTO | Regional Inv. | $R_{eco}$ scaled to get 1 PgC NEE |

the increment ratios from prior (GFAS) to posterior CO fluxes for each grid cell and applied this factor to the GFAS $CO_2$ fire
emissions as previously done in Koren (2020). For regions outside the domain of Naus et al. (2022) we did not scale the GFAS emissions. Thus, our approach assumes that the adjustment of the MOPITT-Inversion in CO is also applicable to $CO_2$. We acknowledge that this is an approximation, as the emission ratio between CO and $CO_2$ could also be off in GFAS. However, here we assume they are constant and interpret the underestimation in CO as an underestimation in fire emissions. This is in line with recent studies of undetected African fire emissions (Ramo et al., 2021).

Furthermore, a non-optimized set of fluxes is used to account for important $CO_2$ sources that contribute to the integrated signal of $CO_2$ in the atmosphere within our domain. Ocean fluxes are based on surface-ocean pCO2 data (Rödenbeck et al., 2013) but specifically processed at higher (1x1 deg) resolution (Run ID: oc_1x1_v2022). Following Steinbach et al. (2011), the EDGAR 4.3 inventory, sector and fuel-type specific and scaled at the national level for each year based on the British Petroleum statistical review (BP Annual report 2020), is used to account for emissions related to the burning of fossil fuels.

The assumed regional prior uncertainty for the domain-wide and annually integrated flux is chosen to be 0.9 PgC year$^{-1}$, which is based on the contribution of the regional domain to the assumed prior uncertainty (2.8 PgC) in the CarboScope global inversion system. Therefore, when spatially aggregating the spatially and temporally correlated prior error, regardless of the correlation length scale, it scales to the assumed prior uncertainty for the domain.

## 2.2 Statistical metrics

To report our results, we have adopted the following metrics. The uncertainty associated with the posterior covariance matrix in the inversion system, what is often referred to as "Bayesian" uncertainty, is used in the context of reporting the ensemble median for particular regions. The posterior flux uncertainty can be calculated from the prior uncertainties given by the prior flux and the measurement covariance matrices as described by Rödenbeck (2005). The posterior uncertainty is calculated for each year of interest (2010-2018) and the regions in Figure 2a as the square root of the covariance matrix multiplied by a regional operator. This uncertainty primarily depends on the observation availability and the assumed uncertainties for model-data-mismatch and prior fluxes, which is independent of the biosphere or diagnostic model used as a prior. The uncertainty reduction (UR, $\Delta\sigma$) is then calculated with equation 4,

$$\Delta\sigma = \frac{\sigma_{prior} - \sigma_{post}}{\sigma_{prior}} \tag{4}$$

For the complete prior and posterior ensemble in CarboScope regional an ensemble mean is calculated for each single year (Equation 5). Then the ensemble IAV is calculated as the standard deviation of the ensemble mean across all years (Equation 6). The equation 6 refer to $F_{NetLand}$, but one can replace it with a different flux component, like NBE, to obtain the same quantities.

$$F_{NetLand_{ens.mean_k}} = (\frac{1}{n}\sum_{i=1}^{n} EnsMember_i)_j, j = 2010 - 2018, k = Region \tag{5}$$

$$F_{NetLand_{IAV_k}} = std(F_{NetLand_{ens.mean_k}}) \tag{6}$$

## 3 Results

### 3.1 Understanding atmospheric and prior constraints

We found that the eastern part of the South American domain has a better observational constraint compared to the west. We obtained a mean (averaged over 2010-2018) Uncertainty Reduction (UR) of 44% for the Amazon region and for all the regions within it a reduction equal to or above 18% (Figure 3). Despite the uncertainty reduction of 44%, the absolute uncertainty of the posterior is 0.33 PgC year$^{-1}$. For the 'Amazon River Flat Plains' and the 'Brazilian Shield Moist Forests' the mean reduction is 53% and 54%, followed by the 'Guianan Shield Moist Forests' with 25% and the 'Amazon and Andes Piedmont' with the lowest mean reduction of 18% within the Amazon region. Note that the 'Cerrado & Caatinga' has a mean UR of 30%, higher than the 'Amazon and Andes Piedmont', indicating that on average there is better observational coverage over this eastern part of the domain.

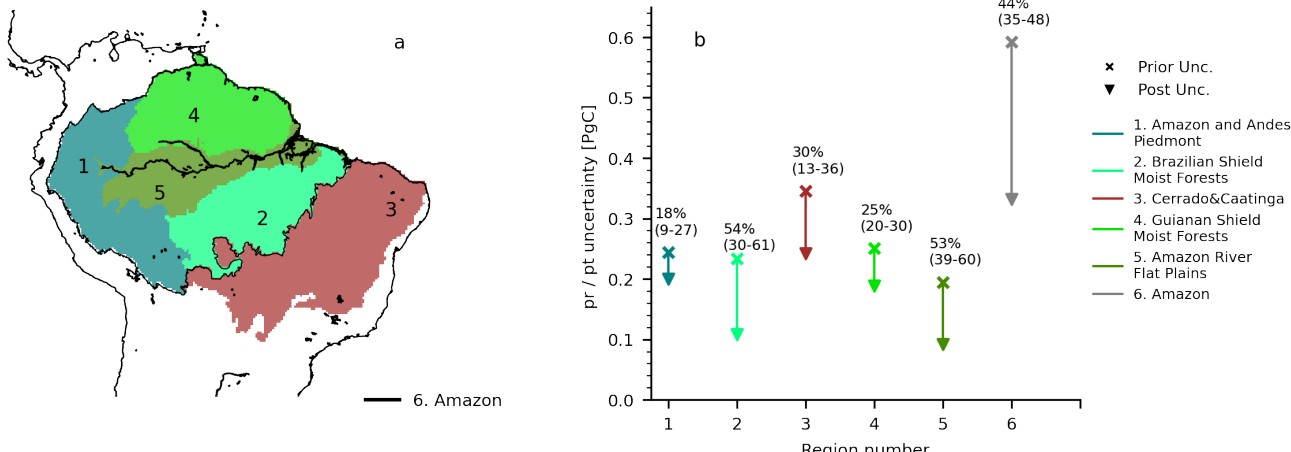

**Figure 3.** Areas for spatial integration of fluxes (a) and prior / posterior uncertainty for each of these areas (b). The percentages represent the mean uncertainty reduction over the period between 2010 and 2018, the values in brackets indicate the min-max range. For a complete time series for each region see Figure A12. Region 6 (biogeographic Amazon) is the sum of 1, 2, 4 and 5.

The UR not only varies spatially but also from year to year, depending on the continuity of the measurement records. As expected, years with data gaps have a low uncertainty reduction (see Figure A12). For example, due to the gaps in 2015 at the sites of TEF, SAN, and partly ALF, the uncertainty reduction was affected largely in the 'Brazilian Shield Moist Forests' (decreasing by 21% from 2014 to 2015), the 'Amazon River Flat Plains' (17%) and the 'Cerrado & Caatinga' (18%). This impact on UR was also observed in the biogeographic Amazon with an effect of 10% (Figure A12). Furthermore, in the 'Amazon and Andes Piedmont' we observe a slight decrease in the UR throughout the years. The highest UR is observed from 2010 to 2012, when the TAB site was active; after that, the UR never gets back to the 2010-2012 values. The latter highlights the effect of the location and the continuity of the measurement record on the UR for a particular region, and specifically the low information content in the west of the domain.

The largest UR caused by adding the ATTO data is for the flux estimate from 'Guianan Shield Moist Forest'. Sensitivity tests (not shown) excluding ATTO from the assimilated data show that on average the UR is 7% lower in this region, but some years reaching 12% (i.e. 2016). The constraint added by ATTO is smaller but also relevant in the 'Amazon River Flat Plains', increasing the UR by 6% also in 2016. At the biogeographic Amazon scale, the mean impact on the UR is small (2%), but in individual years it can amount to more than 5%. These changes in UR are somewhat conservative as in this study we have treated the aircraft data and the continuous data from ATTO in a similar way, inflating the uncertainty depending on the number of observations per week or per vertical profile, as described in Section 2.1.4. It is important to mention that the mean bias error of simulated mole fractions at each site assimilated, shows a better agreement between ensemble members (individual priors) relative to the simulations using prior fluxes, and the magnitude is reduced considerably from prior to posterior (see

Figure A13). For the MAN site, used as validation (i.e. not assimilated), we observed a reduction in the mean bias from -0.4 ppm to 0.1 ppm.

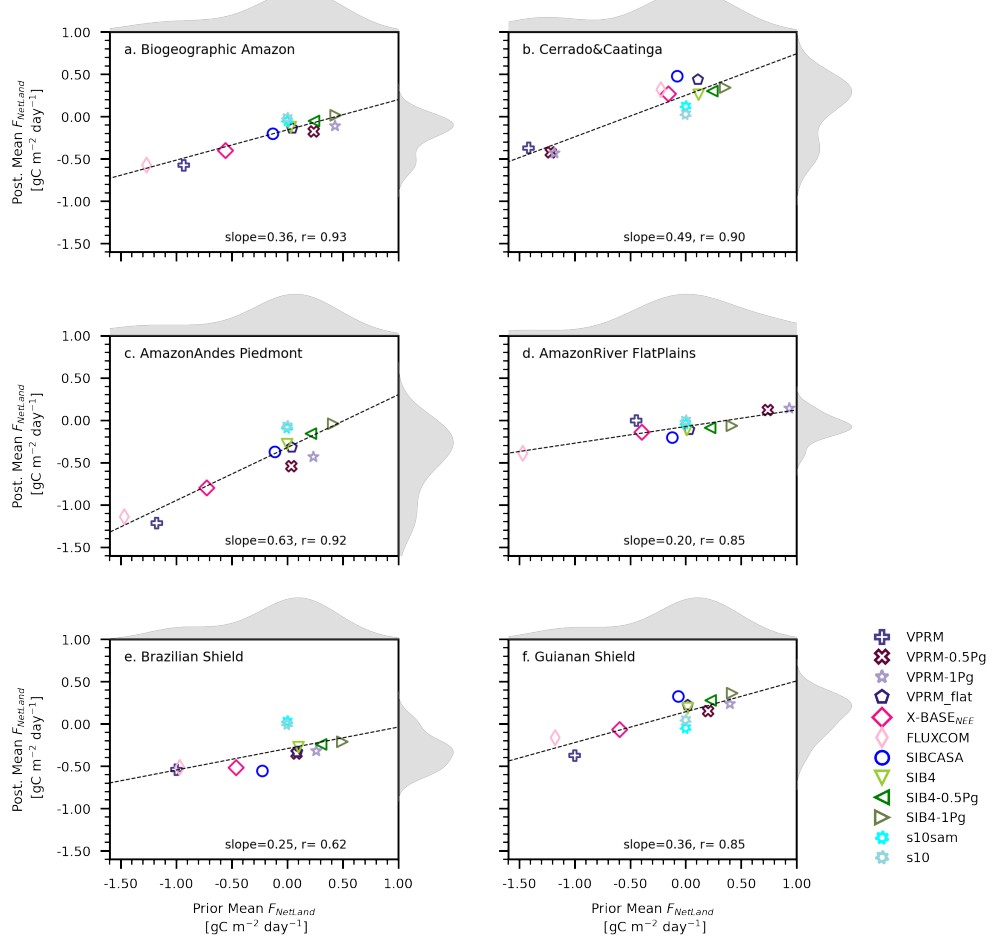

**Figure 4.** Relationship between prior and posterior mean net land flux ($F_{NetLand}$). On the y and x axes of each panel the density distribution is shown.

For the biogeographic Amazon and the 'Cerrado & Caatinga' we find a strong linear dependence of the posterior estimates on the prior (Figure 4a,b). Even though the spread in the marginal distribution is reduced largely from prior to posterior, the models with a large uptake in the prior (e.g. VPRM, FLUXCOM, X-BASE$_{NEE}$) do not converge with the main cluster of posterior estimates. We further evaluated such dependence with the VPRM$flat$ experiment, confirming that after removing the long-term mean, the VPRM$flat$ posterior $F_{NetLand}$ falls closer to the main group of estimates in both regions. Interestingly,

the regions in the eastern part of the Amazon ('Amazon River Flat Plains', 'Brazilian Shield Moist Forests', and the 'Guianan Shield Moist Forests') exhibit superior constraint by atmospheric data, as illustrated in Figure 4c-f. The spread in the posterior

marginal distribution and the slopes of the linear regression in these four regions are inversely proportional to their respective reduction in uncertainty (Figure 3). This inverse relationship indicates that the posterior estimates are more effectively adjusted, irrespective of the prior magnitude, in regions with a higher reduction in uncertainty. Therefore, the 'Amazon and Andes Piedmont' in the west stands out as an area where a bias in the prior fluxes would exert a more substantial impact on posterior estimates.

## 3.2 Carbon balance for tropical South America

The atmospheric inversion allocates a net carbon source ($F_{NetLand}$) to the 'Cerrado & Caatinga' region (except for VPRM, VPRM-0.5Pg and VPRM-1Pg) and consistently identifies a net carbon sink in the biogeographic Amazon (Figure 5 and see Figures A9 and A10 for the spatial patterns). Interestingly, the addition of South American stations amplify this pattern and reduces the posterior uncertainty (compare s10 with s10sam). Despite the considerable variability in magnitude for the biogeographic Amazon, the atmospheric constraint tends to adjust priors with a positive sign, shifting them towards a smaller source (e.g., SIB4-1Pg) or even turning them into a sink with a negative $F_{NetLand}$ (e.g., VPRM-1Pg, VPRM-0.5Pg, and SIB4-0.5Pg). Furthermore, in the global inversion when adding South American data, the resulting fluxes closely follow the sign and magnitude of those obtained in the regional inversion. Therefore, we contend that information suggesting a sink-source gradient between the Amazon and the 'Cerrado & Caatinga' is embedded in the atmospheric measurement record, even with the inherent limitations in adjusting certain individual priors.

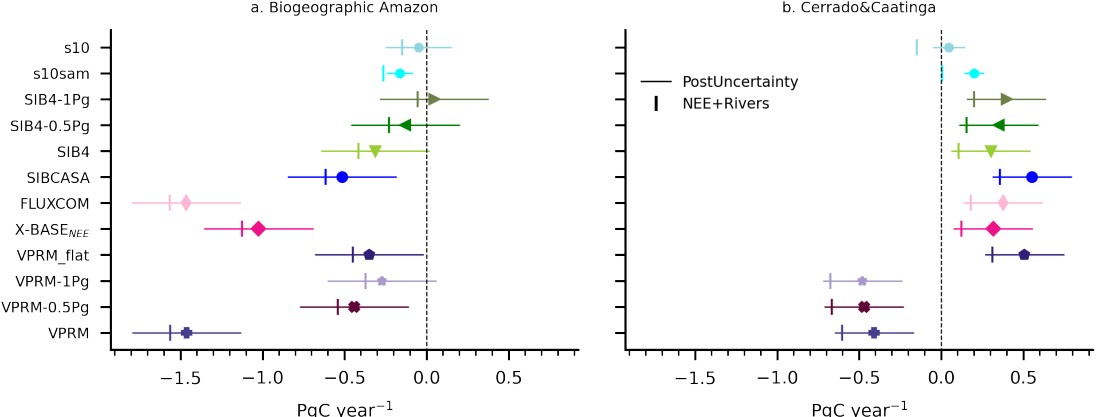

**Figure 5.** Carbon balance for the biogeographic Amazon (a) and the Cerrado & Caatinga (b) regions. The posterior flux components shown with a vertical bar (NEE + $F_{river}$), result from subtracting the $F_{fire}$ and $F_{ff}$ from the posterior $F_{NetLand}$ (shown with the markers).

The estimates obtained using VPRM, VPRM-1Pg, and VPRM-0.5Pg for the 'Cerrado & Caatinga' undergo substantial adjustments, with some cases exceeding +1 PgC year$^{-1}$ (VPRM). However, this diagnostic model (VPRM) exhibits a bias towards a large sink in the prior (<-1 PgC year$^{-1}$) that even after the inversion, the posterior estimate is far from the other

ensemble members and beyond the posterior uncertainty. Similarly, in the biogeographic Amazon a comparable pattern is observed for VPRM, FLUXCOM, and X-BASE$_{NEE}$. Nevertheless, the removal of the long-term mean of the prior flux, as implemented in VPRM$_{flat}$, results in a $F_{NetLand}$ that aligns more closely with the main cluster of posterior estimates at ≈-0.2 PgC year$^{-1}$ for the Biogeographic Amazon and at ≈0.3 PgC year$^{-1}$ for the 'Cerrado & Caatinga'. In other words, a prior flux having zero-mean, but a diurnal cycle would be closer to the main cluster of posterior estimates. To provide further insights on the main drivers of $F_{NetLand}$ and the spatial gradient reported here, we explore the individual components of the $F_{NetLand}$.

The mean fire emission estimates for the biogeographic Amazon and the 'Cerrado & Caatinga' are 0.10 PgC year$^{-1}$ and 0.19 PgC year$^{-1}$, respectively (differences between marker and vertical bar in Figure 5 and shown in Figure 6). Our findings indicate that, for most estimates, the carbon source in the 'Cerrado & Caatinga' does not solely originate from fires (Figure 5b). Within the cluster that exhibits positive posterior estimates (excluding s10), the remaining flux components (NEE + $F_{river}$) remain positive even after subtracting the fires and fossil fuels from the mean $F_{NetLand}$. Values range from neutral for s10sam with 0.00 PgC year$^{-1}$ to 0.35 PgC year$^{-1}$ for SIBCASA. Despite fires contributing 0.19 PgC year$^{-1}$ in this region, the persistence of positive NEE + $F_{river}$ in most inversions suggests a non-fire-related carbon source. Conversely, for the biogeographic Amazon we observe a consistently negative NEE + $F_{river}$ across all posterior estimates, with a smaller fire component (0.10 PgC year$^{-1}$) compared to the adjacent 'Cerrado & Caatinga.' Therefore, given these results the majority of fire-related carbon sources lie outside the biogeographic Amazon, emphasizing the critical role of fire locations in determining remaining flux components in the NBE.

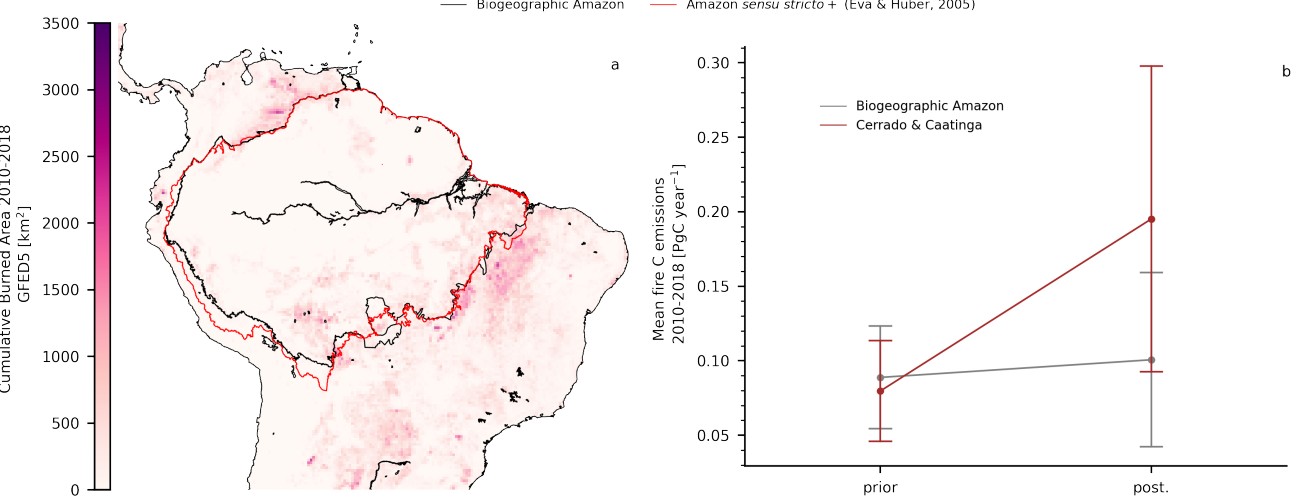

**Figure 6.** Cumulative burned area for the domain of the regional inversion using GFED5 (a) (Chen et al., 2023) and the prior and posterior fire emission estimates from GFAS and optimized with using satellite retrievals from MOPITT (Deeter et al., 2019) (b) for the biogeographic Amazon and the 'Cerrado & Caatinga' regions, the error bars denote the IAV.

The largest contribution (63%) to the mean fire emission in the biogeographic Amazon (0.10 PgC year$^{-1}$) is from the 'Brazilian Shield Moist Forests' (see Figure A14). This fire estimate is lower than the estimates reported in other studies (Gatti et al., 2021; Basso et al., 2023), but our analysis shows that fire emissions are concentrated on the border of the Amazon and the 'Cerrado & Caatinga' region. Note that when adding the fire emissions from the 'Brazilian Shield Moist Forests' and the 'Cerrado & Caatinga', we obtain 0.25 PgC year$^{-1}$, which is close to the fire estimate of Basso et al. (2023). Furthermore, the cumulative burned area in GFED5 (Chen et al., 2023), a fire emission proxy independent of the optimized fires used here, has most of the burning in the 'Cerrado & Caatinga' (Figure 6a), which is also the case when using an alternative Amazon boundary, such as that defined by Eva et al. (2005). Burned area is indicative of fire activity but it does not scale 1:1 with fire carbon emissions, as they depend on factors like fuel load and combustion efficiency (van der Werf et al., 2010; van Wees et al., 2022). However, fire emissions in the 'Cerrado & Caatinga' region increase from prior (0.08 PgC year$^{-1}$) to posterior estimates (0.19 PgC year$^{-1}$) (Figure 6b), consistent with an increase in burned area from GFED4 (Giglio et al., 2013) to GFED5 Chen et al. (2023). It is worth noting that the fire emissions used here results from a top-down optimized version (Naus et al., 2022) based on CO using satellite retrievals from MOPITT (Deeter et al., 2019). Therefore, the difference in fire emissions relative to other studies is likely associated with the distinct methodologies used to attribute the fluxes as well as the spatial resolution used in the inversion, both of which can lead to accounting for fire emissions outside of the regional boundaries in the Amazon region.

### 3.3 Effect of systematic measurement uncertainty on posterior fluxes

The flask-specific $CO_2$ bias-correction (see Section 2.1.5) results in a consistent shift towards a source on the posterior $F_{NetLand}$ for the biogeographic Amazon (Figure 7a). No significant effect is observed on interannual variability or spatial gradients (Figure A15). Applying the correction results in a mean posterior $F_{NetLand}$ with a weaker sink or a larger source of carbon (Figure 7a) for particular years, consistent with the need to simulate higher $CO_2$ levels to match the data. Using the global inversion, we find a mean effect of 0.21 PgC year$^{-1}$, while in the regional inversion the effect is 0.31 PgC year$^{-1}$ (see Table A1). For the 'Cerrado & Caatinga' the mean effect is 0.10 PgC year$^{-1}$ (CarboScope Global) and 0.14 PgC year$^{-1}$ (CarboScope Regional), and for the regions within the Amazon equal or less than 0.10 PgC year$^{-1}$ for both global and regional inversions (see Table A1). Note that the correction is allocated to the regions that were selected previously having an uncertainty reduction larger than 15% consistent with the observational coverage (compare Figure 7b with Figure 2). In other words, the correction only affects the areas covered by the aircraft network.

Assuming that the correction brings the observational data closer to the truth implies that the Amazon is a weaker sink of carbon and that the 'Cerrado & Caatinga' is a larger source. We do not find a strong spatial shift of fluxes within the biogeographic Amazon due to the correction, nor do we find a strong impact on the interannual variations. The response to drought in 2010, 2015 and 2016 is affected in the absolute flux magnitude, but the year-to-year variability remains the same (Figure 7). Therefore, the findings of our sensitivity tests support the hypothesis in Gatti et al. (2023) that the water vapor bias mainly affects the absolute annual flux magnitudes. In both cases, with and without correction, our estimates for the total carbon loss to the atmosphere in the Amazon during 2015 and 2016, are lower (from 0.15 to 0.3 PgC) than other studies (Liu

et al., 2017; Gloor et al., 2018). Our total net flux is closer to the $0.5 \pm 0.3$ PgC from Gloor et al. (2018), but note that they used a time period from September 2015 to June 2016 and the area they refer to as Amazonia is not clearly defined. Compared to the $1.6 \pm 0.29$ PgC in Liu et al. (2017), our estimates are much lower, but the difference in area is large as they refer to tropical
South America, including parts of the 'Cerrado & Caatinga' biomes and central America. Furthermore, they focus their study on the anomaly relative to 2011 at the peak of the El Niño phase.

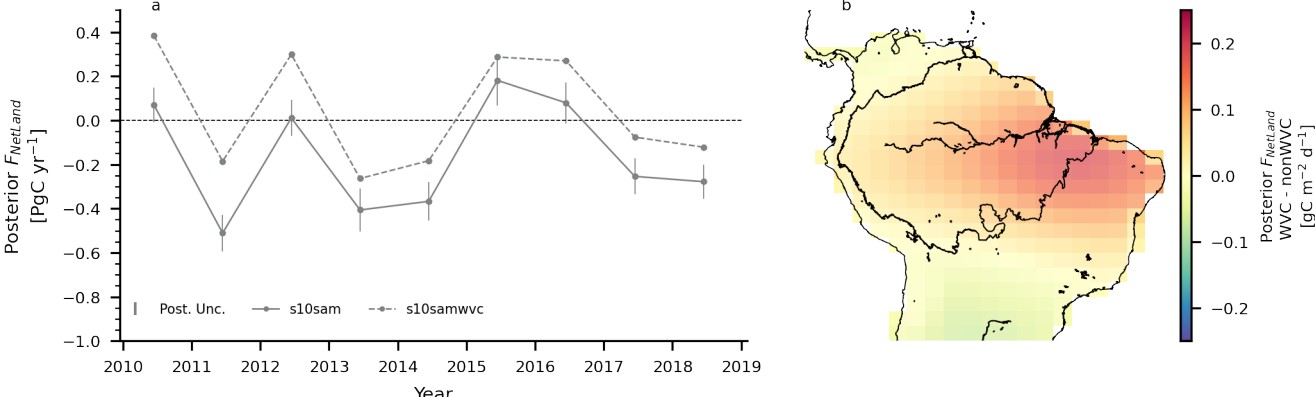

**Figure 7.** Posterior flux estimates using the global inversion assimilating vertical profile data without a bias correction (s10sam) and with the correction (s10samwvc) (a). The integrated flux corresponds to the Biogeographic Amazon. The error bars correspond to the posterior uncertainty for the Biogeographic Amazon. In (b), the difference per grid cell of the two resulting posterior estimates averaged over 2010-2018.

## 4  Discussion

### 4.1  The Amazon carbon exchange in context

Significant progress advancing the monitoring capacity over the continent, particularly in Brazil, has allowed for an increasing
number of studies (Gatti et al., 2014; van der Laan-Luijkx et al., 2015; Alden et al., 2016; Koren, 2020; Gatti et al., 2021; Botía et al., 2022; Basso et al., 2023; Gatti et al., 2023) using atmospheric mole fraction measurements to understand ecosystem carbon exchange. Nevertheless, using atmospheric inversions to estimate net carbon exchange remains a challenging task given the remaining limitations in data constraint, as shown in Figure 3. We have shown that posterior estimates have a linear dependence on the prior magnitude that varies according to the uncertainty reduction and absolute posterior uncertainty. On
the biogeographic Amazon scale, such linear dependence together with the large posterior uncertainty ($\pm 0.33$ PgC year$^{-1}$) makes it difficult to attribute a mean carbon exchange based solely on one prior, as the magnitude and sign could be greatly influenced by that of the prior. To address this issue, our study offers a meticulous evaluation of uncertainty for specific regions, recommending areas where the inversion benefits from superior atmospheric data constraint. In light of this assessment, we

identify potential sites for new measurement stations to reduced the uncertainty in Amazon-wide top-down estimates, like the
'Amazon-Andes foothills'.

In this study, using an ensemble of prior fluxes in a regional inversion and a global inversion with and without South American data, we can infer that the biogeographic Amazon is a net carbon sink (i.e. negative $F_{NetLand}$) given by the direction of the adjustment introduced by the inversion. When subtracting fires and fossil fuels we obtain a negative 'NEE + River' flux component in all of the individual inversion runs, suggesting a consistent ecosystem carbon sink over 2010-2018. Taking
into account all posterior estimates, the median $F_{NetLand}$ is -0.33 PgC year$^{-1}$. The posterior median without considering the inversion runs using priors with a large sink (FLUXCOM, X-BASE$_{NEE}$, and VPRM) is -0.24 PgC year$^{-1}$, both with a posterior uncertainty of $\pm$ 0.33 PgC year$^{-1}$. We avoid reporting an ensemble mean as the large variability in the magnitude given by the setup of our experiments could be biased, and thus we rather interpret the individual posterior estimates in relative terms (as done in Section 3.1).

We acknowledge that a shortcoming of our regional inversion, common to other regional systems (Munassar et al., 2023), is the use of a single global inversion for the far-field contribution (e.g. the $CO_2$ mole fraction advected into the regional domain). For example, if the global inversion has a regional bias in the $CO_2$ mole fractions, it can propagate to the regional inversion via the far-field contribution. Yet, this influence should be smaller in well-constrained areas (those having high uncertainty reduction, Figure 3). In this regional domain, the predominant atmospheric flow changes seasonally from easterly
to northeasterly, both directions constrained by two background stations: Ragged Point Barbados (RPB, 13.1650°N, 59.432°W ) and Ascension Island (ASC, 7.9667°S, 14.4° W). CarboScope global assimilates data from these stations and the mean bias error between posterior $CO_2$ mole fractions and local measurements at these sites is lower than 0.1 ppm (Botía et al. (2022), Supplementary Figures). In addition, Schuh et al. (2019) showed that atmospheric transport models (e.g. TM5) having a slow vertical and meridional transport in northern mid-latitudes can have a weaker sink from 45 N to 90 N, but a larger one from 0 to
45 N and thus result in a stronger global integrated carbon sink compared to the fast vertical mixing models (e.g. GEOS-Chem). We speculate that as a precursor of TM5, the model we used in the global inversion (i.e. TM3) could have a stronger global carbon sink due to a slower vertical and meridional transport, so the contribution of our South American domain to that carbon sink will propagate to the regional inversion via Step 1 of the two-step scheme. The quantification of this potential effect is a source of uncertainty in our regional inversions that should be further quantified. Finally, we have assumed weekly error
correlations for the model-data mismatch uncertainty to be able to assimilate multiple data streams and for consistency with the CarboScope Global inversion system. Such assumption should be revisited and evaluated with further sensitivity tests in future studies in the tropics, but maintaining consistency between the Global (which provides the boundary conditions) and Regional inversion systems.

Our estimates for the biogeographic Amazon are consistent with bottom-up approaches in the negative sign of net land
flux ($F_{NetLand}$) (Table 2), yet a large uncertainty remains in both approaches. For the same area, Basso et al. (2023) reports a small net source of carbon, but considering the uncertainty range, all approaches overlap. Integrating our posterior fluxes for the same Amazon boundaries as in Gatti et al. (2021) and Basso et al. (2023) results in contrasting signs, but again with an overlapping uncertainty range. Therefore, to answer the sink/source question for the Amazon region -regardless of its

boundaries-, reducing uncertainty in both bottom-up and top-down approaches should be the priority in upcoming studies.
From the top-down perspective, such uncertainty reduction can be achieved by expanding the observational network to the west of the Amazon, towards the Andes foothills in Peru, Colombia, and Ecuador, as this region had the smallest uncertainty reduction in our study. Thus, mean estimates over these regions should be interpreted with caution, as the selection of prior fluxes can largely influence the sign and the magnitude of such a mean, as shown in this study.

To further reduce the uncertainty in this domain, top-down estimates could combine in-situ data with satellite retrievals. The inversions assimilating data from the Orbiting Carbon Observatory 2 (OCO2) (Liu et al., 2017; Crowell et al., 2019; Peiro et al., 2022; Wang et al., 2023) have shown that remotely-sensed $CO_2$ columns can provide a valuable constraint of net carbon exchange in tropical regions. However, the OCO2-inversions are still limited by cloud coverage during the wet season (Massie et al., 2017; Peiro et al., 2022) and the adjustment of the prior can be biased to dry season retrievals (Crowell et al., 2019). Nevertheless, our results for the response to the El Niño 2015/2016 coincide with the OCO2 inversions (Liu et al., 2017; Crowell et al., 2019; Peiro et al., 2022) in a carbon source in 2015 and 2016. Yet, a direct comparison of the magnitude in those studies to our results is difficult, as the area for South America in Liu et al. (2017) includes parts of the 'Cerrado & Caatinga' regions and in Crowell et al. (2019); Peiro et al. (2022) they divide South America in three parts: 1. Northern South America: including only the north (north of the equator) of the Amazon basin, the Orinoco basin, a part of central America and the Caribbean islands. 2. Southern Tropical South America, which includes a large part of the Cerrado and the southern part of the Amazon and 3. South America Temperate, which includes the Cerrado and Caatinga, extending until the southernmost point on the continent. None of these regions coincide with our regional distribution, therefore, using our domain definition on the OCO2-MIP results should be part of a next study, as it is out of the scope of this one.

The effect of fires is fundamental to understanding the NBE and quantifying this signal remains challenging. Studies like van der Laan-Luijkx et al. (2015); Koren (2020); Naus (2021); Basso et al. (2023), show that further optimization of fires in a Bayesian setup can have an important impact on the magnitude of the derived fire-CO signal, which can be translated to a fire-$CO_2$ signal and thus impact the net carbon flux on the regional scale. This is why we selected an optimized fire emission for our domain, as did Basso et al. (2023), which results in fire emissions that are generally larger than the prior used, in both cases GFAS. The higher estimates than the original GFAS product are in line with recent findings that the global burned area has been strongly underestimated, resulting in underestimated fire emissions (e.g. Ramo et al. (2021)). Further advances in global burned area mapping (Chen et al., 2023) and fire emission estimates (van Wees et al., 2022; Wiedinmyer et al., 2023) will help to reduce the estimated fire flux and help to further constrain NBE in the future. The fire flux for the Amazon area used in Gatti et al. (2021) is $0.41 \pm 0.05$, which is relatively high compared to the 0.09 PgC year$^{-1}$ (Rosan et al., 2024), the 0.26 PgC year$^{-1}$ (Basso et al., 2023), and the 0.10 PgC year$^{-1}$ from this study. The Gatti et al. (2021) approach with the column budget technique solves for the total flux concerning a background signal linked to each of the aircraft profiles. Using an observation-based CO:$CO_2$ relationship, they obtained the contribution of fires and subtract that from the $F_{NetLand}$, but the area for attributing fluxes corresponds to the regional influence of each aircraft site limited to the Amazon boundaries. As the CO:$CO_2$ is also influenced by areas outside of the Amazon (see Figures 5 and 6 in Cassol et al. (2020) and Figure 1 in Gatti et al. (2014) ), such an approach could assign fire sources from the Cerrado to the Amazon.

In addition to the challenges posed by a limited observational constraint in this region, other sources of uncertainty could be associated with water vapor in the samples of the aircraft vertical profiles, as was mentioned in Gatti et al. (2023). Previous studies (Baier et al., 2020) focusing on the effect of water vapor in flasks, have shown that the linear dependency of the $\Delta CO_2$ on humidity in the flasks is more evident above $\approx 1.5\%$ $xH_2O$. We have fitted the $\Delta CO_2$ to the entire humidity range in the measurements and thus the range simulated by the STILT-ECMWF-IFS model (see Figure A7), so we recognize that the bias estimates at low water vapor levels could be slightly overestimated. Nevertheless, we hypothesized that the effect of that overestimation on the posterior fluxes should be minor, as water vapor mole fractions are predominantly above 1% at all sites. Thus, our approach provides a reference for future studies focusing on characterizing the water dependence in undried PFP samples and the effect of such a bias in an atmospheric inversion. We have quantified the effect of this systematic uncertainty on the posterior estimates, resulting in a weaker sink or a larger carbon source of 0.31 (based on the regional inversion) to 0.21 PgC year$^{-1}$ (based on the global inversion). As we have shown, assimilating a bias-corrected dataset does not affect the IAV nor the spatial patterns of the retrieved fluxes and rather results in an upward shift (towards a source of carbon) of the posterior flux.

The implications of our suggested correction depend on the prior used, but factoring in the correction (see Table 2) in the median $F_{NetLand}$ of -0.33 PgC year$^{-1}$ $\pm$ 0.33 PgC year$^{-1}$, we obtain a close-to-neutral carbon balance (-0.02 PgC year$^{-1}$ $\pm$ 0.33 PgC year$^{-1}$ if the correction from the regional inversion is used) or a small sink (-0.13 PgC year$^{-1}$ $\pm$ 0.33 PgC year$^{-1}$, if the correction from the global inversion is used). Note that in both cases after subtracting fires and fossil fuels the resulting flux will still represent an ecosystem carbon sink (a negative NEE + $F_{rivers}$). In addition, considering the posterior uncertainty of $\pm$ 0.33 PgC year$^{-1}$ together with the potential effect of water vapor in the flasks (0.31 PgC year$^{-1}$, using the regional adjustment), the positive shift amounts to +0.64 PgC year$^{-1}$ suggesting a large change in the carbon balance at the biogeographic Amazon scale. In any case, this change is within the large spread in $F_{NetLand}$ estimates by different approaches (Table 2). Now, placing this in the context of the global carbon budget, an increase of +0.64 PgC year$^{-1}$ is within the uncertainty of the global net land carbon fluxes (1.1 PgC year$^{-1}$) for 2013-2022 reported in Friedlingstein et al. (2023), and would represent a third of the constraint given by atmospheric inversions with a range of 0.5-2.3 PgC year$^{-1}$ (Friedlingstein et al., 2023). Therefore, given the large uncertainties reported for the global carbon budget, it is possible to accommodate such a magnitude shift in the net land carbon flux of the biogeographic Amazon.

### 4.1.1 Rivers

For the carbon balance of the main branch of the Amazon River, we estimated a $F_{NetLand}$ close to neutral. The median $F_{NetLand}$ for the 'Amazon River Flat Plains' region was -0.04 $\pm$ 0.09 PgC year$^{-1}$. Note that this region is relatively well constrained by the atmospheric monitoring network with an mean UR of 53%, thus the posterior estimates have the smallest spread of all regions (see Figure 4d). After removing fires and fossil fuel emissions (0.01 and 0.007 PgC year$^{-1}$) the resulting flux, NEE + $F_{river}$, is slightly negative. In this work we have considered river $CO_2$ evasion ($F_{river}$) as an explicit component of the NBE because we believe is a land flux that should be characterized independently of the NEE, in particular for the Amazon lowland area considering that at least 31% (Fleischmann et al., 2022) can be seasonally flooded.

**Table 2.** Comparison of $F_{NetLand}$ for the Amazon region based on top-down and bottom-up approaches averaged over 2010-2018. We present estimates for two different definitions of Amazon boundaries, the Biogeographic Amazon and the Amazon *sensu-stricto+*. The latter corresponds to four subregions defined in Eva et al. (2005): Amazon *sensu stricto*, Andes, Guiana and Gurupi. We present our estimates for the Amazon *sensu-stricto+* so the area is comparable to previous top-downs studies. The acronym WVC, stands for water vapor correction. Units in PgC year$^{-1}$.

| | Area | Fires | $F_{NetLand}$ | Uncertainty |
|---|---|---|---|---|
| This study (all ensemble members) | Biogeographic Amazon | 0.10 | -0.33 | $\pm$ 0.33 |
| This study (only process-based models) | Biogeographic Amazon | 0.10 | -0.24 | $\pm$ 0.33 |
| Rosan et al. (2024)-Bottom-up | Biogeographic Amazon | 0.09 | -0.15 | $\pm$ 0.19 |
| Rosan et al. (2024)-Hybrid | Biogeographic Amazon | - | -0.25 | $\pm$ 0.19 |
| Rosan et al. (2024)-CARDAMOM | Biogeographic Amazon | - | -0.34 | CI=[-2.94,2.45] |
| Basso et al. (2023) in Rosan et al. (2024) | Biogeographic Amazon | - | 0.02 | $\pm$ 0.13 |
| This study | Amazon *Sensu stricto+* | 0.11 | -0.18 | $\pm$ 0.34 |
| Basso et al. (2023) | Amazon *Sensu stricto+* | 0.26 | 0.13 | $\pm$ 0.17 |
| Gatti et al. (2021) | Amazon *Sensu stricto+* | 0.41 | 0.29 | $\pm$ 0.40 |
| This study (all ensemble members) + WVC-global | Biogeographic Amazon | 0.10 | -0.33+0.21 = -0.12 | $\pm$ 0.33 |
| This study (all ensemble members) + WVC-regional | Biogeographic Amazon | 0.10 | -0.33+0.31 = -0.02 | $\pm$ 0.33 |

Using output from a process-based model (ORCHILEAK, Hastie et al. (2019)), Botía et al. (2022) suggested that river outgassing could play an important role in representing the seasonal pattern of $CO_2$ mole fractions at ATTO, yet there are important processes in that model that are not accounted for (i.e. aquatic plants), which is why in this study we did not used ORCHILEAK as a prior. However, if we consider $F_{river}$ as the $CO_2$ evasion based on the ORCHILEAK model (Hastie et al., 2019) for the 'Amazon River Flat Plains' (0.09 PgC year$^{-1}$), the resulting NEE for this area would be even more negative, suggesting that plant productivity in this region is larger than the respired $CO_2$ from the decomposition of organic matter. Nevertheless, the growth of aquatic plants in rivers could play an important role in the net balance of riverine $CO_2$ fluxes (Science Panel for the Amazon, 2021). This uptake of carbon, which is not taken into account in ORCHILEAK, could potentially balance out the $CO_2$ outgassing due to decomposition of submerged organic carbon and respiration of roots, which are the main sources in ORCHILEAK for the $CO_2$ evasion. Our results for the 'Amazon River Flat Plains' suggest the region is close-to-neutral, but to partition the components NEE + $F_{river}$ using an atmospheric inversion, bottom-up estimates for rivers should consider aquatic plant productivity, which is a crucial process to determine net river evasion or uptake.

## 4.2 Sink-to-source gradient between the biogeographic Amazon and the Cerrado & Caatinga

Our results suggest that the Amazon is a net carbon sink and that the 'Cerrado & Caatinga' biomes are possibly net sources of carbon, even after removing fires in some individual inversions we obtain a positive NBE. This is in contrast to Gatti et al.

(2021) and Basso et al. (2023). Gatti et al. (2021) suggested that vegetation of the southeast of the Amazon was losing the capacity to capture carbon, reporting on average a positive NBE for the years 2010 to 2018, thus locating a net source of carbon within the Amazon in the southeast regions. Basso et al. (2023) report a posterior carbon source in the eastern part of their domain (see Figure A11 in Basso et al. (2023)). In both studies, the analyses were limited to the Amazon *Sensu stricto* + (as defined in Table 2) but used the same aircraft data as we used in this study. Therefore, we believe that the main reasons explaining the differences reported here are: 1. limiting their region of interest to the Amazon boundaries, while using atmospheric data that is influenced by a footprint that goes beyond the Amazon and 2. having a coarse global inversion in which the boundaries between the Amazon biome and the Cerrado are not well-defined leading to attributing sources from the Cerrado to the Amazon.

The spatial attribution of fluxes in Gatti et al. (2021) is performed using a column budget technique. In this method, the residence time of air parcels over each region of influence for each aircraft site is used to account for the time and the area that contribute to an enhancement or depletion of $CO_2$ mole fractions at the site relative to the background. In our study, a similar source is needed to match the atmospheric profile data, but it is further east outside the Amazon bounds. This solution is possible in our system with individual (though correlated) grid points that can be assigned extra flux, but it is not possible in the setup of Gatti et al. (2021), where a mean flux rate is assigned to the areas of influence of the airborne measurements, which is limited to the Amazon boundary. Thus, the degrees of freedom to place sources/sinks in specific biomes further upwind are not present. Sensitivity tests in Botía (2022) indicate that our result is robust against changing the spatial error structure in the inversion settings. While recognizing such spatial differences between the studies, we nevertheless conclude that the east-to-west gradient within Amazonia that Gatti et al. (2021) report is not seen in our posterior ensemble median estimates. Interestingly, an individual ensemble member (i.e. s10sam) does result in a source (0.03 gC m$^{-2}$ d$^{-1}$) in the 'Brazilian Shield Moist Forest', but still having a larger one (0.12 gC m$^{-2}$ d$^{-1}$) in the 'Cerrado & Caatinga'. Therefore, it is likely that the atmospheric signal of a carbon source is unmistakably in the data, but is attributed to different spatial regions by the different methodologies. Understanding this discrepancy, and determining the location of the eastern Brazilian $CO_2$ source should have the highest priority in further work. Next, despite of these discrepancies, we speculate on what could be driving a source of carbon in the 'Cerrado & Caatinga' region.

The 'Cerrado & Caatinga' biomes cover approximately 35% of the Brazilian land mass (Beuchle et al., 2015) and are characterized by a Savanna-type ecosystem (Cerrado) (Sano et al., 2007) and seasonally dry tropical forest (Caatinga) (Prado, 2003). Both biomes have a marked seasonality in precipitation, with mean annual precipitation of less than 750 mm year$^{-1}$ in the Caatinga (Prado, 2003; Leal et al., 2005), and from 800 to 2000 mm year$^{-1}$ in the Cerrado (Ratter et al., 1997; Rodrigues et al., 2022). Studies focusing on ecosystem functioning conducted in both biomes using eddy covariance measurements have shown that these ecosystems have similar seasonal patterns in net ecosystem exchange. In general, with the onset of the rainy season a higher carbon uptake is observed over converted pastures and over natural vegetation (Miranda et al., 1997; Varella et al., 2004; Santos et al., 2004; Silva et al., 2017; Mendes et al., 2020; Alves et al., 2021). Studies focusing on soil $CO_2$ emissions comparing converted pasture and both natural ecosystems, Caatinga (Ribeiro et al., 2016) and Cerrado (Varella et al., 2004), found no significant differences in magnitude between the pasture and the natural ecosystem. Both coincided with

higher $CO_2$ emissions at the beginning of the rainy season (Varella et al., 2004; Ribeiro et al., 2016). The findings of Ribeiro et al. (2016) are in line with Mendes et al. (2020), where they observed an increase in ecosystem respiration with the onset of the rainy season, but offset by GPP. Integrated over time, these ecosystems in their natural form or converted to pasture seemed to be carbon sinks when no disturbance is taken into account (Santos et al., 2004; Bustamante et al., 2012; Silva et al., 2017; Mendes et al., 2020; Alves et al., 2021).

However, these biomes have suffered considerable loss of natural vegetation due to the expansion of the agricultural frontier (Beuchle et al., 2015; Alencar et al., 2019). Beuchle et al. (2015), found that over the period between 1990 and 2010, both biomes had a continued net loss of natural vegetation. More recently, most of the agricultural expansion in the Cerrado has been concentrated in a region called MATOPIBA, which refers to portions of the Maranhão, Tocantins, Piauí and Baia states (Spera et al., 2016; da Conceição Bispo et al., 2024). Drought periods in the Caatinga can extend over years (Leal et al., 2005), making agricultural activities more difficult to sustain, yet considerable pasture conversion for extensive livestock has changed the Caatinga landscape (Leal et al., 2005). Fire is used for the conversion of forest and shrublands to pasture or croplands, thus that conversion leads to fire $CO_2$ emissions (van der Werf et al., 2010; Pivello, 2011; van der Werf et al., 2017). The Cerrado biome was found to have higher fire $CO_2$ emissions than the Caatinga, with an increasing trend in recent years (da Silva Junior et al., 2020). Moreover, the annual fire regime fluctuates between naturally occurring low-intensity fires at the end of the wet season ignited by lighting, and anthropogenic high-intensity fires at the end of the dry season (Ramos-Neto and Pivello, 2000; Pletsch et al., 2022). Frequent fires can lead to aboveground biomass reduction, changing the ecosystem from a sink to a source of carbon (de Azevedo et al., 2020). Moreover, Bustamante et al. (2012) found that a large portion of the $CO_2$ emissions from pasture management (i.e. burning practices) in Brazil originated in the Cerrado. Throughout 2003 to 2013, changes in vegetation stocks due to cropland conversion in the Cerrado and specifically in the MATOPIBA region, contributed to 33% of the forest carbon emissions (Noojipady et al., 2017).

Given these studies and our findings, carbon emissions in these two biomes but primarily in the Cerrado, can be grouped into two categories. The first one is associated directly with fires (either from deforestation or pasture management), and the second arises from degradation after the conversion of natural vegetation to pastures or croplands. For the first category, we showed that fire emissions increased from prior to posterior in the 'Cerrado & Caatinga' region by a factor of 2.3, which is broadly consistent with an increase in burned area from GFED4 to GFED5, by a factor of 1.7. Emissions in the second category are associated with changes in carbon stocks, decomposition, and sink-to-source shifts due to climate change (Bustamante et al., 2012; Marengo et al., 2022). Having this in mind, we hypothesize that the carbon source in the semi-arid ecosystems of the 'Cerrado & Caatinga' given by the atmospheric inversion is likely due to agricultural expansion in the Cerrado, mainly from the second category mentioned above. On top, their capacity for secondary forest regrowth is compromised. This contrasts with the Amazon biome, which is highly resilient (Sakschewski et al., 2016; Poorter et al., 2021), and has shown a relatively rapid recovery of aboveground biomass by secondary forest growth (Poorter et al., 2016) with considerable potential to capture carbon (Heinrich et al., 2021, 2023). It is important to note here that Amazon forest resilience is heavily affected by anthropogenic disturbances (Fawcett et al., 2023; Wang et al., 2024) and it seems to be decreasing since 2000 (Boulton et al., 2022), although this finding has been questioned (Tao et al., 2023). This would make the carbon source we find in this study not only

large but likely also influence the regional carbon balance for decades to come. Therefore, the hypothesis of the 'Cerrado & Caatinga' source, driven by changes in carbon stocks and fires, should be part of a future study in which additional ways of testing the robustness of the inversion results are explored. For example, comparing several inversion systems using same data constraint could shed light on the spatial gradients reported here. Finally, an assessment of having different or even multiple global inversions for constraining the far-field contribution in the regional inversion should also be quantified and studied.

## 5   Conclusions

In this study, we have integrated in-situ $CO_2$ measurements from the Amazon Tall Tower Observatory together with a network of airborne vertical profiles in the CarboScope regional inversion system to estimate carbon fluxes in tropical South America. Our analysis is limited to regions with uncertainty reduction above 15%, which amounts to 67% of the land mass in our domain. Among these regions, the 'Amazon River Flat Plains', the 'Brazilian Shield Moist Forests' and to a lesser extent the 'Guianan Shield Moist Forests' are better constrained than others.

Furthermore, our results suggest a sink-source gradient between the Amazon (sink) and the 'Cerrado & Caatinga' (source). Across all ensemble members, this net absorption amounted to a median of -0.33 PgC year$^{-1}$. When relying solely on process-based models as priors, this value slightly decreased to -0.24 PgC year$^{-1}$, both estimates with a posterior uncertainty of $\pm$ 0.33 PgC year$^{-1}$. The Cerrado & Caatinga biomes together, acted as a median carbon source of $0.31 \pm 0.24$ PgC year$^{-1}$. This finding, which is in contrast to other top-down studies (Gatti et al., 2021; Basso et al., 2023), is explained by two reasons. The first one is that we do not limit our analysis to the Amazon boundaries, instead, we take into account the extended surface influence of the vertical profiles, which goes beyond the Amazon. The second reason is associated with a higher spatial resolution in our regional inversion system (0.25x0.25 degree) than in Basso et al. (2023). The latter allows us to integrate fluxes over the analysis regions with improved precision. This is particularly relevant considering that fires occur on -but are not limited to- the border of the Amazon and the Cerrado, so when having coarser grids one can attribute Cerrado fires to the Amazon.

We have quantified and reported important uncertainties associated with the data and methodology used in this work. Part of this was the assessment of how systematic uncertainties due to water vapor in the aircraft vertical profiles affect the estimated fluxes in our inversion system. In principle, we recommend to dry the air during sampling time to avoid systematic uncertainties and their propagation to flux estimates in an inversion system. Our analysis suggests that the proposed correction for water-vapor leads to an upward adjustment ranging from 0.21 to 0.31 PgC year$^{-1}$ in the biogeographic Amazon carbon flux. Note that other than the shift in magnitude, the interannual variability or the spatial gradients of the posterior fluxes do not change. Considering this correction and the remaining uncertainty (0.33 PgC year$^{-1}$) for the biogeographic Amazon, we conclude that such a change in the carbon budget is within the uncertainty of the global net land carbon flux, but represent further challenges in constraining the recent Amazon carbon balance.

*Code and data availability.* The data that support the findings of this study are, the ATTO $CO_2$ measurements, openly available in: https://attodata.org/ and the aircraft vertical profiles, which are openly available in PANGAEA: https://doi.pangaea.de/10.1594/PANGAEA.926834. The data generated in this paper, the simulated water vapor at each aircraft site with its corresponding bias correction (in ppm) will be made public at the time of publication. For the review process, the reviewers can access the data under this private link https://edmond.mpg.de/privateurl.xhtml?token=36e3c67d-d564-4c15-beaf-c0d46193a5b6. The analysis regions in postprocessing can be found here: https://edmond.mpg.de/privateurl.xhtml?token=a4af161f-1b77-4611-aa01-7de9203639b8. At the time of publication, both links will be made public. The posterior fluxes for each individual inversion run and the fire-$CO_2$ emissions based on the MOPITT-CO inversions can be made available upon request to Santiago Botía (sbotia@bgc-jena.mpg.de). The GFED5 burned area was retrieved from the open repository Zenodo (https://doi.org/10.5281/zenodo.7668423) (Chen et al., 2023).

*Author contributions.* SB wrote the initial manuscript and run the atmospheric inversions. SB together with CG, WP and CR designed the methodology. CR developed the global and the regional inversion systems and he assisted in running the global inversions. SM and TK assisted with the regional inversion runs, DC ran the forward runs simulations for the validation site MAN. LSB helped with the data curation of the aircraft sites and further interpretation of the results. SK, JVL and DW provided the ATTO data and run the ATTO measurement system until 2021. GM is co-PI of the aircraft data and assisted with the interpretation of the results. SN, GK, IL and WP provided the fire-CO2 emission based on the MOPITT-CO inversions. SH contributed with the analysis of burned area. JBM and GM ran the Manaus aircraft measurements and provided the data for the water vapor correction. All authors contributed to the analysis and text editing.

*Competing interests.* Some authors are members of the editorial board of the Atmospheric Chemistry and Physics (ACP) journal.

*Acknowledgements.* This work and the ATTO project was funded by the German Federal Ministry of Education and Research (BMBF, contracts 01LB1001A and 01LK1602A) and supported by the International Max Planck Research School for Global Biogeochemical Cycles (IMPRS-gBGC). The ATTO project is furthermore funded by the Brazilian Ministĺrio da Ciĺncia, Tecnologia e InovaĝĂčo (MC-TI/FINEP contract 01.11.01248.00) and the Max Planck Society. We acknowledge the Instituto Nacional de Pesquisas da Amazonia as well as the Amazon State University (UEA), FAPEAM, LBA/INPA and SDS/CEUC/RDS-Uatumĉ for continuous support and logistical management. Many thanks to the people coordinating the scientific support at ATTO, in particular Susan Trumbore, Carlos Alberto Quesada, Bruno Takeshi, and Reiner Ditz. We express our gratitude to the data providers, Luciana Gatti (Aircraft vertical profiles), and Jacob Nelson, Sophia Walter and Martin Jung for the FLUXCOM and X-BASE NEE fluxes. W.P. and G.K. were funded by an ERC-Consolidator grant (649087) as part of the ASICA (Airborne Stable Isotopes of Carbon from the Amazon) project. Finally, we appreciate the comments received from the two reviewers during the revision phase, they contributed with improving the text and main message of the manuscript.

# Appendix A

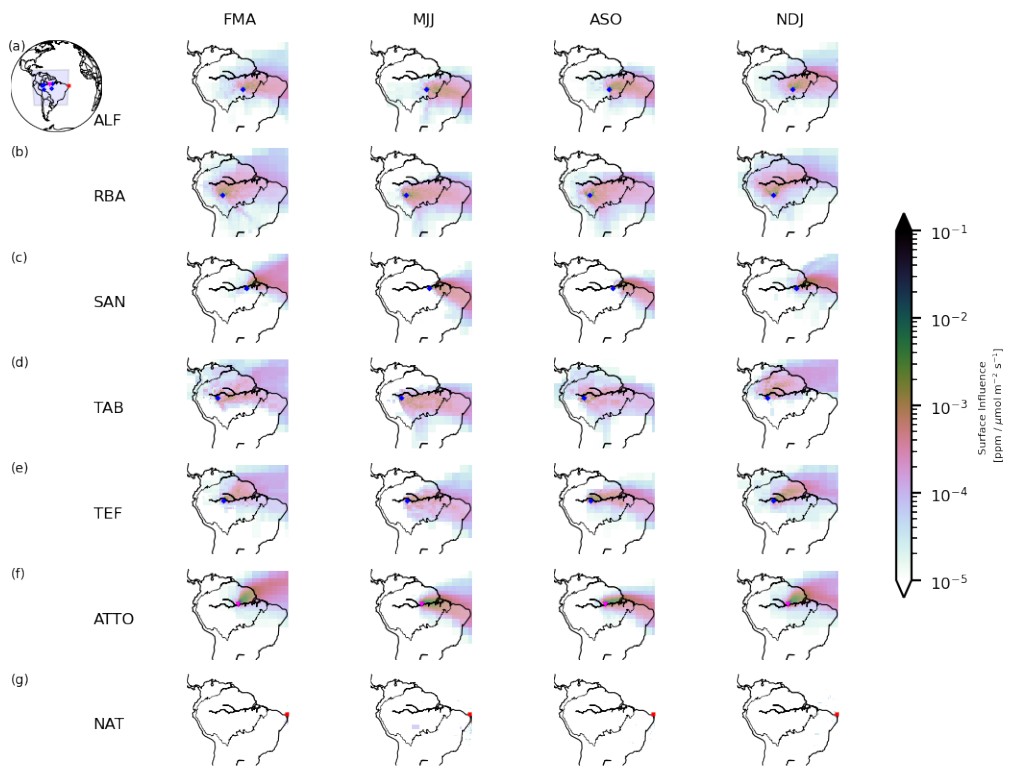

**Figure A1.** Seasonal surface influence for each station used in the regional inversion. The averaging period for each station corresponds to the period of data availability, which is site-specific, see Figure 2.

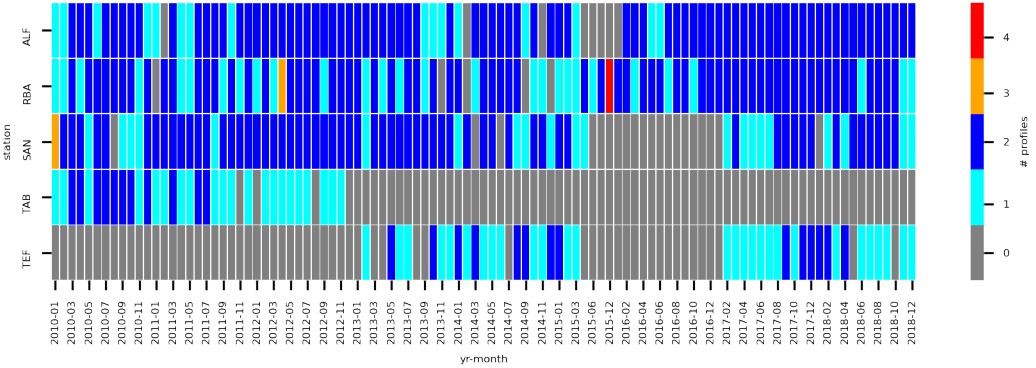

**Figure A2.** Number of aircraft profiles per month over the period of interest in the inversion. An aircraft profile goes up until 4500 m.a.s.l and on average collects samples at 14 heights.

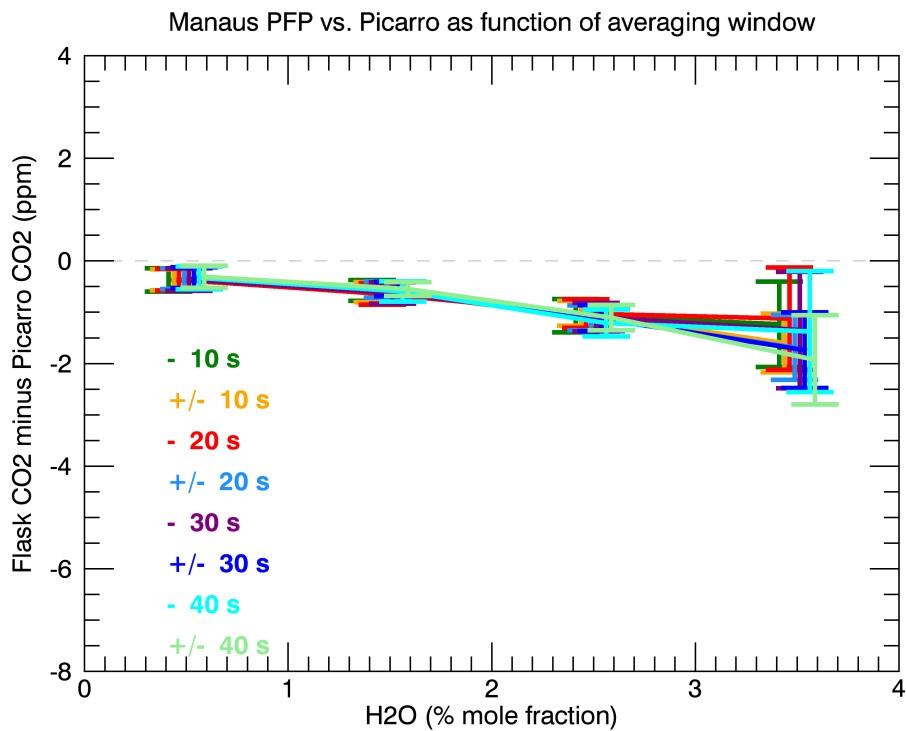

**Figure A3.** Bias between PFPs and CRDS analyzer (Picarro Inc. model G2401-m) $CO_2$ mole fractions at MAN as a function of water vapor mole fraction measured by the CRDS analyzer.

**Table A1.** Mean effect on the posterior estimates of assimilating bias corrected vertical profiles using CarboScope global and regional. The difference was calculated postFluxWVC - postFluxNonWVC, therefore positive numbers indicate that the WVC posterior flux is larger. Units in PgC year$^{-1}$.

|  | CarboScope Global | CarboScope Regional |
|---|---|---|
| Amazon and Andes Piedmont | 0.02 | 0.04 |
| Brazilian Shield Moist Forest | 0.08 | 0.10 |
| Cerrado & Caatinga | 0.10 | 0.14 |
| Guianan Shield Moist Forest | 0.04 | 0.07 |
| Amazon River Flat Plains | 0.05 | 0.09 |
| Biogeographic Amazon | 0.21 | 0.31 |
| All LAND | 0.28 | 0.50 |

**Table A2.** Eddy flux sites used to calibrate the VPRM parameters.

| Site code | Lat | Lon | Site Veg. Description | Veg. Class VPRM |
|---|---|---|---|---|
| STM-K67 | -2.85700 | -54.95900 | Primary Tropical Moist Forest | Evergreen Forest |
| STM-K77 | -3.02020 | -54.88850 | Pasture, then Agriculture | Cropland |
| STM-K83 | -3.01700 | -54.97070 | Primary Tropical Moist Forest,sel. logging Aug/Sept 2001 | Evergreen Forest |
| MAN-k34 | -2.50000 | -60.20910 | Tropical Rainforest | Evergreen Forest |
| PA-CAX | -1.74830 | -51.45360 | Tropical Forest, dense lowland tropical forest | Evergreen Forest |
| RON-FNS | -10.76180 | -62.35720 | Pasture | Grassland |
| RON-RJA | -10.07800 | -61.93310 | Tropical Dry Forest | Evergreen Forest |
| TOC-BAN | -9.824416667 | -50.15911 | Seasonally flooded Forest-Savanna Ecotone | Evergreen Forest |
| SP-PDG | -21.61947222 | -47.64989 | Savanna | Savannas |

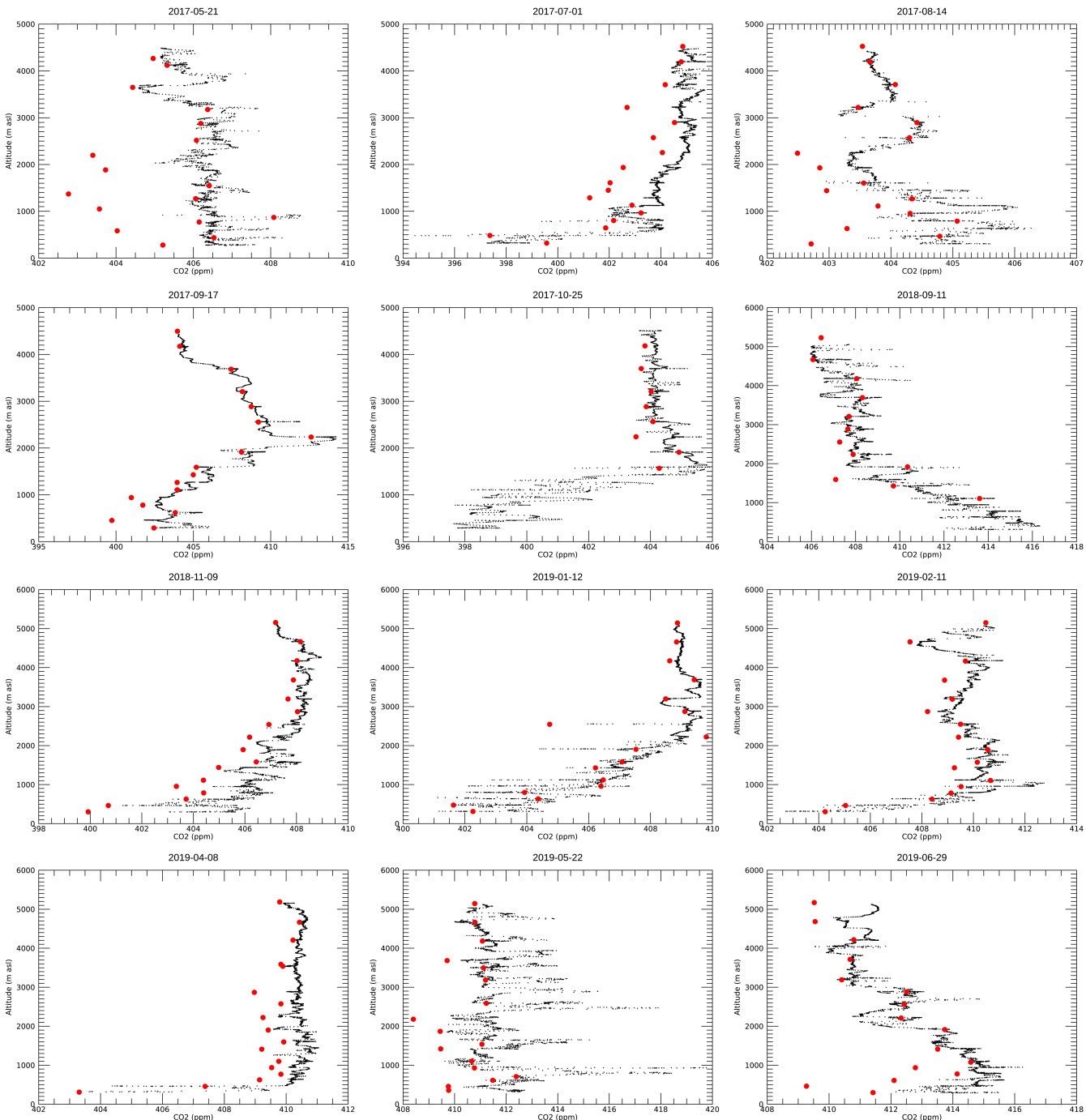

**Figure A4.** Individual vertical profiles with the in-situ (black dots) and the discrete PFPs samples in Manaus (MAN). Each in-situ point is a 1 Hz interpolated value from the calibrated native CRDS signal at 0.3 - 0.4 Hz.

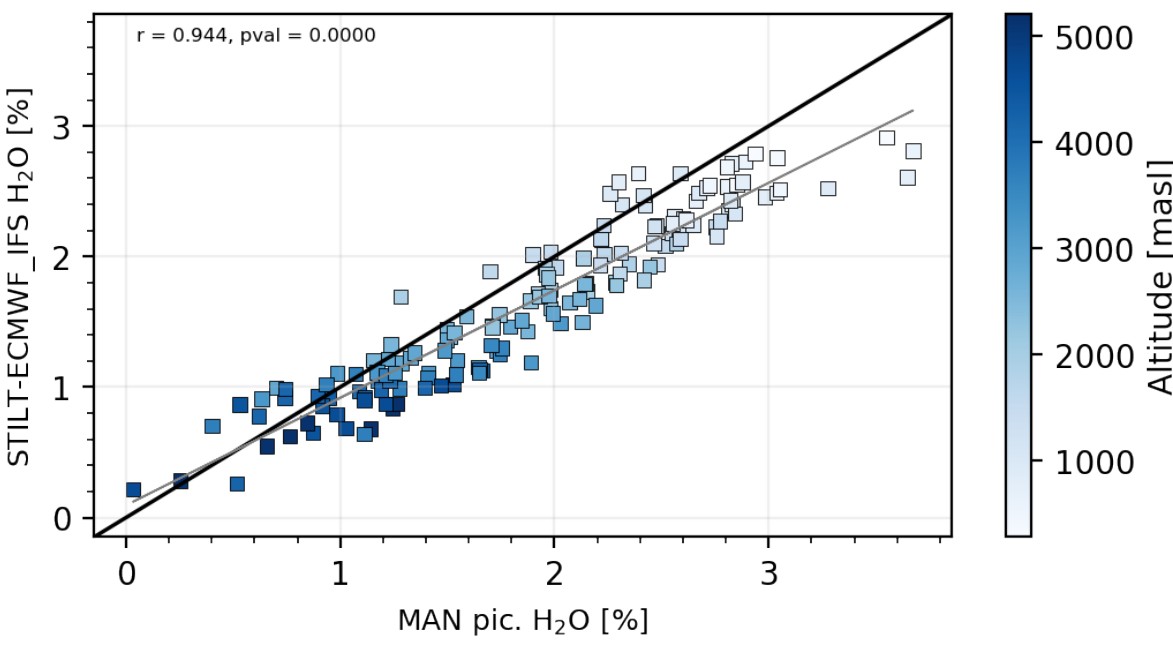

**Figure A5.** Correlation between STILT ECMWF-IFS water vapor and the measured water mole fraction at the Manaus flights with a Picarro (model G2401-m). The grey line corresponds to the predicted y using a linear regression.

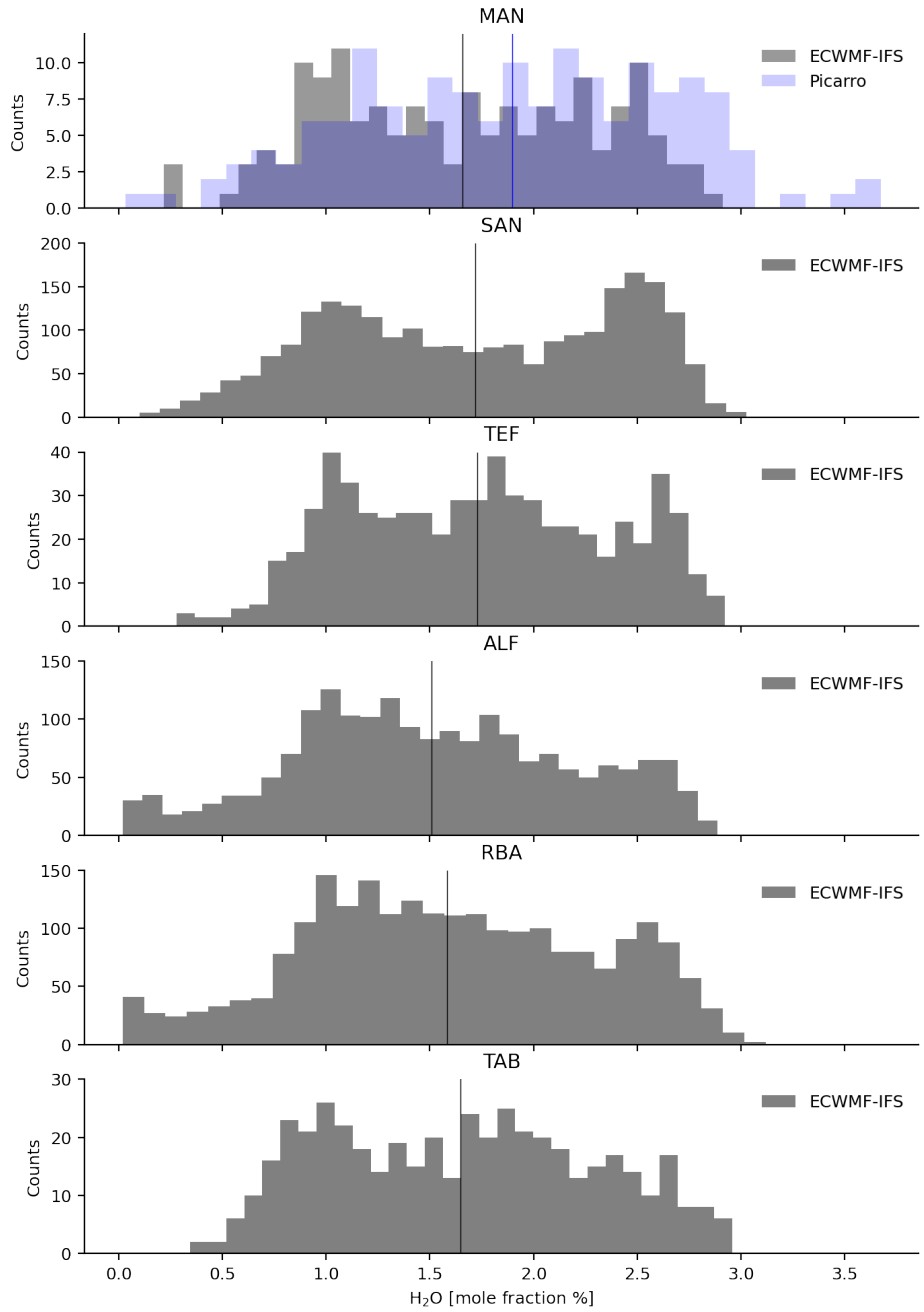

**Figure A6.** Distribution of water vapor mole fractions at all sites extracted from ECMWF-IFS and the measurements at Manaus.

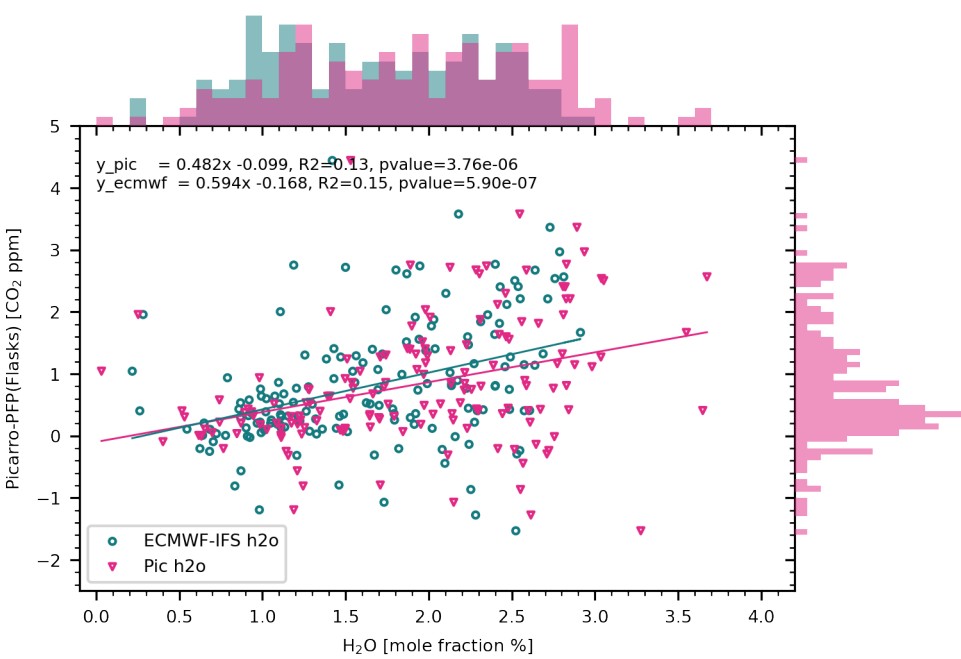

**Figure A7.** Bias between Picarro (model G2401-m) and PFPs $CO_2$ mole fractions at Manaus as a function of water vapor mole fraction measured by the Picarro (model G2401-m) and also extracted from ECMWF-IFS.

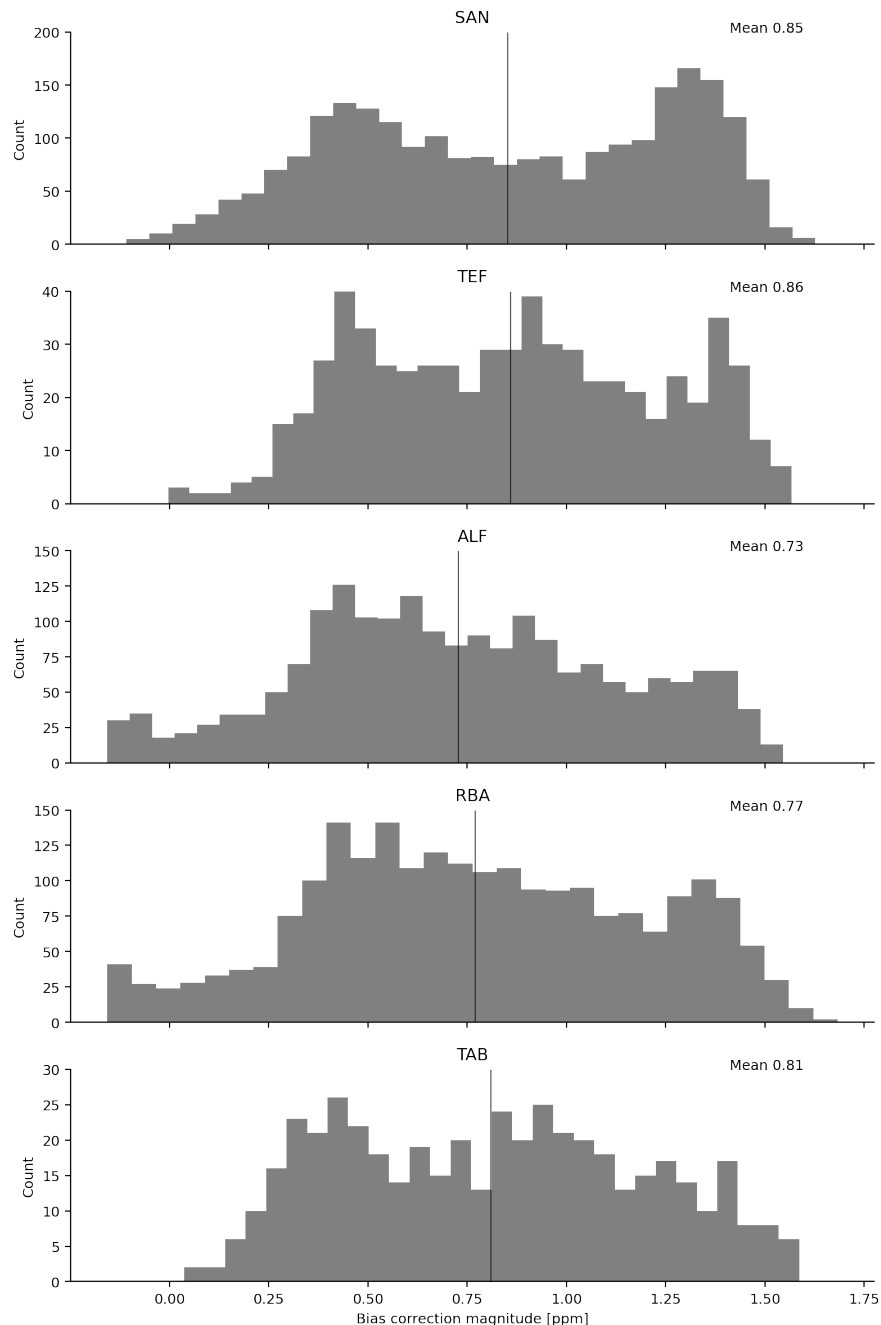

**Figure A8.** Bias (Picarro - PFP) estimated at each site using the fit to ECMWF-IFS water vapor.

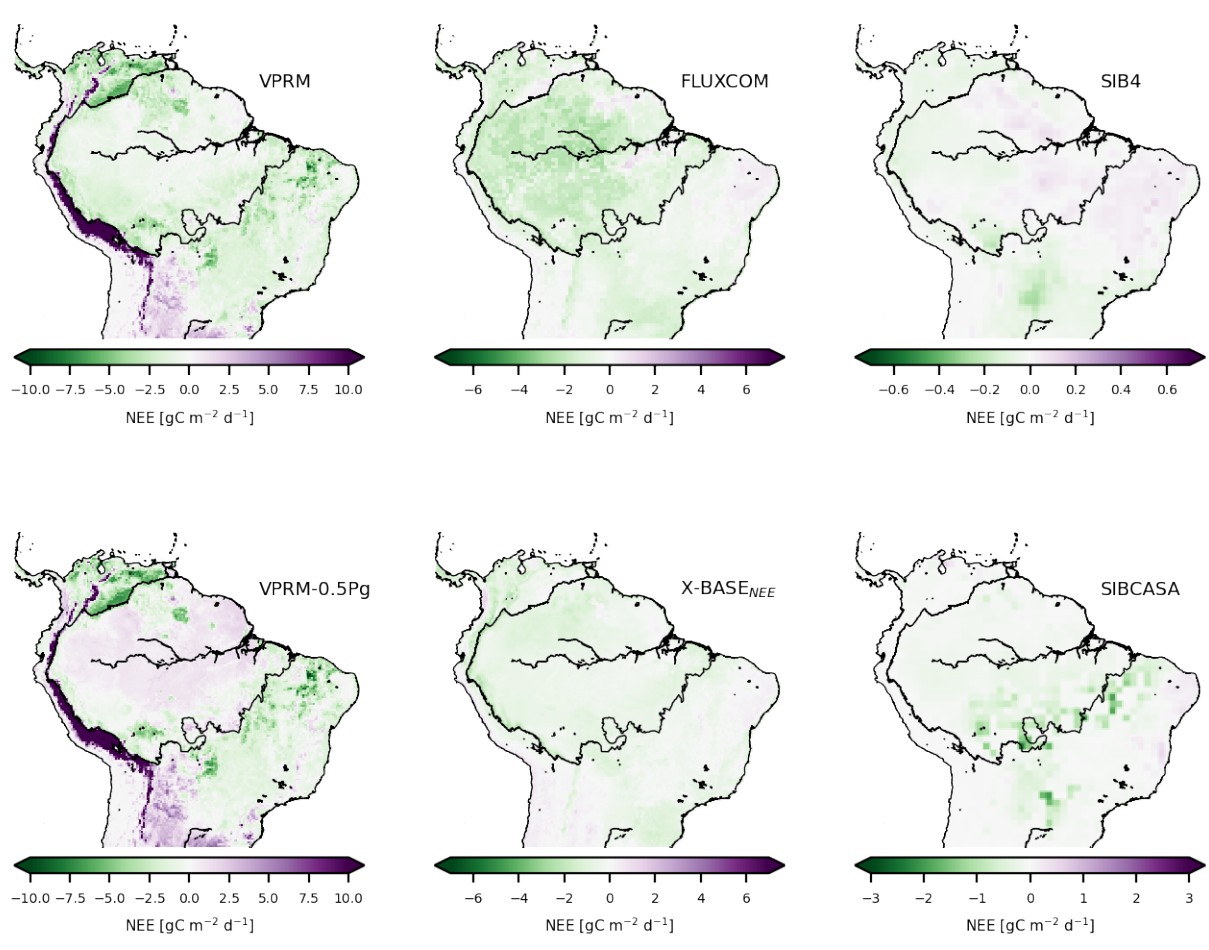

**Figure A9.** Prior mean NEE over 2010-2018 for several of the models used. Note the different range in the colorbar.

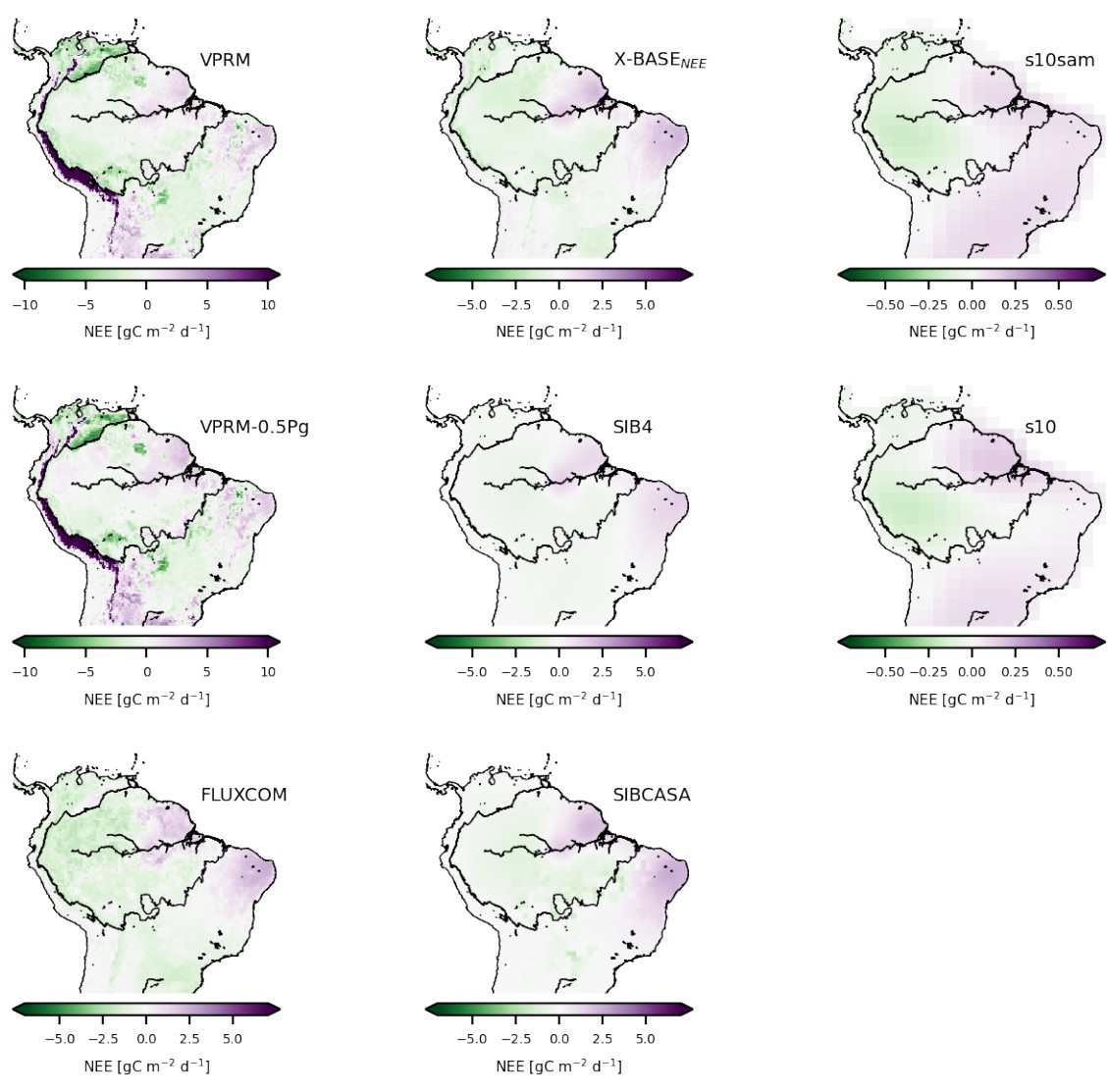

**Figure A10.** Posterior mean NEE over 2010-2018 for several of the models used. Note the different range in the colorbar.

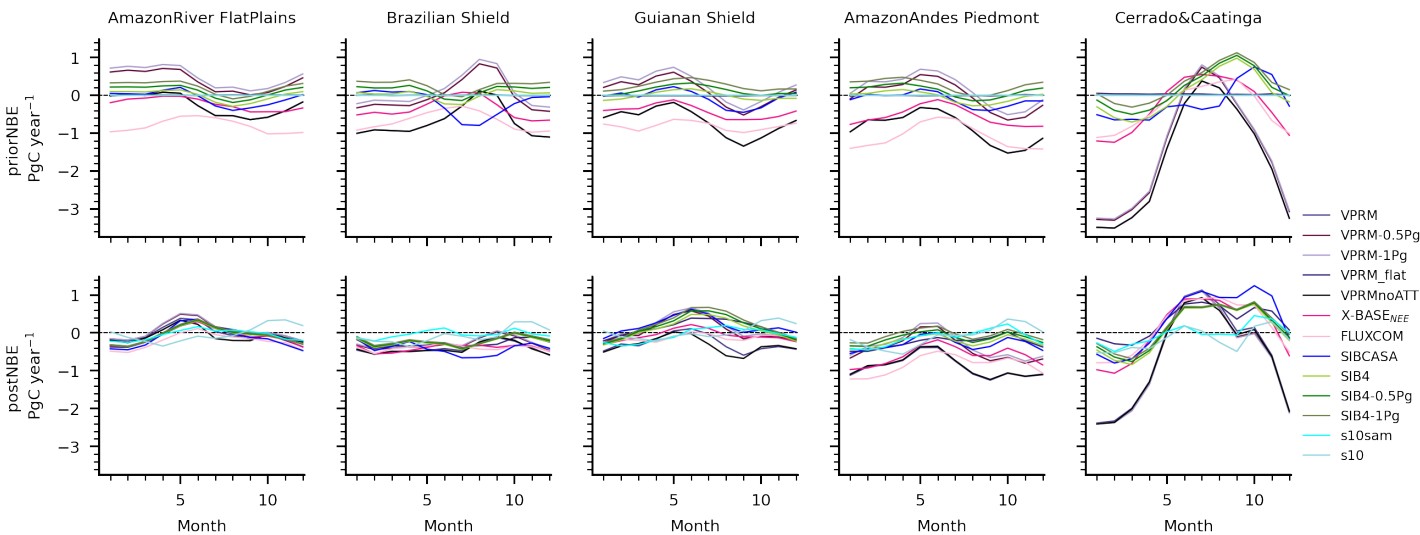

**Figure A11.** Prior (upper panel) and posterior (lower panel) seasonal cycle of NBE for each ensemble member aggregated for each region of interest.

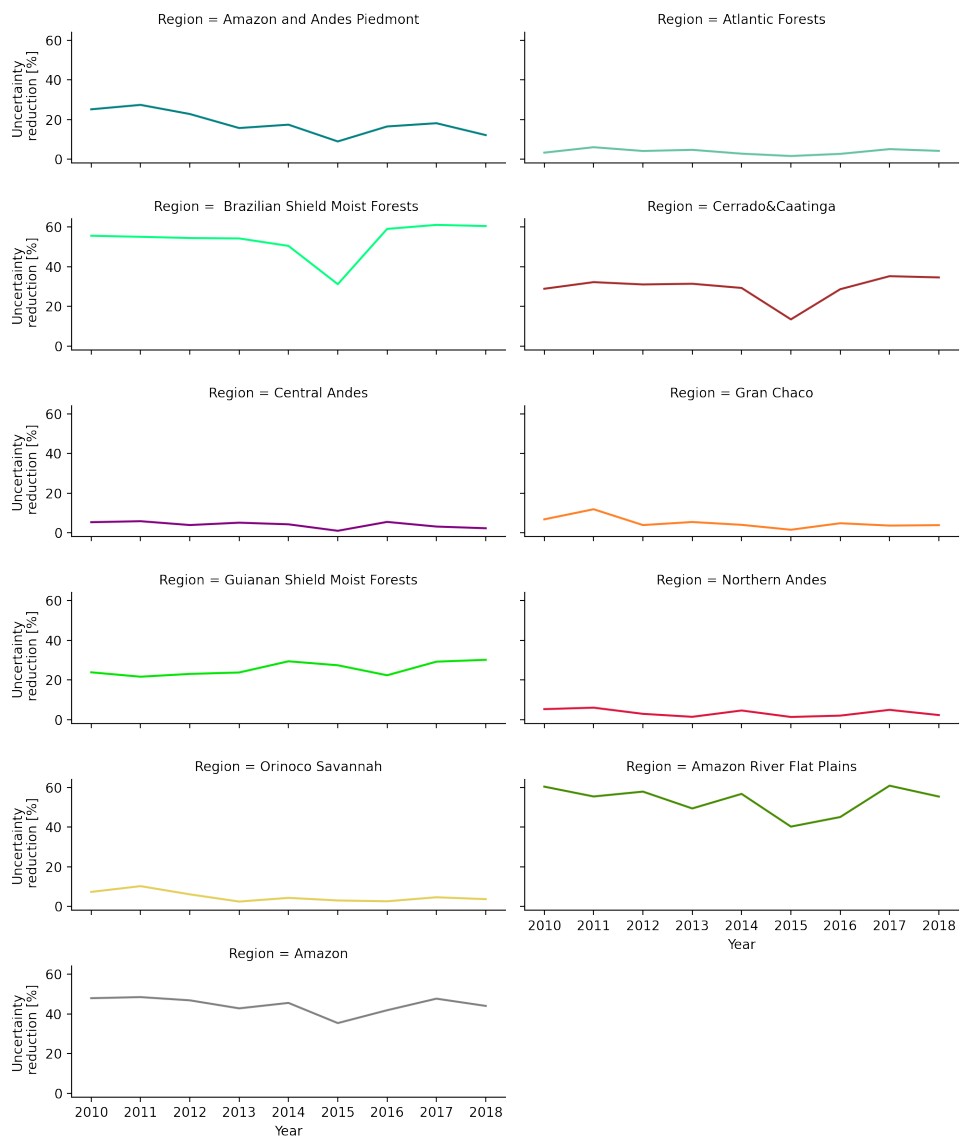

**Figure A12.** Prior to posterior uncertainty reduction throughout the complete inversion period (2010-2018).

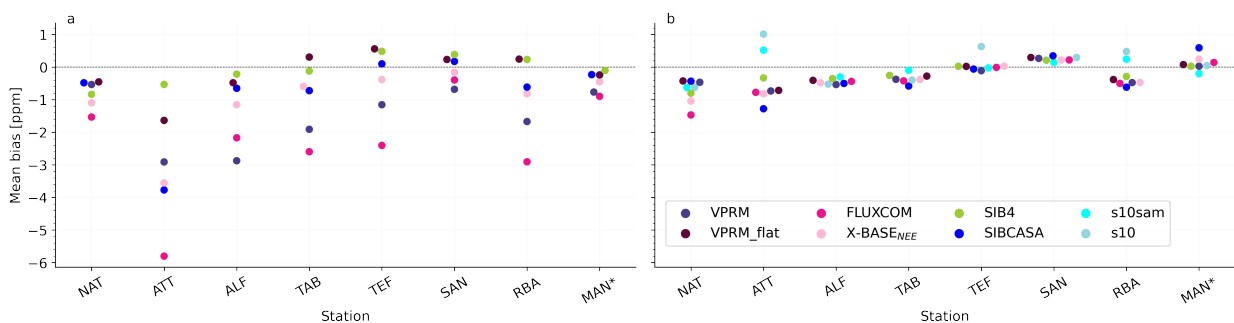

**Figure A13.** Atmospheric forward runs using the prior (a) and posterior fluxes (b). Model-data comparison is done at the sites assimilated in the inversion (NAT, ATT, ALF, TAB, TEF, SAN, RBA) and at one site that was not assimilated (MAN*, v2 (Miller et al., 2023)) as a validation site. Note that the global inversions s10 and s10sam have a zero-prior, so no forward run is available for those. In addition, note that s10 global inversion did not assimilate any of the sites shown here. At the MAN site we use aircraft profiles using a Picarro (model G2401-m) over the 2017 to 2018. For the MAN measurements the surface influence (or footprints) over were calculated for a 10s $CO_2$ mole fraction average. We used the same settings described for STILT-IFS in the methods.

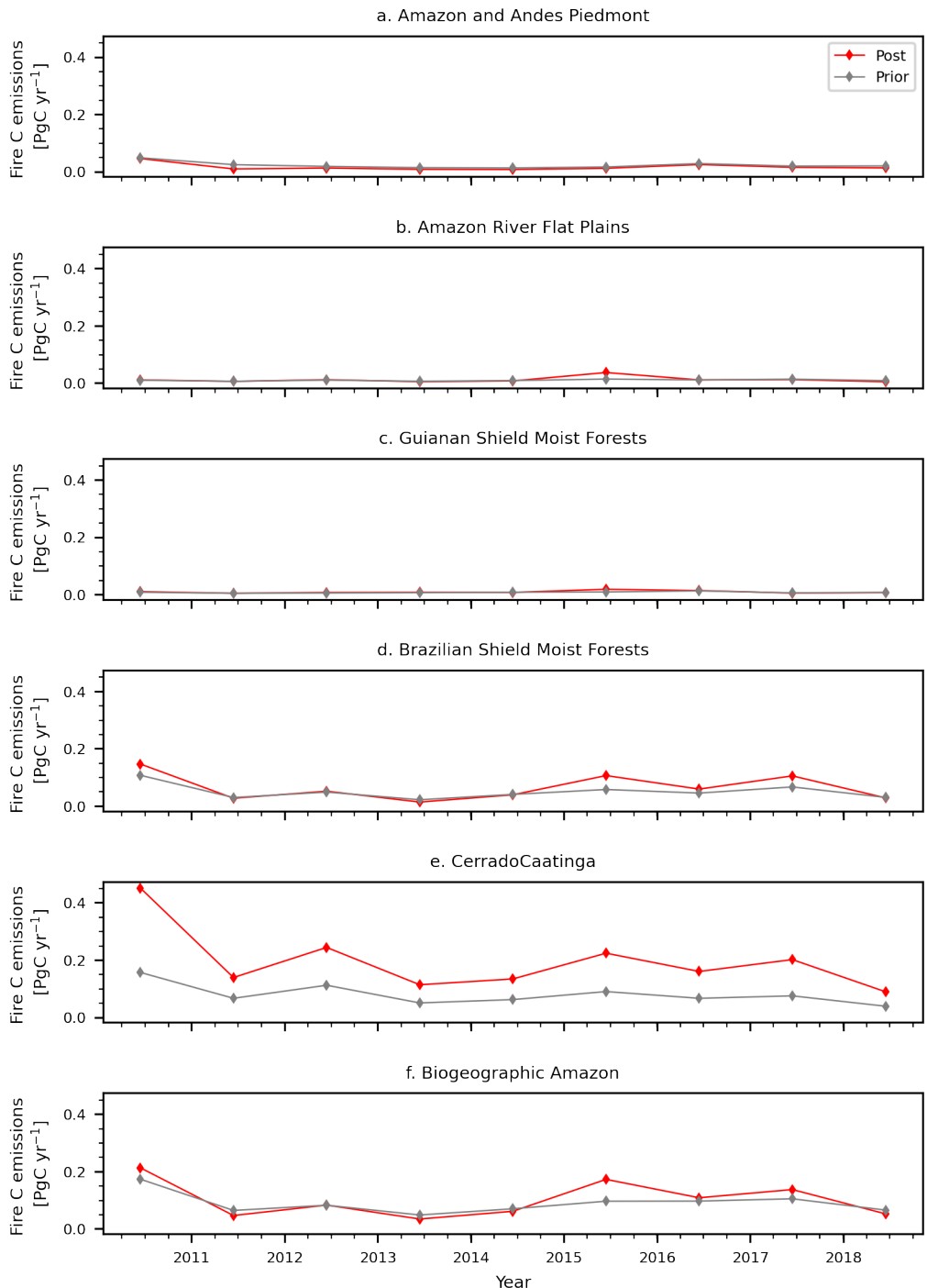

**Figure A14.** Prior and posterior GFAS fire emissions for the regions within the Biogeographic Amazon (a-d), the Cerrado & Caatinga (e) and the Biogeographic Amazon (f).

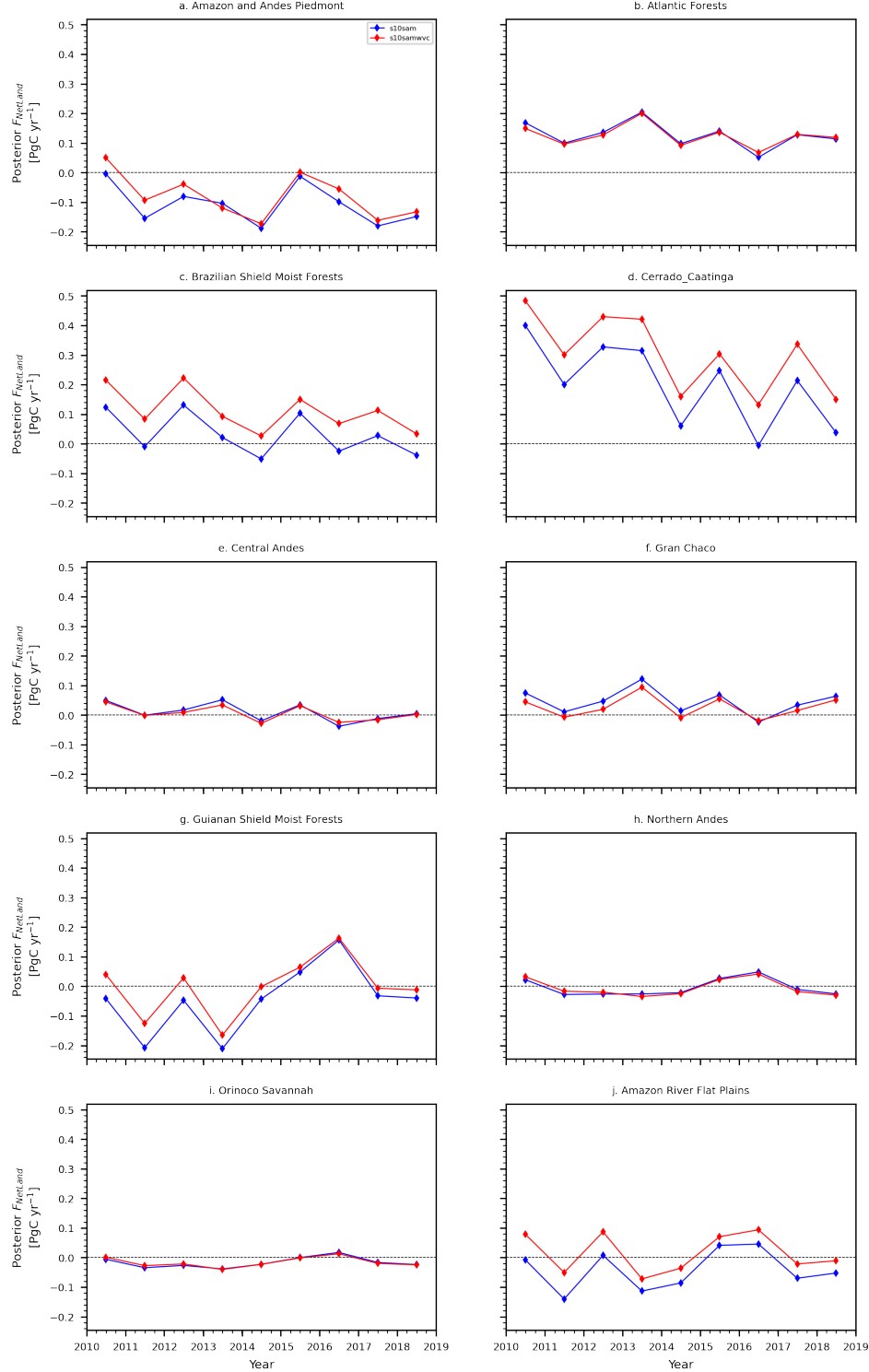

**Figure A15.** Time series of posterior fluxes for each region using the global inversion assimilating data with the water vapor correction and without it.

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
