# Peer review of "Combined CO2 measurement record indicates Amazon forest carbon uptake is offset by Savanna carbon release"

_EGUsphere, 2024_

## Author Comment (AC1)

Reviews from Referee 2 - Received on September 23, 2024

The format in which the response is addressed is the following:

1. Black text shows comment of referee. Comments are numerated as RC2.1, for the first comment. The line in the submitted draft referred to by the referee is also shown.

2. *Text in italics show the author's response and it has the same logic for numeration (e.g. AR2.1). If not stated otherwise, here the reference to figures and line numbers are based on the submitted manuscript.*

3. Red text indicates changes to the manuscript. Here the reference to Figures are based on the revised manuscript.

**General Comment**

**[RC2.1]-Review Synopsis 1**: There is an impressive amount of work here. Unfortunately, it is presented almost as a regurgitation of work completed, and lacks organization and focus. I might describe the paper as 'sprawling'. My recommendation needs to be that the manuscript is not suitable for publication and should be rejected. However, I believe that with reorganization, inclusion of some important context, and condensation this research would contribute to the body of work on the subject. I thereby also recommend resubmission.

*[AR2.1]: We appreciate the recognition of substantial work done, but we strongly disagree with the adjectives used by the reviewer in this rather informal review. To address the concerns raised we have improved the manuscript in multiple places, including a clearer explanation of the structure of this manuscript in the Introduction. Now, it provides more context for the water vapor correction and it emphasizes the main objectives of the paper: 1. to assess the observational constraint of the South American network (Aircraft + ATTO in-situ) by characterizing which areas of the continent have more uncertainty reduction. 2. to provide a continental estimate on the carbon balance, going beyond the Amazon. 3. to evaluate the effect of a water-vapor bias in the aircraft network on the posterior fluxes. For this, we suggest a correction for which we provide the equation and the offsets for each flask assimilated in the inversion. As the reviewer can see, this is the order of the initial submitted manuscript (Section 3.1, 3.2 and 3.3), which follows a logical and focused organization and writing. In other places we have attempted to improve the organization and added context based on the comments provided by the reviewer, at times also removing elements to condense the text. We believe this follows the intent of the recommendation.*

**[RC2.2]-Review Synopsis 2**: The authors are passive and unclear with regard to major findings from this research. The end of the abstract leaves the reader with a bland statement about the effect of systematic uncertainty due to water vapor on flasks and a vague exhortation for more observations. Come on. There was a lot of work done here, point the reader explicitly and clearly to what they should take away from the paper and what it tells them about how the world works.

In my reading, I find the first main point to be that the Amazon hydrologic basin (hereafter, Amazon) has a small annual sink of CO2, and the cerrado/caatinga (hereafter, cerrado) is a small source. Fine. That is known, from the work of others (e.g. Luciana Gatti). Another important finding, for me, is further support of the notion that in the Amazon the net flux is a very small residual from very large component fluxes. It is my opinion and the opinion of several of my colleagues that this is

a very important concept that is perhaps underappreciated by the carbon cycle community at large. These are mentioned in the abstract, and in my opinion the paper would benefit from more emphasis on them.

*[AR2.2]: We thank the reviewer for encouraging us to be more clear with the take-home messages of the paper and we have strengthened our voice (as opposed to passive suggestions) in this regard in the abstract and throughout the paper. However, our "exhortation" to more observations is far from vague. We show that the Amazon Andes Piedmont is not well constrained and is the region with lower uncertainty reduction inside the Amazon. Note that this is something that -to our knowledge- has not been done before in previous studies using the aircraft network. We disagree with the reviewer's statement that the Amazon and Cerrado/Caatinga sink to source gradient is already known or that it was already shown. In Gatti et al 2021, 2023 there was an explicit choice of leaving the Cerrado outside of the analysis. What they report is very clearly focused in the Amazon forest biome and the areas of influence were artificially cut at the Amazon forest border, ignoring atmospheric transport beyond the southeastern border of their domain as acknowledged in the Methods sections of these publications.*

[RC2.3]-**Review Synopsis 3**: Water vapor in flasks. Oh, boy. I understand that this is a topic of much discussion, some of it impassioned. There is a group of people who believe that this issue calls into question some scientific results, and another group who believe that it is not that big of a deal. I am agnostic on the issue, but I see the division in the community. The authors obviously spent a lot of effort studying the issue, but after reading the paper I do not come away with any feeling of resolution. Does the water vapor issue call into question previously published results or not? If you can, make a statement one way or the other. If you can't, then the question is still open and you might want to think about how much attention you pay to it in the manuscript.

*[AR2.3]: We thank the reviewer for the questions posed. However, we wonder if water vapor in flasks is really a topic of much discussion, and if so, among which community? The reviewer has apparently been exposed to some of the third-hand discussion of the water vapor influence on $CO_2$ in pressurized flasks and come away with the impression that there is controversy. However, firstly, it is unclear to which discussions and to which 'groups of people' the reviewer refers to and it would be more appropriate if he would support the comments with references. Secondly, there is in fact no controversy about the fact that humid air in glass flasks when pressurized to 2.7 bar leads to $CO_2$ artifacts during sample storage. From the point of view of the peer-reviewed literature there is uniform agreement that water vapor is a problem when flasks are not dried at the time of sampling [Baier et al., 2020, Paul et al., 2020, Gatti et al., 2021]. This includes the work of Gatti et al., 2021 (see their Methods) which cites Baier et al., 2020, and discusses the potential impact of water vapor-related $CO_2$ mole fraction biases on derived Amazonian net $CO_2$ fluxes. Additionally, NOAA has flagged many thousands of their $CO_2$ measurements from aircraft in humid environments in acknowledgement of the water vapor influence on the reported $CO_2$ mole fractions. Moreover, Gatti et al, 2021 write "[we] found depletions in PFP $CO_2$ with the same tendency as those in ref.56 [i.e. Baier et al., 2020]. This influence is probably greater near the surface, as humidity increases at lower altitudes. Thus, the true $CO_2$ below the boundary layer (about 1.2 km from the surface) may be higher than measured, which means that fluxes at present may be underestimated." And the Gatti et al. 2021 hypothesis is confirmed quantitatively in this study. So, in many ways, there is a resolution (or at the very least substantial progress) on an initial question posed in the literature that we present here. As we state in Section 2.1.5, lines 230-232, our goal is not to question the effect of water vapor in analyzed $CO_2$ mole fractions in the flasks, but to quantify the effect of the bias in the posterior fluxes. We show here that there is a significant signal as a function of water vapor concentration, regardless of the averaging time window used for the Picarro data (Figure A3). Based on this, we provide a method to correct for it and quantify its effect on posterior fluxes, which is a first effort to our knowledge. As we show in Figure 7, considering the effect of water vapor in flasks results in a reduced sink of carbon (in the Amazon) and a larger source (in the Cerrado and Caatinga) (Figure A14), but the inter-annual*

*variability and spatial patterns are not affected. We do make a statement about this, see lines 423-427 of the submitted manuscript, and defend it in the paper (see Section 3.3). Moreover, we have modified the manuscript to make this more clear, see the additions in the Abstract and in the Introduction of the revised manuscript. In addition, we refer the reviewer to AR2.13 in this document for more on the water vapor topic.*

**[RC2.4]-Review Synopsis 4**: During multiple readings, what I took from the paper is that the Amazon is a small sink, the cerrado/caatinga is a small source, and these net fluxes are small residuals from large component fluxes. The authors were able to obtain this result with a method that is consistent with yet independent of other estimates. While not earthshaking, I see that as a worthwhile result that adds to an emerging body of work and pushes forward a narrative that many in the carbon cycle community might not fully appreciate.

I can't tell the authors what paper to write. I can say that there is a lot to keep track of in the paper as is, and it can be hard to keep focus on what is important. The reader is trying to follow information about different regions, water vapor, priors, fire, rivers, mechanisms, spatial attribution of flux signals, and more.

I believe a condensed paper with tightened focus would be beneficial and well-received by the community. A lot of material could be moved to Supplemental Information (fire, rivers) or greatly condensed. There is good material here, I look forward to seeing a revision.

*[AR2.4]: We appreciate the comments calling for a more condensed paper. We have shortened the paper where feasible. In particular, Section 3.1 was shortened. However, the reference to carbon fluxes from fires and rivers is kept, as these are crucial components for the net carbon exchange in South America and often ignored (i.e. Rivers). As mentioned in 4.1.1, river outgassing could be important at the regional scale, but the constraint on this flux is still highly uncertain. In a modeling exercise, we showed that for representing the seasonal cycle of mole fractions at ATTO, river outgassing was relevant [Botía et al., 2022]. Therefore, we prefer to keep the reference to Rivers to make clear that NBE has a riverine flux component.*

**More Detailed Review:**

**[RC2.5]-Observations**: An exhortation to the community for more observational constraint is made several times in the manuscript. This mention is unnecessary, everybody knows it, campaigns are being planned even as we speak. This can be seen as an excuse, or a defense for the authors against being wrong. Make your statement with the available data and analysis. If you're proven wrong later, write another paper describing how, and show what you've learned from it.

*[AR2.5]: We agree that calling for more observations can be seen as unnecessary and perhaps everybody knows it, as the reviewer suggests. However, in situ or discrete flask sampling in tropical South America, as opposed to Europe or North America, is extremely limited and our call to more observations is argued in detail. We say specifically where these observations should be done (i.e. Amazon Andes Piedmont), an area that coincides with a region where cloud cover impacts satellite retrievals (see Figure 1, in Frankenberg et al. [2024]). We also indicate where the observational network used has limitations, an important contribution of this study. Furthermore, the fact that campaigns are being planned is nice, but it is important to emphasize that one month or even a one-year campaign will tell us nothing about the long-term mean carbon exchange in the Amazon, or about its inter-annual variability. We believe that our call for more observations in the Amazon Andes Piedmont is valid and well-argumented, more so in light of the low data yields from the current satellite missions like OCO-2 and OCO-3 [Frankenberg et al., 2024].*

**[RC2.6]-Background-1**: A few things were conspicuous to me by their absence. There is no mention of the OCO-2 Flux MIP (Crowell et al., 2019; Liu et al., 2017; Peiro et al., 2021). Inversions constrained

using retrievals of XCO2 provide a nice counterpart to inversions constrained with flask data. There are published papers (Crowell et al., 2019; Peiro et al., 2021) that describe what XCO2 inversions have shown in the Amazon/cerrado. How do your results compare? For that matter, the inversion results are available on a 1x1 grid. How do these results compare with your regional partitioning?

*[AR2.6]: Thank you for the comment, it is a valid suggestion, yet we believe that a comparison to the OCO-2 Flux MIP inversions using our regional masks is out of the scope of this manuscript. Our focus is to evaluate the constraint of the observational network used here and not to perform an inversion-ensemble analysis. The latter would widen the scope of the study, contradicting the call for conciseness and focus that the reviewer made. We have read the papers suggested and find that even for a comparison referring to the regions presented in these two global studies, it is difficult to draw meaningful conclusions. In Peiro et al., and Crowell et al., the subdivision of South America is done in three regions: 1. Northern South America: including only the north (north of the equator) of the Amazon basin, the Orinoco basin, a part of central America and the Caribean islands. 2. Southern Tropical South America, which includes a large part of the Cerrado and the southern part of the Amazon and 3. South America Temperate, which includes the Cerrado and Caatinga, extending until Ushuaia, the southernmost point on the continent. None of these regions coincide with our regional distribution, not even having a defined region for the Amazon basin. We therefore find a comparison to total fluxes not meaningful, however in the Discussion section of the revised manuscript we have now included a reference to those studies. The text added reads as follows: To further reduce the uncertainty in this domain, top-down estimates could combine in-situ data with satellite retrievals. The inversions assimilating data from the Orbiting Carbon Observatory 2 (OCO2) [Liu et al., 2017, Crowell et al., 2019, Peiro et al., 2022, Wang et al., 2023] have shown that remotely-sensed $CO_2$ columns can provide a valuable constraint of net carbon exchange in tropical regions. However, the OCO2-inversions are still limited by cloud coverage during the wet season [Massie et al., 2017, Peiro et al., 2022] and the adjustment of the prior can be biased to dry season retrievals [Crowell et al., 2019]. Nevertheless, our results for the response to the El Niño 2015/2016 coincide with the OCO2 inversions [Liu et al., 2017, Crowell et al., 2019, Peiro et al., 2022] in a carbon source in 2015 and 2016. Yet, a direct comparison of the magnitude in those studies to our results is difficult, as the area for South America in Liu et al. [2017] includes parts of the 'Cerrado & Caatinga' regions and in Crowell et al. [2019], Peiro et al. [2022] they divide South America in three parts: 1. Northern South America: including only the north (north of the equator) of the Amazon basin, the Orinoco basin, a part of central America and the Caribbean islands. 2. Southern Tropical South America, which includes a large part of the Cerrado and the southern part of the Amazon and 3. South America Temperate, which includes the Cerrado and Caatinga, extending until the southernmost point on the continent. None of these regions coincide with our regional distribution, therefore, using our domain definition on the OCO2-MIP results should be part of a next study, as it is out of the scope of this one.*

**[RC2.7]-Background-2, ENSO (2015-2016 El Nino)**: Maybe this goes into the analysis/discussion, but there really isn't anything in the manuscript about this event. What sort of ENSO response did CarboScope determine? How did that compare with other estimates (e.g. (Crowell et al., 2019; Liu et al., 2017; Peiro et al., 2021).

*[AR2.7]: We acknowledge that the response to El Niño in 2015/2016 is not described in the manuscript. We have added a reference to it in Section 3.3, reading as follows: Assuming that the correction brings the observational data closer to the truth implies that the Amazon is a weaker sink of carbon and that the 'Cerrado & Caatinga' is a larger source. We do not find a strong spatial shift of fluxes within the biogeographic Amazon due to the correction, nor do we find a strong impact on the interannual variations. The response to drought in 2010, 2015 and 2016 is affected in the absolute flux magnitude, but the year-to-year variability remains the same (Figure 7). Therefore, the findings of our sensitivity tests support the hypothesis in Gatti et al. [2023] that the water vapor bias mainly*

*affects the absolute annual flux magnitudes. In both cases, with and without correction, our estimates for the total carbon loss to the atmosphere in the Amazon during 2015 and 2016, are lower (from 0.15 to 0.3 PgC) than other studies [Liu et al., 2017, Gloor et al., 2018]. Our total net flux is closer to the 0.5 ± 0.3 PgC from Gloor et al. [2018], but note that they used a time period from September 2015 to June 2016 and the area they refer to as Amazonia is not clearly defined. Compared to the 1.6 ± 0.29 PgC in Liu et al. [2017], our estimates are much lower, but the difference in area is large as they refer to tropical South America, including parts of the 'Cerrado & Caatinga' biomes and central America. Furthermore, they focus their study on the anomaly relative to 2011 at the peak of the El Niño phase.*

**[RC2.8]-Background-3, Transport**: "errors in atmospheric transport" are mentioned on line 51, but nothing further is said. An analysis of different transport models (TM5 and GEOS) is described in (Schuh et al., 2019). Does the transport in CarboScope align with either of these? How might this influence results? At the very least the Schuh paper should be mentioned; if the findings of that paper are meaningful for CarboScope inversions (or not), the reader should be made aware.

*[AR2.8]: We thank the reviewer for referring us to Schuh et al (2019). However, in the submitted manuscript we do say more about how we deal with the model-data mismatch uncertainty, which includes the representation error of the measurements within the transport model. We refer the reviewer to Section 2.1.4, lines 157-174 in the submitted manuscript. Now, regarding how TM3 (CarboScope) aligns with either TM5 or GEOS-Chem based on the Schuh et al. findings, we can state the following. First, part of the main findings in Schuh et al., indicate that GEOS-Chem has (compared to TM5) a faster vertical and meridional transport that quickly ventilates northern midlatitudes, leading to a more rapid mixing of tracers emitted in the northern midlatitudes into the troposphere in the Southern Hemisphere. Second, it is worth noting that TM3 is the precursor of TM5 (see Section 2.1 in Krol et al. [2005]), so they share several parametrizations. Most relevant to tracer transport, the schemes used for advection [Russell and Lerner, 1981], vertical transport due to convection [Tiedtke, 1989] and vertical diffusion in the free troposhere [Louis, 1979] are the same. Vertical transport in the boundary layer in TM3 is done using the Louis [1979] scheme, while in TM5 the revised LTG (Louis, Tiedtke and Geleyn) scheme of Holtslag and Boville [1993] is used. Therefore, we expect that vertical and meridional transport and thus hemispheric mixing in TM5 should be similar to that of TM3. In other words, we expect TM3 to have slower vertical and meridional transport than GEOS-Chem and therefore also present a similar effect on estimated fluxes for the latitudinal bands shown in Schuh et al., (2019): TM5 having a weaker sink from 45 N to 90 N, but a larger one from 0 to 45 N, and a stronger global integrated carbon sink than GEOS-Chem. However, the implications for our inversions are difficult to assess as the model-to-model (TM5 and GEOS-Chem) difference and the latitudinal compensation are more pronounced north of the equator (see Section 3.3.2 and Figure 7 in Schuh et al.,), and our domain spans 30 S to 14 N. Nevertheless, we can speculate that if the CarboScope Global Inversion (TM3) has a stronger global carbon sink than GEOS-Chem, then the contribution of our South American domain to that sink will propagate to the regional inversion via Step 1 of the two-step scheme. We have added part of this response to the Discussion: In addition, Schuh et al. [2019] showed that atmospheric transport models (e.g. TM5) having a slow vertical and meridional transport in northern mid-latitudes can have a weaker sink from 45 N to 90 N, but a larger one from 0 to 45 N and thus result in a stronger global integrated carbon sink compared to the fast vertical mixing models (e.g. GEOS-Chem). We speculate that as a precursor of TM5, the model we used in the global inversion (i.e. TM3) could have a stronger global carbon sink due to a slower vertical and meridional transport, so the contribution of our South American domain to that carbon sink will propagate to the regional inversion via Step 1 of the two-step scheme. The quantification of this potential effect is a source of uncertainty in our regional inversions that should be further quantified.*

**[RC2.9] Regions**: I frequently had to refer to Figure 1A to figure out what the authors were talking about. By the end of the paper, my impression is that I should primarily be concerned with the Amazon region and the cerrado region, and that there is heterogeneity in the Amazon region with

regard to sub-regions (Amazon River Flat Plains, Amazon and Andes Piedmont, Guianian Shield Moist Forest, Brazilian Shield Moist Forest). I'm not sure why the other regions are included in the map or the analysis-are they adding to the scientific narrative, or just thrown. In my opinion the 'story' would be much clearer (and more concise) if the analysis was limited to the Amazon and cerrado regions, with discussion of Amazon heterogeneity included with regard to the sub-regions within the Amazon basin region. This brings up another point-Figure 1B suggests that there really isn't enough surface influence in these other regions (e.g. Orinoco Savanna, Central Andes, etc) to have much confidence in results obtained there. So why include them in the paper? It adds confusion to the analysis. When the authors say "there is almost no information" that's a pretty strong suggestion that these regions should not be included in the paper submitted for publication.

*[AR2.9]: Following the suggestion by the reviewer, we have modified Figure 1 and 2, masking out the regions with no observational constraint. Furthermore, the short reference to the other regions (Orinoco, northern and central Andes, the Gran Chaco and Atlantic forest) was left out of the revised manuscript. We thereby more strongly focus on the Amazon, the Cerrado and Caatinga, and the subregions within the Amazon.*

[RC2.10] **Regions**: Another idea I found myself thinking about while reading about the regional analyses was this: How much information do you need to be able to partition optimized fluxes into smaller and smaller regions? In my discussions with inversion modelers, I see a tension between wanting to describe fluxes with higher and higher spatial resolution, and not wanting to overstep what the data can tell you. I'm thinking about work done by Martha Butler and Thomas Lauvaux (not sure I know the proper citations), among others. This goes back to the previous paragraph and my concern that the authors are making statements about regional flux in regions where there is not much information about the local contribution to flux. The authors should justify, even if only briefly, **why** they can subdivide an optimized flux into the small regions described in the paper.

*[AR2.10]: As we are using a regional inversion with 0.25 x 0.25 degree of spatial resolution with a correlation length of about 200 km, the posterior budgets as well as posterior uncertainties for these subregions within the Amazon can be individually quantified. We have added this justification in Section 2.1.3.*

[RC2.11] **Carboscope description**: I had to read multiple papers to get my head around Carbo-Scope. I have worked with the inversion community for years, and I'm familiar with the general idea, but understanding CarboScope took some doing. Not sure I'm there yet. All this is to say that the description in Section 2.1 was not clear to me. I'm not expert enough to say exactly how this section should be modified, but I would recommend that the author who writes this section should work with someone who is not as well-versed in inversions to make sure the description is understandable to someone who does not work with inversions on a daily basis. I think this work will ultimately reach a wider audience, so the effort taken to describe CarboScope in a more 'accessible' manner will be well-spent.

*[AR2.11]: We have included a diagram as a complement to the text in Section 2.1. We believe this will add clarity explaining the Two-step scheme and further illustrates how the Global Inversion is coupled to the Regional one. The diagram is added here as well for easy access:*

[Figure]

Figure 1: CarboScope Regional Inversion Two-step scheme [Rödenbeck et al., 2009, Trusilova et al., 2010] flow diagram showing the inputs (purple polygons) to the processes (blue squares) and their specific output (green circles).

**[RC2.12] Observational Network**: I'm generally happy with this section. But I am curious: Can you justify assuming that the weekly error correlations are driven by synoptic-scale transport variability? That might make sense in extratropical regions where baroclinic systems frequently operate on something approximating a seven day cycle. I haven't read Kontouris (2018) or Munassaur (2021), but they are both studies based in Europe. In the tropics, however, this assumption may not hold: the weather is very different, with variability imposed by convection and squall lines. If there is published literature to support treating the tropics and extratropics similarly, cite it. If not, the authors might think about what this means. I'll leave it up to them as to how much they want to talk about this in a revised paper.

*[AR2.12]: As we answered to reviewer 1: Continuous data have always been treated that way in CarboScope, originally this was referred to as "data density weighting" introduced in Rödenbeck et al., 2005. For the regional inversions with CarboScope-Regional, the justification of model-data mismatch error correlation has been introduced in Kountouris et al., 2018b. So even though the motivation for the data density weighting is not the same in Europe as in South America, we nevertheless need to use it to allow for a combined use of flask and continuous data in the inversion. We agree that this assumption calls for additional sensitivity tests and we acknowledge it in the Discussion. We thank the reviewer for this comment and have added part of this reply in Section 2.1.4. It reads now as follows: The model-data mismatch uncertainty (including the representation error of the measurements within the transport model) for the three types of sites (in-situ tower, aircraft, and weekly flasks) is chosen to be 1.5 ppm for weekly time scales, following common practice in CarboScope global [Rödenbeck, 2005, Rödenbeck et al., 2018] which assimilates a large set of weekly flask samples. To assimilate multiple data streams, we apply a data density weighting Rödenbeck [2005]: For the hourly ATTO data, the error will be inflated by $\sqrt{N_{hours/week}}$ (details see Kountouris et al. [2018]), while for aircraft profiles (composed of several flasks) the error is scaled with $\sqrt{N_{flasks/profile}}$. The data-density weighting practically ensures that one week of hourly ATTO observations, one aircraft profile, or one weekly flask sample have the same weight in the inversion, reflecting the assumption that they provide the same amount of information due to roughly weekly error correlations.*

**[RC2.13] Systematic Uncertainties (flask moisture)-1**: The 'fix' employed here is interesting, but I'm not sure that the bias correction employed here provides a conclusive resolution. The wording in the section certainly doesn't suggest so. If the authors think they can make a declarative statement about this issue, then make it. Some thoughts I have about this section: Is this a universal problem with flasks gathered in humid environments? If so, is it fair to single out research in one location (Amazon) if other regions (e.g. Southeastern USA) have the same issue?

*[AR2.13]: We have made a declarative statement about the issue in the end of Section 2.1.5. It reads: We note that this approach was implemented to diagnostically quantify the effect of water vapor in the PFPs and how that propagates to the estimated fluxes in an inversion system. The effects of water vapor on not-dried flask samples has been established and documented previously [Baier et al., 2020, Paul et al., 2020]. Here, we establish the offsets on this specific set of flask samples collected over the Amazon. The offsets used in this study are provided as a public dataset (https://edmond.mpg.de/privateurl.xhtml?token=a6fba176-8a6b-4b59-a371-3acc804adaf9), such that the community can use them in their inversion systems and compare their magnitude to other correction methods. Furthermore, it is a universal problem with flasks that are not dried properly at the time of sampling. As we stated earlier (AR2.3), NOAA GMD has decided to retroactively flag all their reported $CO_2$ mole fractions from undried PFP flasks in the period 2000-now. Of course, in less humid environments the effect will be attenuated, but in the Amazon this is not the case. In Section 2.1.5 we added a statement emphasizing that the problem is with undried flasks at sampling time, it reads as follows: Gatti et al. [2023] (Methods Section) reported a source of uncertainty in the aircraft profiles given by moisture in undried flask samples at time of collection. Such moisture can lead to biases in the measured $CO_2$ mole fractions.*

**[RC2.14] Systematic Uncertainties (flask moisture)-2**: In figure A6 I see the linear fits, but I don't see any statistics about tightness of fit. I don't think I'm going out on a limb when I say that the r2 is probably not a large number for either line (flask vs. Picarro, or ECMWF vs. Picarro). Is the variability explained more or less than half? What would that value suggest for interpretation, either of the initial 'problem' or of the 'solution'?

*[AR2.14]: The explained variability (R2) was added to Figure A6. The reviewer is right when stating that the explained variability is low (13-15%), but note that this is what drives the systematic effect dependent on the water vapor concentration. The linear fit used based on the water vapor concentration from ECMWF-IFS explains 15% of the variability in the bias (Picarro - PFP) and after taking it out there is no systematic water-vapor effect anymore. The rest of the variability responsible for much of the variance remains, having a larger spread at high water and normally distributed after removing the linear trend (see Figure 2 ). Furthermore, as the slope is highly significant (pvalue << 0.01, N=158) the ECMWF-IFS water vapor concentration is still a good predictor of the bias. Having knowledge about the water vapor effect, we preferred to perform sensitivity tests to assess how this can affect the retrieved fluxes in an inversion, and with this provide a reference to the community.*

[Figure]

Figure 2: Residuals of Figure A6 for the fit using the water vapor from the Picarro (y-pic) and the fit using the ECMWF-IFS water vapor (y-ecmwf).

**[RC2.15] Systematic Uncertainties (flask moisture)-3**: The idea as I understand it is that in humid conditions the condensation of water in flasks, and absorption of CO2 into that water, presents a persistent negative bias in flask measurements. If that is the case, what is the meaning of negative Picarro-PFP values? Is that indicative of mismatch between air 'parcels' (I don't think I want to call them air masses) being sampled? Or is that an indication of uncertainty in the evaluation?

*[AR2.15]: We attribute the negative values of Picarro-PFP values to the uncertainty in the evaluation. We discard the possibility of being a mismatch in different air parcels sampled because of the collocated sampling lines and the tests we have done with different averaging times with the Picarro data (Figure A3). Note that these negative values are more frequent at low water vapor concentrations, which occur more often in the free troposphere (Figure A4). We have added part of this response to Section 2.1.5:* *Note that the negative values in Figure A7 are due to the variable nature of the atmosphere causing uncertainty in both measurement strategies. We discard the possibility of being a mismatch in different air parcels sampled because of the collocated sampling lines and the*

*tests we have done with different averaging times with the Picarro data (Figure A3). Note that these negative values are more frequent at low water vapor concentrations, which occur more often in the free troposphere (Figures A4 and A5).*

**[RC2.16] Systematic Uncertainties (flask moisture)-4**: I have not seen a time series of the Picarro CO2 samples; is the CO2 concentration fairly continuous, or does it exhibit large excursions in value over short times and distances? What about water vapor? Are there small spatiotemporal variations in humidity associated with convection, rain, updrafts/downdrafts, or is the dominant variability that associated with wet and dry seasons? Does the scale and amplitude of CO2/H2O variability have any bearing on this problem?

*[AR2.16]: We have added a plot with the 12 vertical profiles used in the Picarro-Flask comparison. See new Figure A4. In addition, as stated in line 250, the 30 s averages in the Picarro data were filtered out if the std was above 0.5 ppm. With this we are taking care of the possible excursions in value over the time scale in which we compared to the PFP. As for the $CO_2/H_2O$ variability, for each single flight the covariance of $CO_2$ and $H_20$ is mainly driven by altitude and convection, but when averaging using multiple flights on seasonal time-scales the dry and wet season play a more important role.*

**[RC2.17] Systematic Uncertainties (flask moisture)-5**: The hypothesis as I understand it seems to be that the flask bias is due to a greater likelihood of CO2 absorption and low measure CO2 value as H2O mole fraction (humidity) increases. The error in the ECMWF prediction will be due to errors in transport, surface CO2 flux, and advection of long-range signal. Does higher ECMWF error with higher humidity suggest errors in cumulus transport in ECMWF? Or does it stratify along wet/dry season lines? Either way, does this suggest a 'right answer for the wrong reason' solution?

*[AR2.17]: The flask is biased because at the time of sampling the air is not dried. From ECMWF-IFS we only take the water vapor concentration, there is no simulated $CO_2$ value in our bias correction. We explicitly use the Picarro - PFP in conjunction with the water vapor concentration obtained from ECMWF-IFS-STILT, so there are no errors associated with $CO_2$ fluxes as the reviewer suggests. Errors in vertical transport in ECMWF-IFS will indeed propagate to the simulated water vapor, this is probably the reason why ECWMF-IFS-STILT underestimates the water vapor below 4000 m (Figure A4). However, we are fitting the Picarro-PFP bias to the STILT water vapor and based on that relationship we derive the biases at the other sites.*

**[RC2.18] Priors, and Inversion Results-1**: OK, I don't think I understood, after reading the paper multiple times, what the benefit of using 10 priors brought to the analysis. If there is a compelling reason, other than to show that the authors did a lot of work, it escaped me. In that case the authors need to do a better job of making the reason for including them all clear to the reader. Here's what I got out of this section: The uncertainty reduction is largest (Figure 2B) in the areas with the most observational constraint on surface influence (Figure 1B). Well, yeah.

*[AR2.18]: This result might seem obvious for the reviewer, but let us explain why this is important for the community using the same observational network used in this study. The conclusions drawn from this dataset have less uncertainty in the east Amazon than in the west as we show in the manuscript. Therefore, attributing fluxes to the west with this dataset could be problematic if its done without acknowledging this limitation and quantifying the uncertainty of those fluxes. Therefore, being clear about the regions more sensitive to this observational network is important and we consider this as an important aspect of our study. Furthermore, priors have been shown to influence posterior fluxes in many previous studies, especially when data constraints are weak or absent. For this study region in particular, even the prior knowledge on fluxes is poor and representing it with only one mean value would not do justice to the uncertainties in for example TRENDY models, or CMIP6 fluxes. The 10 inversions create an ensemble with different assumptions on prior fluxes, their uncertainty and the weight relative to sparse observations. In that way, it goes beyond published flux estimates [Gatti et al., 2021, 2023] and follows more recent work [Basso et al., 2023]. Finally, the posterior spread*

*in an inversion using an ensemble of priors gives additional confidence on the posterior uncertainty derived from statistical methods.*

**[RC2.19] Priors, and Inversion Results-2**: The between-model spread in the prior mean NLF is really just an indicator of model 'responsiveness' and bias (Gallup et al., 2021; Hoffman et al., 2014). It seems to me that the value of the optimized flux (and therefore the slope of the line) is more an indication of model-data mismatch than anything else. Is that true? The way I think about it is that every inversion has tension (Scott Denning always called them 'rubber bands') between the optimized flux and the prior, and between the optimized flux and the observations. A strong coupling to the prior means that the optimized flux won't stray far from it, and a strong coupling to the observations means that it can. Some inversion systems are tightly coupled to the prior, and some can 'move away' from the prior more freely. In this case, I might think what is being shown in Figure 3 is more indicative of the CarboScope setup (model-data mismatch) than it is of any physical processes.

*[**AR2.19**]: The value of the optimized flux will be modulated by two things: 1. the assumed prior uncertainty (and it's structure) and 2. by the model-data mismatch (and by how the covariance matrix is defined). The analysis associated with Figure 3 (now Figure 4 in revised manuscript) aims to answer the following question: Given the assumed prior uncertainty and the same model-data mismatch, what is the dependence of the optimized flux in the magnitude of the selected prior flux. With this analysis we show that the selection of the prior will have a different effect across the spatial domain. We do not target physical processes, but the skill of the Inversion to adjust the prior. So it is indeed specific to the CarboScope setup, but it informs where a dependence on the prior magnitude could be stronger. We agree with the nice analogy of a rubber band presented by the reviewer. However, we note that in our simulations the model-data mismatch was the same for all inversions. So, the slope shown shows the sensitivity to structural differences (such as spatial patterns and diurnal cycles) in the prior fluxes used. This sensitivity indeed is system dependent, as the reviewer remarks, but not less valuable to quantify and understand in our opinion.*

**[RC2.20] Priors, and Inversion Results-3**: VPRMflat, If I remove the long-term mean, the seasonality, and the IAV from a flux product, what is left? The secular trend?

*[**AR2.20**]: The diurnal cycle is left, we have modified the text to make this more clear. We modified Table 1 and added:* *Two additional sensitivity tests were performed using the original VPRM. In one, we removed the long-term mean, seasonality, and interannual variability (IAV) from VPRM (called VPRM$_{flat}$) and run the inversion only with a diurnal cycle in the prior.*

**[RC2.21] Priors, and Inversion Results-4**: What is the reason for changing the sign of priors (0.5X and 1.0X priors)? Is this because CarboScope can't move the optimized flux very far away from the prior, so you move it artificially? Couldn't the same result (much different optimized flux) be achieved by relaxing the model-data mismatch?

*[**AR2.21**]: Compensating the effect of a large prior uncertainty (causing the VPRM and FLUX-COM/X priors to be outliers) with a change in the model-data mismatch uncertainty (which is by definition independent of the prior uncertainty) for two out of 8 priors is not the appropriate approach. The next lines are the response to a similar comment made by Reviewer 1, which motivate the 1x and 0.5 experiments: The reason that motivates the 1x and 0.5x scaling of NEE in VPRM and SiB4 is the following. The total land flux of the Amazon region is highly uncertain, in Table 3 of the manuscript we present estimates from recent studies, where the net land flux spans from -0.34 to 0.29 PgC year$^{-1}$ (Rosan et al., 2024, Gatti et al., 2021). This range of 0.62 PgC becomes larger than 1 PgC when considering on each end the uncertainties associated with each estimate. Therefore, given such a large range for the total land flux in the Amazon, we decided to keep the eddy-flux based prior NEE products (VPRM and FLUXCOM), as they can be considered as a plausible first guess in an inversion. To account for the positive part of the uncertainty range we then decided to make an experiment scaling VPRM and SiB4 such that NEE = 0.5 and 1 PgC. Modifying the model-data mismatch or relaxing the assumed prior uncertainty does not have the same meaning as starting from*

*a different first guess prior. We added a modified part of this reply to the revised manuscript:Note that the total land flux in the Amazon is highly uncertain, spanning from -0.34 to 0.29 PgC year$^{-1}$ [Gatti et al., 2021, Rosan et al., 2024], but this range gets larger than 1 PgC considering the uncertainties associated with each estimate, thus we decided to keep the eddy-flux based prior NEE products (VPRM and FLUXCOM), as they can be considered as a plausible first guess in an inversion. Furthermore and regardless of how they compare to current independent estimates we proceeded to make an experiment scaling two of our priors (i.e. VPRM and SiB4) such that NEE = 0.5 and 1 PgC, and thus we can test an opposing (in sign) prior scenario. To achieve this, we scaled ecosystem respiration in VPRM and SiB4 such that the total NEE integral for the biogeographic Amazon equals 0.5 PgC year$^{-1}$ and 1 PgC year$^{-1}$ (namely VPRM-0.5Pg, VPRM-1Pg, SIB4-0.5Pg, and SIB4-1Pg). An example for VPRM-0.5Pg is shown in Figure A9.*

**1  Specific Comments**

:

**[RC2.22]**: Check all citations: some citations that should be in parentheses are inline.

*[AR2.22]: All citations were corrected, thanks for pointing it out.*

**[RC2.23]** **Lines 84-86**: Incomplete sentence.

*[AR2.23]: Sentence was corrected.*

**[RC2.24]** **Line 187-188**: didn't the in situ analyzer also measure CO2?

*[AR2.24]: Yes, it is mentioned in Line 184.*

**[RC2.25]** **Lines 353-361**: Are these values even large enough to worry about? Are they meaningful?

*[AR2.25]: We refer the reviewer to answer AR1.22 for a thorough response on this matter to Reviewer 1. The values are significant and meaningful for the Cerrado and Caatinga region as the posterior uncertainty does not overlap with the posterior uncertainty of the inversion run without SAM stations. This is not so clear for the Amazon, but the posterior uncertainty is reduced as well in this region. To shorten the manuscript we have removed this part and kep the main message in Section 3.2, given that this was easier to communicate in this Section. We have added the following in that section and removed lines 353-361 from the revised manuscript. The addition reads:The atmospheric inversion allocates a net carbon source ($F_{NetLand}$) to the 'Cerrado & Caatinga' region (except for VPRM, VPRM-0.5Pg and VPRM-1Pg) and consistently identifies a net carbon sink in the biogeographic Amazon (Figure 5 and see Figures A9 and A10 for the spatial patterns). Interestingly, the addition of South American stations amplify this pattern and reduce the posterior uncertainty (compare s10 with s10sam). Despite the considerable variability in magnitude for the biogeographic Amazon, the atmospheric constraint tends to adjust priors with a positive sign, shifting them towards a smaller source (e.g., SIB4-1Pg) or even turning them into a sink with a negative $F_{NetLand}$ (e.g., VPRM-1Pg, VPRM-0.5Pg, and SIB4-0.5Pg). Furthermore, in the global inversion when adding South American data, the resulting fluxes closely follow the sign and magnitude of those obtained in the regional inversion. Therefore, we contend that information suggesting a sink-source gradient between the Amazon and the 'Cerrado & Caatinga' is embedded in the atmospheric measurement record, even with the inherent limitations in adjusting certain individual priors.*

**[RC2.26]** **Lines 376-378**: Does this mean that a prior flux of 0 would work as well as any of the other priors used?

*[AR2.26]: It means that a prior flux having zero-mean, but a diurnal cycle, would be closer to the main cluster of posterior estimates. We have added this more clearly in that part of the Results. reading now like: Nevertheless, the removal of the long-term mean of the prior flux, as implemented in $VPRM_{flat}$, results in a $F_{NetLand}$ that aligns more closely with the main cluster of posterior estimates at ≈-0.2 PgC year$^{-1}$ for the Biogeographic Amazon and at ≈0.3 PgC year$^{-1}$ for the 'Cerrado &*

*Caatinga'. In other words, a prior flux having zero-mean, but a diurnal cycle would be closer to the main cluster of posterior estimates.*

[**RC2.27**] **Lines 468-469**: How does the issue or prior selection convolve with the water vapor issue?

*[**AR2.27**]: The effect of the water vapor bias is independent of the prior selection. This means that regardless of the selected prior, the magnitude shift in posterior estimates due to the bias correction will be the same.*

**References**

Bianca C. Baier, Colm Sweeney, Yonghoon Choi, Kenneth J. Davis, Joshua P. DiGangi, Sha Feng, Alan Fried, Hannah Halliday, Jack Higgs, Thomas Lauvaux, Benjamin R. Miller, Stephen A. Montzka, Timothy Newberger, John B. Nowak, Prabir Patra, Dirk Richter, James Walega, and Petter Weibring. Multispecies Assessment of Factors Influencing Regional CO2 and CH4 Enhancements During the Winter 2017 ACT-America Campaign. *Journal of Geophysical Research: Atmospheres*, 125(2), 2020. ISSN 2169-8996. doi: 10.1029/2019JD031339.

Luana S. Basso, Chris Wilson, Martyn P. Chipperfield, Graciela Tejada, Henrique L. G. Cassol, Egídio Arai, Mathew Williams, T. Luke Smallman, Wouter Peters, Stijn Naus, John B. Miller, and Manuel Gloor. Atmospheric $CO_2$ inversion reveals the Amazon as a minor carbon source caused by fire emissions, with forest uptake offsetting about half of these emissions. *Atmospheric Chemistry and Physics*, 23(17):9685–9723, September 2023. ISSN 1680-7316. doi: 10.5194/acp-23-9685-2023.

Santiago Botía, Shujiro Komiya, Julia Marshall, Thomas Koch, Michał Gałkowski, Jost Lavric, Eliane Gomes-Alves, David Walter, Gilberto Fisch, Davieliton M. Pinho, Bruce W. Nelson, Giordane Martins, Ingrid T. Luijkx, Gerbrand Koren, Liesbeth Florentie, Alessandro Carioca de Araújo, Marta Sá, Meinrat O. Andreae, Martin Heimann, Wouter Peters, and Christoph Gerbig. The CO2 record at the Amazon Tall Tower Observatory: A new opportunity to study processes on seasonal and inter-annual scales. *Global Change Biology*, 28(2):588–611, 2022. ISSN 1365-2486. doi: 10.1111/gcb.15905.

Sean Crowell, David Baker, Andrew Schuh, Sourish Basu, Andrew R. Jacobson, Frederic Chevallier, Junjie Liu, Feng Deng, Liang Feng, Kathryn McKain, Abhishek Chatterjee, John B. Miller, Britton B. Stephens, Annmarie Eldering, David Crisp, David Schimel, Ray Nassar, Christopher W. O'Dell, Tomohiro Oda, Colm Sweeney, Paul I. Palmer, and Dylan B. A. Jones. The 2015–2016 carbon cycle as seen from OCO-2 and the global in situ network. *Atmospheric Chemistry and Physics*, 19(15):9797–9831, August 2019. ISSN 1680-7316. doi: https://doi.org/10.5194/acp-19-9797-2019. Publisher: Copernicus GmbH.

C. Frankenberg, Y. M. Bar-On, Y. Yin, P. O. Wennberg, D. J. Jacob, and A. M. Michalak. Data Drought in the Humid Tropics: How to Overcome the Cloud Barrier in Greenhouse Gas Remote Sensing. *Geophysical Research Letters*, 51(8):e2024GL108791, 2024. ISSN 1944-8007. doi: 10.1029/2024GL108791. URL `https://onlinelibrary.wiley.com/doi/abs/10.1029/2024GL108791`. _eprint: https://onlinelibrary.wiley.com/doi/pdf/10.1029/2024GL108791.

Luciana V. Gatti, Luana S. Basso, John B. Miller, Manuel Gloor, Lucas Gatti Domingues, Henrique L. G. Cassol, Graciela Tejada, Luiz E. O. C. Aragão, Carlos Nobre, Wouter Peters, Luciano Marani, Egidio Arai, Alber H. Sanches, Sergio M. Corrêa, Liana Anderson, Celso Von Randow, Caio S. C. Correia, Stephane P. Crispim, and Raiane A. L. Neves. Amazonia as a carbon source linked to deforestation and climate change. *Nature*, 595(7867):388–393, July 2021. ISSN 1476-4687. doi: 10.1038/s41586-021-03629-6.

Luciana V. Gatti, Camilla L. Cunha, Luciano Marani, Henrique L. G. Cassol, Cassiano Gustavo Messias, Egidio Arai, A. Scott Denning, Luciana S. Soler, Claudio Almeida, Alberto Setzer, Lucas Gatti Domingues, Luana S. Basso, John B. Miller, Manuel Gloor, Caio S. C. Correia, Graciela Tejada, Raiane A. L. Neves, Raoni Rajao, Felipe Nunes, Britaldo S. S. Filho, Jair Schmitt, Carlos Nobre, Sergio M. Corrêa, Alber H. Sanches, Luiz E. O. C. Aragão, Liana Anderson, Celso Von Randow, Stephane P. Crispim, Francine M. Silva, and Guilherme B. M. Machado. Increased Amazon carbon emissions mainly from decline in law enforcement. *Nature*, 621(7978):318–323, September 2023. ISSN 1476-4687. doi: 10.1038/s41586-023-06390-0.

M. Gloor, Chris Wilson, Martyn P. Chipperfield, Frederic Chevallier, Wolfgang Buermann, Hartmut Boesch, Robert Parker, Peter Somkuti, Luciana V. Gatti, Caio Correia, Lucas G. Domingues, Wouter Peters, John Miller, Merritt N. Deeter, and Martin J. P. Sullivan. Tropical land carbon cycle responses to 2015/16 El Niño as recorded by atmospheric greenhouse gas and remote sensing data. *Philosophical Transactions of the Royal Society B: Biological Sciences*, 373(1760):20170302, October 2018. doi: 10.1098/rstb.2017.0302. Publisher: Royal Society.

A. a. M. Holtslag and B. A. Boville. Local Versus Nonlocal Boundary-Layer Diffusion in a Global Climate Model. October 1993. ISSN 1520-0442. URL `https://journals.ametsoc.org/view/journals/clim/6/10/1520-0442_1993_006_1825_lvnbld_2_0_co_2.xml`. Section: Journal of Climate.

Panagiotis Kountouris, Christoph Gerbig, Christian Rödenbeck, Ute Karstens, Thomas F. Koch, and Martin Heimann. Atmospheric $CO_2$ inversions on the mesoscale using data-driven prior uncertainties: quantification of the European terrestrial $CO_2$ fluxes. *Atmospheric Chemistry and Physics*, 18 (4):3047–3064, March 2018. ISSN 1680-7316. doi: https://doi.org/10.5194/acp-18-3047-2018.

M Krol, S Houweling, B Bregman, and P Bergamaschi. The two-way nested global chemistry-transport zoom model TM5: algorithm and applications. *Atmos. Chem. Phys.*, page 16, 2005.

Junjie Liu, Kevin W. Bowman, David S. Schimel, Nicolas C. Parazoo, Zhe Jiang, Meemong Lee, A. Anthony Bloom, Debra Wunch, Christian Frankenberg, Ying Sun, Christopher W. O'Dell, Kevin R. Gurney, Dimitris Menemenlis, Michelle Gierach, David Crisp, and Annmarie Eldering. Contrasting carbon cycle responses of the tropical continents to the 2015–2016 El Niño. *Science*, 358(6360), October 2017. ISSN 0036-8075, 1095-9203. doi: 10.1126/science.aam5690.

Jean-François Louis. A parametric model of vertical eddy fluxes in the atmosphere. *Boundary-Layer Meteorology*, 17(2):187–202, September 1979. ISSN 1573-1472. doi: 10.1007/BF00117978. URL `https://doi.org/10.1007/BF00117978`.

Steven T. Massie, K. Sebastian Schmidt, Annmarie Eldering, and David Crisp. Observational evidence of 3-D cloud effects in OCO-2 CO2 retrievals. *Journal of Geophysical Research: Atmospheres*, 122(13):7064–7085, 2017. ISSN 2169-8996. doi: 10.1002/2016JD026111. URL `https://onlinelibrary.wiley.com/doi/abs/10.1002/2016JD026111`. _eprint: https://onlinelibrary.wiley.com/doi/pdf/10.1002/2016JD026111.

Dipayan Paul, Hubertus A. Scheeren, Henk G. Jansen, Bert A. M. Kers, John B. Miller, Andrew M. Crotwell, Sylvia E. Michel, Luciana V. Gatti, Lucas G. Domingues, Caio S. C. Correia, Raiane A. L. Neves, Harro A. J. Meijer, and Wouter Peters. Evaluation of a field-deployable Nafion™-based air-drying system for collecting whole air samples and its application to stable isotope measurements of $CO_2$. *Atmospheric Measurement Techniques*, 13(7):4051–4064, July 2020. ISSN 1867-1381. doi: 10.5194/amt-13-4051-2020. Publisher: Copernicus GmbH.

Hélène Peiro, Sean Crowell, Andrew Schuh, David F. Baker, Chris O'Dell, Andrew R. Jacobson, Frédéric Chevallier, Junjie Liu, Annmarie Eldering, David Crisp, Feng Deng, Brad Weir, Sourish Basu, Matthew S. Johnson, Sajeev Philip, and Ian Baker. Four years of global carbon cycle observed from the Orbiting Carbon Observatory 2 (OCO-2) version 9 and in situ data and comparison to OCO-2 version 7. *Atmospheric Chemistry and Physics*, 22(2):1097–1130, January 2022. ISSN 1680-7316. doi: 10.5194/acp-22-1097-2022. URL `https://acp.copernicus.org/articles/22/1097/2022/`. Publisher: Copernicus GmbH.

Thais M. Rosan, Stephen Sitch, Michael O'Sullivan, Luana S. Basso, Chris Wilson, Camila Silva, Emanuel Gloor, Dominic Fawcett, Viola Heinrich, Jefferson G. Souza, Francisco Gilney Silva Bezerra, Celso von Randow, Lina M. Mercado, Luciana Gatti, Andy Wiltshire, Pierre Friedlingstein, Julia Pongratz, Clemens Schwingshackl, Mathew Williams, Luke Smallman, Jürgen Knauer, Vivek Arora, Daniel Kennedy, Hanqin Tian, Wenping Yuan, Atul K. Jain, Stefanie Falk, Benjamin Poulter, Almut Arneth, Qing Sun, Sönke Zaehle, Anthony P. Walker, Etsushi Kato, Xu Yue, Ana Bastos, Philippe Ciais, Jean-Pierre Wigneron, Clement Albergel, and Luiz E. O. C. Aragão. Synthesis of the land carbon fluxes of the Amazon region between 2010 and 2020. *Communications Earth & Environment*, 5(1):1–15, January 2024. ISSN 2662-4435. doi: 10.1038/s43247-024-01205-0.

Gary L. Russell and Jean A. Lerner. A New Finite-Differencing Scheme for the Tracer Transport Equation. December 1981. ISSN 1520-0450. URL `https://journals.ametsoc.org/view/journals/apme/20/12/1520-0450_1981_020_1483_anfdsf_2_0_co_2.xml`. Section: Journal of Applied Meteorology and Climatology.

C Rödenbeck, C Gerbig, K Trusilova, and M Heimann. A two-step scheme for high-resolution regional atmospheric trace gas inversions based on independent models. *Atmos. Chem. Phys.*, page 12, 2009.

C. Rödenbeck, S. Zaehle, R. Keeling, and M. Heimann. History of El Niño impacts on the global carbon cycle 1957–2017: a quantification from atmospheric $CO_2$ data. *Philosophical Transactions of the Royal Society B: Biological Sciences*, 373(1760):20170303, November 2018. ISSN 0962-8436, 1471-2970. doi: 10.1098/rstb.2017.0303.

Christian Rödenbeck. Estimating CO2 sources and sinks from atmospheric mixing ratio measurements using a global inversion of atmospheric transport. Technical report, Max Planck Institute for Biogeochemistry, 2005. URL `https://www.bgc-jena.mpg.de/~christian.roedenbeck/download/2005-Roedenbeck-TechReport6.pdf`.

Andrew E. Schuh, Andrew R. Jacobson, Sourish Basu, Brad Weir, David Baker, Kevin Bowman, Frédéric Chevallier, Sean Crowell, Kenneth J. Davis, Feng Deng, Scott Denning, Liang Feng, Dylan Jones, Junjie Liu, and Paul I. Palmer. Quantifying the Impact of Atmospheric Transport Uncertainty on CO2 Surface Flux Estimates. *Global Biogeochemical Cycles*, 33(4):484–500, 2019. ISSN 1944-9224. doi: 10.1029/2018GB006086. URL `https://onlinelibrary.wiley.com/doi/abs/10.1029/2018GB006086`.

M. Tiedtke. A Comprehensive Mass Flux Scheme for Cumulus Parameterization in Large-Scale Models. *Monthly Weather Review*, 117(8):1779–1800, August 1989. ISSN 0027-0644. doi: 10.1175/1520-0493(1989)117⟨1779:ACMFSF⟩2.0.CO;2. Publisher: American Meteorological Society.

K Trusilova, C Rödenbeck, C Gerbig, and M Heimann. Technical Note: A new coupled system for global-to-regional downscaling of CO2 concentration estimation. *Atmos. Chem. Phys.*, page 9, 2010.

Jun Wang, Ning Zeng, Meirong Wang, Fei Jiang, Frédéric Chevallier, Sean Crowell, Wei He, Matthew S. Johnson, Junjie Liu, Zhiqiang Liu, Scot M. Miller, Sajeev Philip, Hengmao Wang, Mousong Wu, Weimin Ju, Shuzhuang Feng, and Mengwei Jia. Anomalous Net Biome Exchange Over Amazonian Rainforests Induced by the 2015/16 El Niño: Soil Dryness-Shaped Spatial Pattern but Temperature-dominated Total Flux. *Geophysical Research Letters*, 50(11):e2023GL103379, 2023. ISSN 1944-8007. doi: 10.1029/2023GL103379.

---

## Author Comment (AC2)

Reviews from Referee 1 - Received on September 19, 2024

The format in which the response is addressed is the following:

1. Black text shows comment of referee. Comments are numerated as RC1.1, for the first comment. The line in the submitted draft referred to by the referee is also shown.

2. *Text in italics show the author's response and it has the same logic for numeration (e.g. AR1.1). If not stated otherwise, here the reference to figures and line numbers are based on the submitted manuscript.*

3. Red text indicates changes to the manuscript. Here the reference to Figures are based on the revised manuscript.

**General Comment**

[**RC1.1**] This comprehensive study uses dry air CO2 mole fractions measured at the Amazon Tall Tower Observatory (ATTO) and airborne vertical CO2 profiles in the CarboScope Regional atmospheric inversion system to estimate the net carbon exchange in tropical South America. They find that the biogeographic Amazon is a net carbon sink after accounting for vegetation fluxes, river efflux, and fire emissions. They note that treatment of Cerrado and Caatinga biomes in previous analyses have been historically lacking and include these biomes specifically, They further note that Cerrado and Caatinga biomes roughly offset the net uptake making the entirety of tropical South America close to neutral. The paper also addresses the role of measurement uncertainties on their results , namely water vapor corrections to aircraft profiles and low representation of measurements in the Amazon-Andes foothills. Overall, this is an important study spotlighting a key region (from both climate and ecological angles) that is underrepresented in existing carbon cycle model/measurement studies. I recommend the following minor revisions.

*[**AR1.1**] We thank the reviewer for the comments and proceed to answer the specific comments.*

**Specific Comments**

[**RC1.2**] **Title**: Combined CO2 measurement record indicates decreased Amazon forest carbon uptake, offset by Savannah carbon release". I find the title confusing – consider rewording. Decreasing Amazon forest uptake implies that the Savanna carbon release is doing the opposite of offsetting. That is, a weaker uptake signal in the Amazon forest combined with an increasing Savanna release means an \*amplified\* overall release rather than a counterbalance/offset. Perhaps you mean to say the Amazon Forest C uptake signal is diminishing in its capacity to offset the Savanna C release?

**Minor suggestion** – Savanna is far more common as spelling; I recommend changing from Savannah to Savanna; fix throughout (including title).

*[AR1.2]: Yes, indeed, the confusion stems from the use of "decreased" to describe the Amazon forest carbon uptake. We have modified the title to: Combined $CO_2$ measurement record indicates Amazon forest carbon uptake is offset by Savanna carbon release. We have also replaced Savannah with Savanna throughout the manuscript.*

**[RC1.3] Abstract**: Specify the study period (2010-2018) in the abstract for clarity also.

*[AR1.3]: The study period has been added to the abstract.*

**[RC1.4] Abstract**: uncertainties – specify whether SD, or 95%CI, or other.

*[AR1.4]: The uncertainties reported correspond to the posterior uncertainty. The abstract was modified accordingly.*

**[RC1.5] L30**: a little weak, fails to highlight main points of paper expand on this. Bring more attention to the results summarized in L423-427 re: impact of water vapor correction and L533-537. In addition, there is an information mismatch with the title– the overall results suggest a biogeographic amazon sink (abstract L7), so if you also mean to say there is a diminished sink trend from 2010-2018, make that clear in the abstract. However, this "diminished sink trend" requires further analysis – and meanwhile Figure 6a and Figure A15 don't seem to suggest a strong trend from 2010-2018 for the Amazon as a whole or by region.

*[AR1.5]: We have updated the abstract to include the suggestions from the referee. However, note that we do not report a trend, the diminished sink we refer to, corresponds to the effect of the water vapor correction on the magnitude of the posterior fluxes, but not to a trend. Throughout the manuscript, the word "trend" is only used in the Introduction in another context.*

**[RC1.6] L30**: Assis et al. (2020) has parentheses in the wrong place, also throughout, check the formatting of citations and incorrect placement of parentheses as this happens in multiple locations (e.g., L112, L152, L153...).

*[AR1.6]: Thanks for the suggestion, we have fixed the citation format throughout the manuscript where applicable.*

**[RC1.7] L34**: Inconsistency with L21 (230 PgC). Use a consistent estimate; perhaps in both places indicate 150-230 rather than 150-200.

*[AR1.7]: The reference to 230 PgC is for tropical ecosystems, not only the Amazon. But, we acknowledge that this confusion can arise and thus have removed the reference to 230 PgC. The modification reads as follows:* Furthermore, tropical ecosystems store substantial carbon reserves in aboveground ground living biomass (Brando et al., 2019), that can be released rapidly further amplifying the $CO_2$ growth rate.

**[RC1.8] L63**: vegetation-related source

*[AR1.8]: Corrected.*

**[RC1.9] L83** Do you mean 'With this we [conclude]'?

*[AR1.9]: With the use of "With this" we were referring to the previous sentences, where we*

*describe the main contributions of the manuscript. The last paragraph of the Introduction now reads as:* In this study, we use the CarboScope Global and Regional inversion system to assimilate the 2010-2018 airborne $CO_2$ profile record and the continuous and long-term $CO_2$ record at the Amazon Tall Tower Observatory (ATTO). We build on previous studies using the CarboScope Regional system in Europe [Kountouris et al., 2018b,a, Munassar et al., 2021], to explore its ability to constrain the $F_{NetLand}$ at the continental scale over a larger domain, but with a sparser observational network. The study is structured as follows. First, we aim to quantify where the atmospheric inversion using this set of atmospheric data can provide a constraint based on uncertainty reduction. Second, a sub-continental analysis of the carbon budget, with a strong focus on the biogeographic Amazon, but not limited to it, is performed to shed light on spatial gradients. Last, we present a detailed quantification of how systematic uncertainties in measured mole fractions in the aircraft network affect the estimated fluxes in an atmospheric inversion. With this study, we provide a broad perspective on carbon exchange in tropical South America, going beyond the Amazon biome and highlighting where do we need to expand our observational efforts to reduce the uncertainty in carbon exchange estimates in the region.

**[RC1.10] L84**: Sentence fragment; join to previous sentence.

*[AR1.10]: We have removed the sentence starting with "An aspect that..". We decided to finish the introduction without referring to RECCAP2, as at the moment the RECCAP2 synthesis for South America has not been released yet.*

**[RC1.11] L86**: You bring up RECCAP2, but only once in the introduction. Consider expanding more on contributions in the conclusions.

*[AR1.11]: We removed the reference to RECCAP2, see AR1.10 and AR1.9.*

**[RC1.12] L100**: "for all sites are obtained" – you mean all the aircraft sites + MAN from Figure 1b?

*[AR1.12]: Here we refer to all the global sites that are used in Step 1 of the two step scheme. This will include aircraft sites, but not MAN, as this site was left out of the inversion for evaluation. We have made this clearer by adding:* Using that optimized NBE flux field and the same atmospheric transport set-up, simulated mole fractions increments for all sites are obtained, except the site left for validation (i.e. the s10 station set plus the South American stations, see Sect 2.1.4).

**[RC1.13] Methodology Sec 2.1.6**: What eddy flux sites are you using to calibrate VPRM? Provide in appendix, and include in acknowledgments.

*[AR1.13]: We have added a list with all the eddy flux sites that were used for calibrating VPRM. See Table A2 (also here for easy access), the reference to it and a citation of the data in Sect 2.1.6.*

**[RC1.14] L153**: This is confusing. If you are using the ATTO+NAT+Aircraft for the global inversion whose posterior is then the prior for the regional inversion (L99) you are using the ATTO+NAT+Aircraft data twice. So it doesn't seem like this is giving you any new information (ie you're optimizing with the same constraints twice). Wouldn't it be better to, say, use ATTO as an NBE constraint for the step 1 process and then use NAT+Aircraft for step 2? However, I could just be misunderstanding as

Table 1: Eddy flux sites used to calibrate the VPRM parameters.

| Site code | Lat | Lon | Site Veg. Description | Veg. Class VPRM |
|---|---|---|---|---|
| STM-K67 | -2.85700 | -54.95900 | Primary Tropical Moist Forest | Evergreen Forest |
| STM-K77 | -3.02020 | -54.88850 | Pasture, then Agriculture | Cropland |
| STM-K83 | -3.01700 | -54.97070 | Primary Tropical Moist Forest,sel. logging Aug/Sept 2001 | Evergreen Forest |
| MAN-k34 | -2.50000 | -60.20910 | Tropical Rainforest | Evergreen Forest |
| PA-CAX | -1.74830 | -51.45360 | Tropical Forest, dense lowland tropical forest | Evergreen Forest |
| RON-FNS | -10.76180 | -62.35720 | Pasture | Grassland |
| RON-RJA | -10.07800 | -61.93310 | Tropical Dry Forest | Evergreen Forest |
| TOC-BAN | -9.824416667 | -50.15911 | Seasonally flooded Forest-Savanna Ecotone | Evergreen Forest |
| SP-PDG | -21.61947222 | -47.64989 | Savanna | Savannas |

Table 1 suggests that all the subsequent prior fluxes for the regional inversion are based off the s10 (rather than s10sam). Can you clarify all this please?

*[**AR1.14**]: In Step 1 of the 2-step scheme we use the default station (i.e. s10) set from CarboScope global, and to that we add ATTO+NAT+Aircraft. However, the posterior of that global run is **not** used as the prior of the regional inversion. As stated in Sect 2.1.1, the NBE flux field resulting from the global inversion is used to obtain the mole fraction increments in two sequential runs. The first one is done at the coarse global resolution for all sites globally and the second, only for the regional domain. The difference between these runs corresponds to the far-field contribution, which is then subtracted from the measurements and used as data constraint in the high-resolution regional inversion. The priors used in each of the regional inversions are then independent from the global inversion. We have added a flow diagram in Section 2.1 to complement the explanation of the CarboScope Regional system. We refer the reviewer to our response to Reviewer 2 (AR2.11), where we answered a similar question and the diagram is shown.*

[**RC1.15**] **L174**: Expand – Provide citation/justification for assumption.

*[**AR1.15**]: The assumption of weekly error correlations in model-data mismatch can be justified by the following. Continuous data have always been treated that way in CarboScope, originally this was referred to as "data density weighting" introduced in Rödenbeck [2005]. For the regional inversions with CarboScope-Regional, the justification of model-data mismatch error correlation has been introduced in Kountouris et al. [2018a]. So even though the motivation for the data density weighting is not the same Europe as in South America, we nevertheless need to use it to allow for a combined use of flask and continuous data in the inversion. We have modified Section 2.1.4 as follows: The model-data mismatch uncertainty (including the representation error of the measurements within the transport model) for the three types of sites (in-situ tower, aircraft, and weekly flasks) is chosen to be 1.5 ppm for weekly time scales, following common practice in CarboScope global [Rödenbeck, 2005, Rödenbeck et al., 2018] which assimilates a large set of weekly flask samples. To assimilate multiple data streams, we apply a data density weighting Rödenbeck [2005]: For the hourly ATTO data, the error will be inflated by $\sqrt{N_{hours/week}}$ (details see Kountouris et al. [2018a]), while for aircraft profiles (composed of several*

*flasks) the error is scaled with $\sqrt{N_{flasks/profile}}$. The data-density weighting practically ensures that one week of hourly ATTO observations, one aircraft profile, or one weekly flask sample have the same weight in the inversion, reflecting the assumption that they provide the same amount of information due to roughly weekly error correlations.*

**[RC1.16] L250-L251**: What is the physical basis for selecting a 0.5x scaling factor for VPRM and SiB4? Related, Figure A8 – the 0.5VPRM Prior shows a change in sign as well in large portions of the Amazon indicating you applied a scaling factor of >1 to the respiration which had a net result of 0.5xVPRMNEE correct? What was the exact respiration scaling factor? Did you apply that SF because you expected respiration to have been underestimated by VPRM and SiB4 (rather than GPP to be overestimated)? If so, why?

*[AR1.16]: The authors thank the reviewer for pointing this out. The reason that motivates the 1x and 0.5x scaling of NEE in VPRM and SiB4 is the following. The total land flux of the Amazon region is highly uncertain, in Table 3 of the manuscript we present estimates from recent studies, where the net land flux spans from -0.34 to 0.29 PgC year$^{-1}$ [Gatti et al., 2021, Rosan et al., 2024]. This range of 0.62 PgC becomes larger than 1 PgC when considering on each end the uncertainties associated with each estimate. Therefore, given such a large range for the total land flux in the Amazon, we decided to keep the eddy-flux based prior NEE products (VPRM and FLUXCOM), as they can be considered as a plausible first guess in an inversion. To account for the positive part of the uncertainty range we then decided to make an experiment scaling VPRM and SiB4 such that NEE = 0.5 and 1 PgC. The way in which we did the scaling was via the Respiration, which is the source component in NEE and for VPRM it is more uncertain due to the very simple parameterization as a linear function of temperature. The scaling factor for SiB4, where NEE = TER-GPP, was calculated as: SF = (NEE_{target} + GPP)/TER. Where, NEE_{target} can be 0.5 or 1 PgC for the long-term mean over the entire period of interest. With that SF factor TER was scaled for each year individually such that the long-term mean of the scaled NEE = NEE_{target}. For VPRM a similar procedure was applied but considering NEE = -GEE + TER. The exact scaling factors for SiB4 and VPRM are: 0.5xVPRM = 1.197, 1xVPRM = 1.229, 0.5xSiB4 = 1.031, 1xSiB4 = 1.059. We have added part of this description and justification to the revised manuscript and it reads as follows: Originally the eddy-covariance-based products, like the two FLUXCOM versions and VPRM, have a large sink magnitude for the Amazon. Note that the total land flux in the Amazon is highly uncertain, spanning from -0.34 to 0.29 PgC year$^{-1}$ [Gatti et al., 2021, Rosan et al., 2024], but this range gets larger than 1 PgC considering the uncertainties associated with each estimate, thus we decided to keep the eddy-flux based prior NEE products (VPRM and FLUXCOM), as they can be considered as a plausible first guess in an inversion. Furthermore and regardless of how they compare to current independent estimates we proceeded to make an experiment scaling two of our priors (i.e. VPRM and SiB4) such that NEE = 0.5 and 1 PgC, and thus we can test an opposing (in sign) prior scenario. To achieve this, we scaled ecosystem respiration in VPRM and SiB4 such that the total NEE integral for the biogeographic Amazon equals 0.5 PgC year$^{-1}$ and 1 PgC year$^{-1}$ (namely VPRM-0.5Pg, VPRM-1Pg, SIB4-0.5Pg,*

*and SIB4-1Pg). An example for VPRM-0.5Pg is shown in Figure A9. Two additional sensitivity tests were performed using the original VPRM. In one, we removed the long-term mean, seasonality, and interannual variability (IAV) from VPRM (called VPRM$_{flat}$) and run the inversion only with a diurnal cycle in the prior. In the second one, we used VPRM as prior but left the ATTO data out from the assimilated station set (called VPRMnoATT).*

[**RC1.17**] **L268**: Expand on this methodology and discuss drawbacks/limitations. You are assuming the same scaling factor for COprior/COpost as for CO2? As CO/CO2 relates to combustion efficiency, it's possible that the GFAS CO prior relates to the CO posterior in a way that is not necessarily mirrored in a CO2prior/CO2post relationship. What if instead you do a biome-specific COprior/COpost factor (with uncertainties) – that way you get a sense of combustion efficiency across a specific biome that integrates dominant plant functional types and accounts for the average expectation of CO / CO2 combustion efficiencies?

*[**AR1.17**]: We thank the reviewer for this suggestion. However, we believe that doing a biome-specific inversion is out of the scope of this study. For the CO-MOPITT inversion we refer the reviewer to Naus et al., 2022, where it was shown that GFAS is biased low for CO. Assuming that the CO/CO$_2$-ratios in GFAS are accurate, we increase the CO$_2$ fire emission estimates. We do acknowledge that this is an assumption, and state this more clearly in the manuscript now:* Thus, our approach assumes that the adjustment of the MOPITT-Inversion in CO is also applicable to CO$_2$. We acknowledge that this is an approximation, as the emission ratio between CO and CO$_2$ could also be off in GFAS. However, here we assume they are constant and interpret the underestimation in CO as an underestimation in fire emissions. This is in line with recent studies of undetected African fire emissions [Ramo et al., 2021].

[**RC1.18**] **L300**: "consistent with predominant air transport..." Can you re-phrase and/or clarify this statement? You seem to be correlating observational density and air transport which is confusing.

*[**AR1.18**]: We meant to say that the observational constraint and thus where the uncertainty reduction is the largest, is where the predominant air is coming from. We realized that this can be confusing and removed that first part of the sentence.*

[**RC1.19**] **L319**: Clarify – is the mean impact 2% or >5%? Do you mean to say that the maximum UR can be >5 %?

*[**AR1.19**]: The mean impact over 2010-2018 is 2% for all the biogeographic Amazon. But for individual years it can be larger than 5%. We have modified the sentence to:* At the biogeographic Amazon scale, the mean impact on the UR is small (2%), but in individual years it can increase to more than 5%.

[**RC1.20**] **L250-251; L338**: The VPRM (and SiB4) scaling nomenclature is confusing. 1xVPRM seems like it should be just VPRM (mathematically) but I think what you are trying to say is that 1xVPRM is VPRM constrained by respiration to have a total NEE of 1PgCy-1. If this is the case, then can you change the nomenclature? Something like VPRM, VPRM-0.5Pg, VPRM-1.0Pg.

*[**AR1.20**]: Thanks for the suggestion. We have modified the nomenclature throughout the manuscript*

*and Figures to avoid this confusion.*

[RC1.21] **L335, Fig 3**: I don't quite follow what your main message is with these results. Are you just trying to show the spread and/or convergence among flux model ensemble members, and break that down regionally? i.e., with the Cerrado & Caatinga region, you are showing that there are two families of ensemble members, namely the VPRM family that suggests the cerrado & caatinga region have net uptake in prior and post (-0.4gCm-2d-1), and all other models showing neutral to net release at least in their posteriors. Can you clarify your main message here? Related, in the next paragraph you are stating the superior constraint by atmospheric data for Figs 3c-f – is this statement based on the change in the posterior distributions (ie approaching normal/biggest reduction in uncertainty) on the y axes? If so, the same can be said for Cerrado and Caatinga region but the issue there is that the VPRM family seems to be driving the bimodality in Fig 3b y and x axis. Again, clarify main message as it seems to be getting a little lost. (Your main message seems to be L350-352.)

*[AR1.21]: We thank the reviewer for pointing this out. The main message here is to show in which region there is more influence of the prior on the posterior. First, we focus on the large regions, the Amazon and the Cerrado & Caatinga. Then we focus on the smaller regions inside the Amazon. To clarify the main message of this analysis, we have modified the text from line 335 to 352 in the submitted manuscript. The revised manuscript is now more concrete, reading as follows:* *For the biogeographic Amazon and the 'Cerrado & Caatinga' we find a strong linear dependence of the posterior estimates on the prior (Figure 4a,b). Even though the spread in the marginal distribution is reduced largely from prior to posterior, the models with a large uptake in the prior (e.g. VPRM, FLUXCOM, X-BASE$_{NEE}$) do not converge with the main cluster of posterior estimates. We further evaluated such dependence with the VPRMflat experiment, confirming that after removing the long-term mean, the VPRMflat posterior $F_{NetLand}$ falls closer to the main group of estimates in both regions. Interestingly, the regions in the eastern part of the Amazon ('Amazon River Flat Plains', 'Brazilian Shield Moist Forests', and the 'Guianan Shield Moist Forests') exhibit superior constraint by atmospheric data, as illustrated in Figure 4c-f. The spread in the posterior marginal distribution and the slopes of the linear regression in these four regions are inversely proportional to their respective reduction in uncertainty (Figure 3). This inverse relationship indicates that the posterior estimates are more effectively adjusted, irrespective of the prior magnitude, in regions with a higher reduction in uncertainty. Therefore, the 'Amazon and Andes Piedmont' in the west stands out as an area where a bias in the prior fluxes would exert a more substantial impact on posterior estimates.*

[RC1.22] **L353**: How significant are these results? South American stations are ATTO+NAT+Aircraft? Somewhere early on state that ATTO+NAT+Aircraft will continue to be referred to as South American Stations and keep that terminology consistent throughout. How does adding more stations (and having no impact on Andes Amazon Piedmont) reinforce lack of observational constraint in that region? Are you trying to say that the stations added are irrelevant to that region, and you need more stations *within* that region?

*[AR1.22]: The South American (SAM) stations are indeed ATTO+NAT+Aircraft. The effect of*

*adding stations in South America was assessed with the global inversion. We found that when adding the SAM stations the Amazon becomes a stronger sink and the Cerrado & Caatinga becomes a stronger source. At these scales, the results are significant for the Cerrado & Caatinga as the mean $F_{NetLand}$ with the posterior uncertainty in s10 does not overlap with that of s10sam. Furthermore, there is a reduction in posterior uncertainty from s10 to s10sam (see Figure 4 on submitted manuscript). When looking into the regional partitioning of this effect inside the Amazon, we found the smallest difference is in the 'Amazon and Andes Piedmont', where the observational constraint or coverage of the SAM station is the lowest. In other words, the station network used in SAM, has very little information content to constrain the fluxes in the 'Amazon and Andes Piedmont' as the difference between posterior fluxes with and without SAM stations is the lowest of all regions. So, the reviewer is right when stating that the SAM stations are "irrelevant" for that region. The lines 353 to 361 in the submitted manuscript were removed from the revised manuscript and a small section referring to the effect of adding SAM stations was added in Section 3.2:* The atmospheric inversion allocates a net carbon source ($F_{NetLand}$) to the 'Cerrado & Caatinga' region (except for VPRM, VPRM-0.5Pg and VPRM-1Pg) and consistently identifies a net carbon sink in the biogeographic Amazon (Figure 5 and see Figures A9 and A10 for the spatial patterns). Interestingly, the addition of South American stations amplify this pattern and reduce the posterior uncertainty (compare s10 with s10sam). Despite the considerable variability in magnitude for the biogeographic Amazon, the atmospheric constraint tends to adjust priors with a positive sign, shifting them towards a smaller source (e.g., SIB4-1Pg) or even turning them into a sink with a negative $F_{NetLand}$ (e.g., VPRM-1Pg, VPRM-0.5Pg, and SIB4-0.5Pg). Furthermore, in the global inversion when adding South American data, the resulting fluxes closely follow the sign and magnitude of those obtained in the regional inversion. Therefore, we contend that information suggesting a sink-source gradient between the Amazon and the 'Cerrado & Caatinga' is embedded in the atmospheric measurement record, even with the inherent limitations in adjusting certain individual priors.*

**[RC1.23] L356**: For Brazilian Shield Moist Forests, did you mean to say movement from neutral to net release of 0.03? That is clearer than "there is a shift in sign".

*[**AR1.23**]: The "shift in sign" was referring not only to the 'Brazilian Shield Moist Forests' (from neutral to positive) but also to the 'Guianan Shield Moist Forests'. However, given that we shortened the manuscript, this was one of the paragraphs that we decided to leave out. The main message of this paragraph (the effect of adding SAM station) was added to Section 3.2 in the revised manuscript. The new text added is shown in AR1.22.*

**[RC1.24] Table 2**: In title, add "averaged over 2010-2018".

*[**AR1.24**]: We have added the reference period in the caption of Table 2.*

**[RC1.25] L383**: "...between these two regions"— change to ""...between the biogeographic amazon and the cerrado&caatinga"

*[**AR1.25**]: We have added your suggestion and now it reads as follows:* Therefore, we contend that information suggesting a sink-source gradient between the Amazon and the 'Cerrado & Caatinga'

is embedded in the atmospheric measurement record, even with the inherent limitations in adjusting certain individual priors.

**[RC1.26] L407**: Typo – change to Figure 5b

*[AR1.26]: Thanks, we corrected the typo.*

**[RC1.27] Figure 6b**: This is on average across 2010-2018? Specify in caption.

*[AR1.27]: We have added this in the caption. Thanks for the suggestion.*

**[RC1.28] Discussion/Conclusions**: With all the satellite data now available, it would be worthwhile to have a brief discussion on the value those data could add in tropical NLF constraints.

*[AR1.28]: We agree with the reviewer and thus have added a reference to the OCO2 Inversions in the Discussion, it reads now as: To further reduce the uncertainty in this domain, top-down estimates could combine in-situ data with satellite retrievals. The inversions assimilating data from the Orbiting Carbon Observatory 2 (OCO2) [Liu et al., 2017, Crowell et al., 2019, Peiro et al., 2022, Wang et al., 2023] have shown that remotely-sensed $CO_2$ columns can provide a valuable constraint of net carbon exchange in tropical regions. However, the OCO2-inversions are still limited by cloud coverage during the wet season [Massie et al., 2017, Peiro et al., 2022] and the adjustment of the prior can be biased to dry season retrievals [Crowell et al., 2019]. Nevertheless, our results for the response to the El Niño 2015/2016 coincide with the OCO2 inversions [Liu et al., 2017, Crowell et al., 2019, Peiro et al., 2022] in a carbon source in 2015 and 2016. Yet, a direct comparison of the magnitude in those studies to our results is difficult, as the area for South America in Liu et al. [2017] includes parts of the 'Cerrado & Caatinga' regions and in Crowell et al. [2019], Peiro et al. [2022] they divide South America in three parts: 1. Northern South America: including only the north (north of the equator) of the Amazon basin, the Orinoco basin, a part of central America and the Caribbean islands. 2. Southern Tropical South America, which includes a large part of the Cerrado and the southern part of the Amazon and 3. South America Temperate, which includes the Cerrado and Caatinga, extending until the southernmost point on the continent. None of these regions coincide with our regional distribution, therefore, using our domain definition on the OCO2-MIP results should be part of a next study, as it is out of the scope of this one.*

**References**

Sean Crowell, David Baker, Andrew Schuh, Sourish Basu, Andrew R. Jacobson, Frederic Chevallier, Junjie Liu, Feng Deng, Liang Feng, Kathryn McKain, Abhishek Chatterjee, John B. Miller, Britton B. Stephens, Annmarie Eldering, David Crisp, David Schimel, Ray Nassar, Christopher W. O'Dell, Tomohiro Oda, Colm Sweeney, Paul I. Palmer, and Dylan B. A. Jones. The 2015–2016 carbon cycle as seen from OCO-2 and the global in situ network. *Atmospheric Chemistry and Physics*, 19(15):9797–9831, August 2019. ISSN 1680-7316. doi: https://doi.org/10.5194/acp-19-9797-2019. Publisher: Copernicus GmbH.

Luciana V. Gatti, Luana S. Basso, John B. Miller, Manuel Gloor, Lucas Gatti Domingues, Henrique L. G. Cassol, Graciela Tejada, Luiz E. O. C. Aragão, Carlos Nobre, Wouter Peters, Luciano Marani, Egidio Arai, Alber H. Sanches, Sergio M. Corrêa, Liana Anderson, Celso Von Randow, Caio S. C. Correia, Stephane P. Crispim, and Raiane A. L. Neves. Amazonia as a carbon source linked to deforestation and climate change. *Nature*, 595(7867):388–393, July 2021. ISSN 1476-4687. doi: 10.1038/s41586-021-03629-6.

Panagiotis Kountouris, Christoph Gerbig, Christian Rödenbeck, Ute Karstens, Thomas F. Koch, and Martin Heimann. Atmospheric $CO_2$ inversions on the mesoscale using data-driven prior uncertainties: quantification of the European terrestrial $CO_2$ fluxes. *Atmospheric Chemistry and Physics*, 18 (4):3047–3064, March 2018a. ISSN 1680-7316. doi: https://doi.org/10.5194/acp-18-3047-2018.

Panagiotis Kountouris, Christoph Gerbig, Christian Rödenbeck, Ute Karstens, Thomas Frank Koch, and Martin Heimann. Technical Note: Atmospheric $CO_2$ inversions on the mesoscale using data-driven prior uncertainties: methodology and system evaluation. *Atmospheric Chemistry and Physics*, 18(4):3027–3045, March 2018b. ISSN 1680-7316. doi: 10.5194/acp-18-3027-2018.

Junjie Liu, Kevin W. Bowman, David S. Schimel, Nicolas C. Parazoo, Zhe Jiang, Meemong Lee, A. Anthony Bloom, Debra Wunch, Christian Frankenberg, Ying Sun, Christopher W. O'Dell, Kevin R. Gurney, Dimitris Menemenlis, Michelle Gierach, David Crisp, and Annmarie Eldering. Contrasting carbon cycle responses of the tropical continents to the 2015–2016 El Niño. *Science*, 358(6360), October 2017. ISSN 0036-8075, 1095-9203. doi: 10.1126/science.aam5690.

Steven T. Massie, K. Sebastian Schmidt, Annmarie Eldering, and David Crisp. Observational evidence of 3-D cloud effects in OCO-2 CO2 retrievals. *Journal of Geophysical Research: Atmospheres*, 122(13):7064–7085, 2017. ISSN 2169-8996. doi: 10.1002/2016JD026111. URL https://onlinelibrary.wiley.com/doi/abs/10.1002/2016JD026111. _eprint: https://onlinelibrary.wiley.com/doi/pdf/10.1002/2016JD026111.

Saqr Munassar, Christian Rödenbeck, Thomas Koch, Kai U. Totsche, Michał Gałkowski, Sophia Walther, and Christoph Gerbig. NEE estimates 2006–2019 over Europe from a pre-operational ensemble-inversion system. *Atmospheric Chemistry and Physics Discussions*, pages 1–28, November 2021. ISSN 1680-7316. doi: 10.5194/acp-2021-873.

Hélène Peiro, Sean Crowell, Andrew Schuh, David F. Baker, Chris O'Dell, Andrew R. Jacobson, Frédéric Chevallier, Junjie Liu, Annmarie Eldering, David Crisp, Feng Deng, Brad Weir, Sourish Basu, Matthew S. Johnson, Sajeev Philip, and Ian Baker. Four years of global carbon cycle observed from the Orbiting Carbon Observatory 2 (OCO-2) version 9 and in situ data and comparison to OCO-2 version 7. *Atmospheric Chemistry and Physics*, 22(2):1097–1130, January 2022. ISSN 1680-7316. doi: 10.5194/acp-22-1097-2022. URL https://acp.copernicus.org/articles/22/1097/2022/. Publisher: Copernicus GmbH.

Ruben Ramo, Ekhi Roteta, Ioannis Bistinas, Dave van Wees, Aitor Bastarrika, Emilio Chuvieco, and Guido R. van der Werf. African burned area and fire carbon emissions are strongly impacted by small fires undetected by coarse resolution satellite data. *Proceedings of the National Academy of Sciences*, 118(9):e2011160118, March 2021. doi: 10.1073/pnas.2011160118.

Thais M. Rosan, Stephen Sitch, Michael O'Sullivan, Luana S. Basso, Chris Wilson, Camila Silva, Emanuel Gloor, Dominic Fawcett, Viola Heinrich, Jefferson G. Souza, Francisco Gilney Silva Bezerra, Celso von Randow, Lina M. Mercado, Luciana Gatti, Andy Wiltshire, Pierre Friedlingstein, Julia Pongratz, Clemens Schwingshackl, Mathew Williams, Luke Smallman, Jürgen Knauer, Vivek Arora, Daniel Kennedy, Hanqin Tian, Wenping Yuan, Atul K. Jain, Stefanie Falk, Benjamin Poulter, Almut Arneth, Qing Sun, Sönke Zaehle, Anthony P. Walker, Etsushi Kato, Xu Yue, Ana Bastos, Philippe Ciais, Jean-Pierre Wigneron, Clement Albergel, and Luiz E. O. C. Aragão. Synthesis of the land carbon fluxes of the Amazon region between 2010 and 2020. *Communications Earth & Environment*, 5(1):1–15, January 2024. ISSN 2662-4435. doi: 10.1038/s43247-024-01205-0.

C. Rödenbeck, S. Zaehle, R. Keeling, and M. Heimann. History of El Niño impacts on the global carbon cycle 1957–2017: a quantification from atmospheric $CO_2$ data. *Philosophical Transactions of the Royal Society B: Biological Sciences*, 373(1760):20170303, November 2018. ISSN 0962-8436, 1471-2970. doi: 10.1098/rstb.2017.0303.

Christian Rödenbeck. Estimating CO2 sources and sinks from atmospheric mixing ratio measurements using a global inversion of atmospheric transport. Technical report, Max Planck Institute for Biogeochemistry, 2005. URL `https://www.bgc-jena.mpg.de/~christian.roedenbeck/download/2005-Roedenbeck-TechReport6.pdf`.

Jun Wang, Ning Zeng, Meirong Wang, Fei Jiang, Frédéric Chevallier, Sean Crowell, Wei He, Matthew S. Johnson, Junjie Liu, Zhiqiang Liu, Scot M. Miller, Sajeev Philip, Hengmao Wang, Mousong Wu, Weimin Ju, Shuzhuang Feng, and Mengwei Jia. Anomalous Net Biome Exchange Over Amazonian Rainforests Induced by the 2015/16 El Niño: Soil Dryness-Shaped Spatial Pattern but Temperature-dominated Total Flux. *Geophysical Research Letters*, 50(11):e2023GL103379, 2023. ISSN 1944-8007. doi: 10.1029/2023GL103379.

---

## Author Response (AR3)

Reviews from Referee 3 - Received on Feb 1, 2025

The format in which the response is addressed is the following:

1. Black text shows comment of referee. Comments are numerated as RC3.1, for the first comment. The line in the submitted draft referred to by the referee is also shown.

2. *Text in italics show the author's response and it has the same logic for numeration (e.g. AR3.1). If not stated otherwise, here the reference to figures and line numbers are based on the submitted manuscript.*

3. Red text indicates changes to the manuscript. Here the reference to Figures are based on the revised manuscript.

**General Comment**

**[RC3.1]-Review Synopsis 1**: Botia et al. quantified the carbon budget over tropical South America by assimilating comprehensive surface, tower, and aircraft observations from 2010 to 2018 using a regional top-down flux inversion system. With ensemble prior fluxes spanning a wide range of annual carbon fluxes, their study revealed a regional carbon sink-source contrast between the Amazon forest and the savanna region over the Cerrado. Additionally, they examined the dependence of their results on prior fluxes and the water vapor correction applied to aircraft observations. Overall, this is a comprehensive study on regional carbon budget estimates for tropical South America. However, I believe the paper would greatly benefit from reorganizing of the results and a more distilled take-home message, as also recommended by the second reviewer. Please see my main comments below:

*[AR3.1]: We thank the third reviewer for the comments and constructive feedback. Before answering the specific comments, here we want to highlight that the Results were reorganized and shortened, giving more importance to the water vapor correction and its effect on the carbon budget of tropical South America.*

**Specific Comments:**

**[RC3.2]**: I would recommend using the flux inversion results that account for the water vapor correction in the main results (section 3) rather than in the discussion section, as the water vapor contamination is a well-recognized limitation of the original data. The impact of the water vapor correction can still be addressed in the discussion section, particularly when comparing this study's results to those of Gatti et al.. While the water vapor correction has minimal impact on the interannual variability, it significantly affects the absolute flux estimation, which is the focus of this study.

*[AR3.2]: In the submitted manuscript the water vapor Section is part of the main results, see Section 3.3, so it is not shown in the Discussion. However, following this comment we have rearranged the Results section in the revised manuscript. Sections 3.2 and 3.3 have been merged and condensed into a single section, now labeled 3.2. The first thing we report on Section 3.2 is the water vapor effect and in Figure 5 we contrast the $F_{NetLand}$ for the Amazon and 'Cerrado & Caatinga' both with and without the water vapor correction.*

[**RC3.3**]: While the paper qualitatively discusses the comparison with OCO-2 MIP results, as suggested by previous reviewers, I recommend including a more quantitative analysis. The latest OCO-2 V10 MIP results are publicly available and were documented in Byrne et al., 2023. Byrne et al., National CO2 budgets (2015–2020) inferred from atmospheric CO2 observations in support of the global stocktake, Earth Syst. Sci. Data, 15, 963–1004, https://doi.org/10.5194/essd-15-963-2023, 2023.

*[**AR3.3**]: We understand the request of the reviewer, which follows what Reviewer 2 suggested earlier and we agree that such a comparison is an interesting way forward. However, we think that this effort is of limited use here given the scope of our study, which is: 1. understanding the constraint given by in-situ and flask data in CarboScope and 2. Correcting the systematic bias in aircraft data and evaluating the effect of that in posterior fluxes and thus in the carbon budget of tropical South America. Now, we did look at the OCO2-v10 MIP fluxes based on the data suggested by the reviewer (i.e. Byrne et al. [2023]) (see Figure 1 below) and during this exercise we were even more convinced that the comparison is out of the scope of this paper and there are several reasons for this. The first one is that those data span from 2015 to 2020 (inclusive), so there are only 4 years in which the MIP and our results overlap: 2015 to 2018, where 2/4 years were anomalous. Secondly, the differences between our inversions and the OCO2-MIP inversions are many (satellite data, regional vs global inversions, the use data inside the Amazon, transport resolution, inversion method, correlation length scales, amongst others). Thus, to make justice to the work done by the MIP modelers and the work we have done here, such a comparison will require an entire new paper discussing all these differences. That said, the comparison shown in Figure 1 is done for Net Biome Exchange (NBE), which is defined as the sum of Net Ecosystem Exchange (NEE) and carbon emissions from fires, basically NBE = NEE + $F_{Fires}$. This is different from what is presented in the manuscript ($F_{NetLand}$), which includes fossil fuel emissions ($F_{NetLand} = NBE + F_{ff}$), a relatively small flux component in these two regions (based on the EDGAR 4.3 inventory, we get 0.02 PgC/year and 0.04 PgC/year in the Amazon and the 'Cerrado and Caatinga' respectively). The regional flux contrast in the OCO2-MIP (IS and LNLGIS) for 2015 to 2018, differs from our results, with the 'Cerrado and Caatinga' region as a small sink ($-NBE$) and the Amazon as a source of carbon ($+NBE$). Disentangling the reasons of discrepancy between our results and the inverse estimates of OCO2-MIP is out of the scope of this paper, but here we can highlight some hypotheses. There could be structural differences when assimilating in-situ data and satellite data only. The studies of Crowell et al. [2019] and Peiro et al. [2022] reported large differences in flux estimates at continental scales between in-situ- and the OCO2-driven inversions. The reason for these differences was attributed to sparse in-situ network coverage and likely to residual biases in the OCO2 retrievals [Crowell et al., 2019]. Furthermore, [Peiro et al., 2022] argues that the difference in seasonal cycle amplitude between the in-situ and the OCO2 inversions in tropical South America, could probably be due to the seasonal bias induced by clouds in the OCO2 retrievals. The seasonal effect due to clouds could also vary spatially, as cloud cover tends to be more prominent in the northwest of the Amazon. In our system, we do not constrain this area either, therefore we do not discard a dipole effect as a result of network coverage. Nevertheless, as we assimilate data sources that are not subject to very strong seasonal cloud biases and actually have calibrated data over the continent, we believe that in the regions with a large uncertainty reduction our results should fundamentally be more trustworthy.*

[**RC3.4**]: Since there are large data gaps in the aircraft observations in 2015-2016, I would suggest caution when comparing the top-down flux inversion results from this study to previous studies during this period.

*[**AR3.4**]: Thanks for the comment. We agree with this caveat and we have added a note of caution in the revised manuscript where we compare the 2015/2016 response to other systems. It is likely that what we obtain in the Inversion during 2015 from August to October, is dominated by the effect of ATTO, as it is the only station available during those months. In light of this, we added a sentence in Section 3.2 warning the reader about the response to the 2015/2016 drought: Therefore, the findings*

[Figure]

Figure 1: Carbon budget (NBE) for the biogeographic Amazon (a) and the Cerrado & Caatinga (b) regions.

*of our sensitivity tests support the hypothesis in Gatti et al. (2023) that the water vapor bias mainly affects the absolute annual flux magnitudes. In both cases, with and without correction, our estimates of the total carbon loss to the atmosphere in the Amazon during 2015 and 2016, are lower (from 0.15 to 0.3 PgC) than other studies (Liu et al., 2017; Gloor et al., 2018). However, it should be noted that the response to the 2015-2016 drought in our system must be interpreted with caution, as there are large gaps in the observational record during 2015 and 2016. Having this in mind, our total net flux is closer to the 0.5 ± 0.3 PgC of Gloor et al., (2018), but note that they used a time period from September 2015 to June 2016 and the area they refer to as Amazonia is not clearly defined. Compared to 1.6 ± 0.29 PgC in Liu et al., (2017), our estimates are much lower, but the difference in area is large, as they refer to tropical South America, including parts of the 'Cerrado & Caatinga' biomes and central America. Additionally, in the Discussion we have added: Yet, a direct comparison of the magnitude in those studies to our results is challenging for the following reasons. First, the response in our system could be biased to ATTO as there was a large gap in the aircraft data in 2015 and 2016.*

[**RC3.5**]: The dependence of the posterior fluxes on the prior fluxes is striking (Figure 4). Therefore, for the regional flux contrast shown in Figure 5, I recommend including a plot of the regional flux contrast in the prior fluxes, which could be added in the supplementary material. Additionally, it would be valuable to discuss the extent to which the posterior regional flux contrast is influenced by the prior fluxes.

*[**AR3.5**]: We thank the reviewer for this comment. The suggested figure has been added to the Supplementary material (Figure A15) and also here, see Figure 2, below. If we compare the posterior flux contrast for the whole ensemble (Figure 5, on the submitted manuscript) with the prior ensemble flux contrast (Figure 2 in this document and Figure A15 in the revised manuscript), we conclude that the change from prior to posterior on an ensemble level, results in a solid convergence of a sink of carbon ($F_{NetLand}$) for the Amazon and a source of carbon for the 'Cerrado & Caatinga'. Now, when looking at particular ensemble members, the dependence on the prior is more evident. For example, this is evident for VPRM, FLUXCOM and X-BASE$_{NEE}$, but here is where the "flat" experiment with VPRM becomes relevant, as the mean in the prior is zero for both regions and the direction*

*of the adjustment coincides with the pattern for the whole ensemble (Amazon* $->$ *sink, 'Cerrado & Caatinga'* $->$ *source). In Section 3.2 of the revised manuscript we have added the following text: Despite variations in magnitude, the inversion consistently shifts priors toward a smaller Amazonian source (e.g., SIB4-1Pg) or even a sink (e.g., VPRM-1Pg, VPRM-0.5Pg, and SIB4-0.5Pg). The prior vs. posterior flux contrast (Figures ??, ??) confirms a robust sink-source gradient, embedded in the atmospheric measurements, despite limitations in adjusting individual priors.*

[Figure]

Figure 2: Prior carbon budget for the biogeographic Amazon (a) and the Cerrado & Caatinga (b) regions. The prior flux component shown with a vertical bar (NEE), result from subtracting the $F_{fire}$ and $F_{ff}$ from the prior $F_{NetLand}$ (shown with the markers).

[**RC3.6**]: Figure A9 and A10 plot annual mean prior and posterior fluxes from some of the models. I would recommend the authors to include all the prior fluxes (except the flat one) and posterior fluxes. The fact that the posterior fluxes all show this consistent east-west contrast pattern (A10) is very promising.

*[**AR3.6**]: In Figure A9 we did not add the prior of s10 and s10sam because in these inversions the prior is zero; in the original submitted manuscript this was mentioned in Section 2.1.6. In the newly revised manuscript, we added the additional non-zero priors and all posteriors, as the reviewer recommended. The two plots are shown here for practicality as well (Figures 3 and 4).*

[Figure]

Figure 3: Prior mean NEE over 2010-2018 for several of the models used. Note the different range in the colorbar.

[Figure]

Figure 4: Posterior mean NBE over 2010-2018 for several of the models used. Note the different range in the colorbar.

**References**

Brendan Byrne, David F. Baker, Sourish Basu, Michael Bertolacci, Kevin W. Bowman, Dustin Carroll, Abhishek Chatterjee, Frédéric Chevallier, Philippe Ciais, Noel Cressie, David Crisp, Sean Crowell, Feng Deng, Zhu Deng, Nicholas M. Deutscher, Manvendra K. Dubey, Sha Feng, Omaira E. García, David W. T. Griffith, Benedikt Herkommer, Lei Hu, Andrew R. Jacobson, Rajesh Janardanan, Sujong Jeong, Matthew S. Johnson, Dylan B. A. Jones, Rigel Kivi, Junjie Liu, Zhiqiang Liu, Shamil Maksyutov, John B. Miller, Scot M. Miller, Isamu Morino, Justus Notholt, Tomohiro Oda, Christopher W. O'Dell, Young-Suk Oh, Hirofumi Ohyama, Prabir K. Patra, Hélène Peiro, Christof Petri, Sajeev Philip, David F. Pollard, Benjamin Poulter, Marine Remaud, Andrew Schuh, Mahesh K. Sha, Kei Shiomi, Kimberly Strong, Colm Sweeney, Yao Té, Hanqin Tian, Voltaire A. Velazco, Mihalis Vrekoussis, Thorsten Warneke, John R. Worden, Debra Wunch, Yuanzhi Yao, Jeongmin Yun, Andrew Zammit-Mangion, and Ning Zeng. National $CO_2$ budgets (2015–2020) inferred from atmospheric $CO_2$ observations in support of the global stocktake. *Earth System Science Data*, 15(2):963–1004, March 2023. ISSN 1866-3508. doi: 10.5194/essd-15-963-2023. URL https://essd.copernicus.org/articles/15/963/2023/. Publisher: Copernicus GmbH.

Sean Crowell, David Baker, Andrew Schuh, Sourish Basu, Andrew R. Jacobson, Frederic Chevallier, Junjie Liu, Feng Deng, Liang Feng, Kathryn McKain, Abhishek Chatterjee, John B. Miller, Britton B. Stephens, Annmarie Eldering, David Crisp, David Schimel, Ray Nassar, Christopher W. O'Dell, Tomohiro Oda, Colm Sweeney, Paul I. Palmer, and Dylan B. A. Jones. The 2015–2016 carbon cycle as seen from OCO-2 and the global in situ network. *Atmospheric Chemistry and Physics*, 19(15):9797–9831, August 2019. ISSN 1680-7316. doi: https://doi.org/10.5194/acp-19-9797-2019. Publisher: Copernicus GmbH.

Hélène Peiro, Sean Crowell, Andrew Schuh, David F. Baker, Chris O'Dell, Andrew R. Jacobson, Frédéric Chevallier, Junjie Liu, Annmarie Eldering, David Crisp, Feng Deng, Brad Weir, Sourish Basu, Matthew S. Johnson, Sajeev Philip, and Ian Baker. Four years of global carbon cycle observed from the Orbiting Carbon Observatory 2 (OCO-2) version 9 and in situ data and comparison to OCO-2 version 7. *Atmospheric Chemistry and Physics*, 22(2):1097–1130, January 2022. ISSN 1680-7316. doi: 10.5194/acp-22-1097-2022. URL https://acp.copernicus.org/articles/22/1097/2022/. Publisher: Copernicus GmbH.